# Comparative cellular analysis of motor cortex in human, marmoset and mouse

The primary motor cortex (M1) is essential for voluntary fine-motor control and is functionally conserved across mammals[1]. Here, using high-throughput transcriptomic and epigenomic profiling of more than 450,000 single nuclei in humans, marmoset monkeys and mice, we demonstrate a broadly conserved cellular makeup of this region, with similarities that mirror evolutionary distance and are consistent between the transcriptome and epigenome. The core conserved molecular identities of neuronal and non-neuronal cell types allow us to generate a cross-species consensus classification of cell types, and to infer conserved properties of cell types across species. Despite the overall conservation, however, many species-dependent specializations are apparent, including differences in cell-type proportions, gene expression, DNA methylation and chromatin state. Few cell-type marker genes are conserved across species, revealing a short list of candidate genes and regulatory mechanisms that are responsible for conserved features of homologous cell types, such as the GABAergic chandelier cells. This consensus transcriptomic classification allows us to use patch–seq (a combination of whole-cell patch-clamp recordings, RNA sequencing and morphological characterization) to identify corticospinal Betz cells from layer 5 in non-human primates and humans, and to characterize their highly specialized physiology and anatomy. These findings highlight the robust molecular underpinnings of cell-type diversity in M1 across mammals, and point to the genes and regulatory pathways responsible for the functional identity of cell types and their species-specific adaptations.

Single-cell transcriptomic and epigenomic methods have been effective in elucidating the cellular makeup of complex brain tissues from patterns of gene expression and underlying regulatory mechanisms[2–6]. In the mouse and human neocortex, diverse neuronal and non-neuronal cell types can be defined[2,3,5,7] by their distinct transcriptional profiles and regions of accessible chromatin or of DNA methylation (DNAm)[4,8], and can be aligned between species[3,9–11] on the basis of these profiles. Studies such as these have shown the feasibility of quantitatively studying the evolution of cell types, but have limitations: different cortical regions have been profiled in humans and mice; different sets of transcripts have been captured with single-cell and single-nucleus assays; and transcriptomic and epigenomic studies have mostly been carried out independently.

The primary motor cortex (M1, also known as MOp in mice) is an ideal region with which to address questions about cellular evolution in rodents and primates. M1 is essential for fine-motor control and is functionally conserved across mammals[1]. The layer 5 (L5) region of carnivore and primate M1 contains specialized 'giganto-cellular' corticospinal neurons (Betz cells in primates[12–16]) with distinctive action-potential properties that support a high conduction velocity[17–19]. Some Betz cells synapse directly onto spinal motor neurons, unlike rodent corticospinal neurons, which synapse indirectly via spinal interneurons[20]. These observations suggest that Betz cells possess species-adapted intrinsic mechanisms to support rapid communication that should be reflected in their molecular signatures. To explore the evolutionary conservation and divergence of M1 cell types and their underlying molecular

regulatory mechanisms, we analysed single-nucleus transcriptomic and epigenomic data from mouse, marmoset, macaque and human M1.

## Multi-omic taxonomies of cell types

To characterize the molecular diversity of M1 neurons and non-neuronal cells, we applied single-nucleus transcriptomic assays (plate-based SMART-seq v4 (SSv4) and droplet-based Chromium v3 (Cv3) RNA sequencing) and epigenomic assays (single-nucleus methylcytosine sequencing 2 (snmC-seq2) and single-nucleus chromatin accessibility and messenger RNA expression sequencing (SNARE–seq2)) to isolated M1 samples from human, marmoset and mouse brains (Extended Data Fig. 1a–d); we also applied Cv3 to M1 L5 from macaque brains. Single nuclei were dissociated from all layers combined or from individual layers (in the case of human SSv4 assays), and sorted using the neuronal marker NeuN to enrich cellular input to roughly 90% neurons and 10% non-neuronal cells (Extended Data Fig. 1e). Datasets from mice are reported in a companion paper[5]. The median detection of neuronal genes in humans was higher when we used SSv4 (7,296 genes) as compared with Cv3 (5,657 genes), partially because of the 20-fold greater read depth, and detection was lower in marmosets (4,211) and mice (5,046) when using Cv3 (Extended Data Fig. 1f–m).

For each species, we defined a diverse set of neuronal and non-neuronal clusters of cell types on the basis of unsupervised clustering of snRNA-seq datasets (Extended Data Fig. 1n–r and Supplementary Tables 1, 2). We organized cell types into hierarchical taxonomies

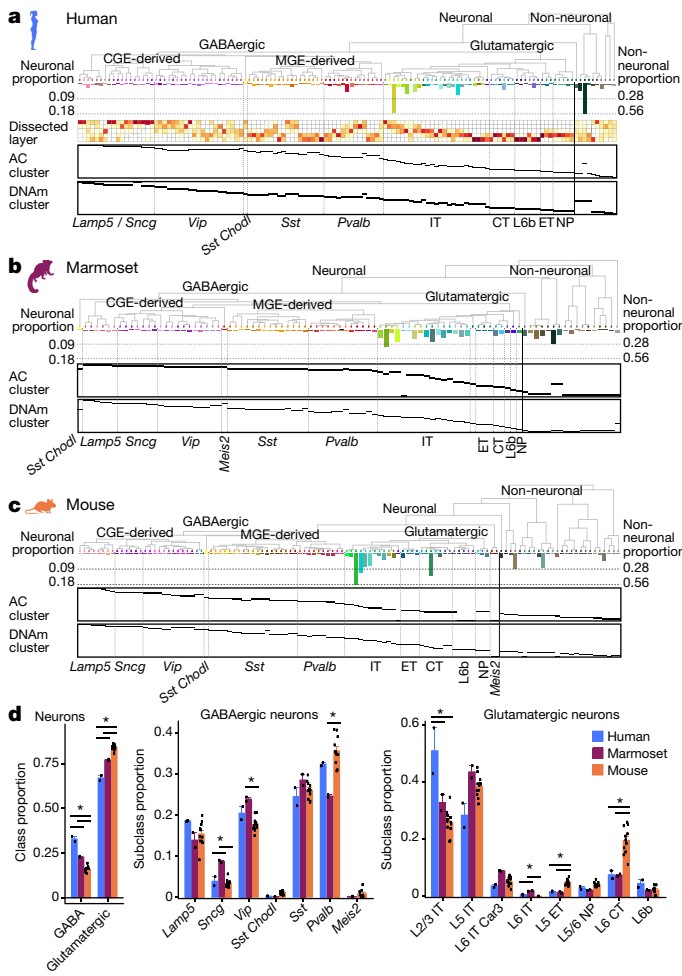

**Fig. 1 | Molecular taxonomy of cell types in the primary motor cortex (M1) of humans, marmosets and mice. a–c**, Dendrograms showing cell-type clusters defined by RNA sequencing (RNA-seq; using Cv3) for humans (**a**), marmosets (**b**) and mice (**c**), annotated with the cluster proportions of total neuronal or non-neuronal cells and (for humans) with dissected layers (L1–L6). RNA-seq clusters mapped to clusters of accessible chromatin (AC) and DNAm. **d**, Relative proportions of some neuronal cell types were significantly different between species, based on analysis of variance (ANOVA) followed by Tukey's HSD two-sided tests (degrees of freedom = 13; *$P < 0.05$ (Bonferonni-corrected)). Data in **d** are means ± s.d., and points represent individual donor specimens for humans (*n* = 2), marmosets (*n* = 2), and mice (*n* = 12). Marmoset silhouettes are from www.phylopic.org (public domain).

on the basis of transcriptomic similarities (Fig. 1a–c, Extended Data Fig. 2 and Supplementary Table 3). As previously described for temporal cortex (middle temporal gyrus, MTG)[3], taxonomies were broadly conserved across species, and neuronal subclasses reflected developmental origins and targets of long-range neuronal projections. Cell-type labels include the dissected layer (if available), major class, subclass marker gene and most-specific marker gene (Supplementary Tables 4–6). GABAergic (γ-aminobutyric acid-producing) types were uniformly rare (fewer than 4.5% of neurons), whereas glutamatergic and non-neuronal types were more variable in number (0.01–18.4% of neurons and 0.15–56.2% of non-neuronal cells, respectively). Finally, independent clustering of epigenomic data resulted in diverse clusters that were associated one-to-one with RNA clusters or at a slightly higher level in the hierarchy on the basis of shared marker expression.

Single-nucleus sampling provides a relatively unbiased survey of cellular diversity[3,21] and enables an estimation of cell-type frequencies. Consistent with histological measurements (reviewed in ref. [22]), we

identified twice as many GABAergic neurons in human M1 (33%) as in mouse M1 (16%), and an intermediate proportion (23%) in marmosets (Fig. 1d). L2 and L3 intratelencephalic neurons were significantly more common in humans than in marmosets and mice (Fig. 1d)[23], while L6 corticothalamic and L5 extratelencephalic neurons, including corticospinal neurons and Betz cells in primate M1, were significantly rarer in primates than in mice.

## Consensus M1 taxonomy across species

We integrated Cv3 datasets across species on the basis of shared patterns of coexpression for GABAergic neurons (Fig. 2 and Extended Data Fig. 3), glutamatergic neurons (Extended Data Fig. 4) and non-neuronal cells (Extended Data Fig. 5). GABAergic nuclei were well mixed across species and segregated into six subclasses (Fig. 2a); 17 to 54 subclass markers were conserved across species (Fig. 2b, c, Extended Data Fig. 3a and Supplementary Tables 7, 8), while most markers had enriched expression in only one species. To establish a consensus taxonomy of cross-species clusters, we over-split the integrated space (Extended Data Fig. 3b) and merged clusters until they included nuclei from all species. We defined 24 GABAergic cell types on the basis of consistent overlap of clusters across species (Fig. 2d–f); these cell types had conserved marker genes (Extended Data Fig. 3c) and high classification accuracy (Extended Data Fig. 3d, e and Supplementary Table 9). Distinct consensus types such as ChC and Sst-*Chodl* were more robust (mean area under the receiving operating characteristic (AUROC) curve = 0.99 within species, 0.88 across species) than were closely related types such as *Sncg* and *Sst* subtypes (mean AUROC = 0.84 within species, 0.50 across species). Most types were enriched in the same layers in humans and mice (Fig. 2g), with notable differences. ChCs were enriched in L2/3 in mice and in all layers in humans, as was seen in MTG[3]. Sst-*Chodl* was restricted to L6 in mice and was also found in L1 and L2 in humans, consistent with the reported sparse expression of *SST* in L1 in human but not mouse cortex[24].

More consensus clusters could be resolved by pairwise alignment between humans and marmosets than between either of these primates and mice, particularly for *Vip* subtypes (Fig. 2h and Extended Data Fig. 3f, g). Genes related to neuronal connectivity and signalling were most informative of cell-type identity (Fig. 2i), and showed similar classification performance when trained and tested in the same species (*r* values of greater than 0.95) but reduced performance when trained and tested in different species (62% as high in humans and marmosets, and 40% in primates and mice). Therefore, similar genes show selectivity for subsets of cell types across species, yet individual genes often change the specific cell types in which they are expressed.

Glutamatergic neuron subclasses also aligned well across species, with 6–66 conserved markers and many more species-enriched markers (Extended Data Fig. 4a–c and Supplementary Tables 10, 11). We defined a consensus taxonomy of 13 types as above, which was similarly robust to the GABAergic taxonomy (GABAergic AUROC = 0.86; glutamatergic, 0.85; Extended Data Fig. 4i, j and Supplementary Table 9) but had fewer conserved markers (Extended Data Fig. 4h). Human and marmoset consensus types shared more markers (25%) with each other than with mice (16%) for 13 of 14 neuronal subclasses (Fig. 2b and Extended Data Fig. 4b). Moreover, humans and marmosets could be aligned at somewhat higher resolution (Extended Data Fig. 4k), particularly for L5/6 near-projecting and L5 intratelencephalic subclasses.

Non-neuronal consensus types were clearly defined by conserved marker genes, except for rare or immature types that were undersampled in humans and marmosets (Extended Data Fig. 5a–d). The human cortex contains several morphologically distinct astrocyte types[25]. We reported two transcriptomic clusters in human MTG that corresponded to protoplasmic and interlaminar (ILA) astrocytes[3], and we validated these types in M1 by in situ hybridization (ISH; Extended Data Fig. 5f, g). We identified a third type, Astro L1-6 *FGFR3 AQP1*, that expresses

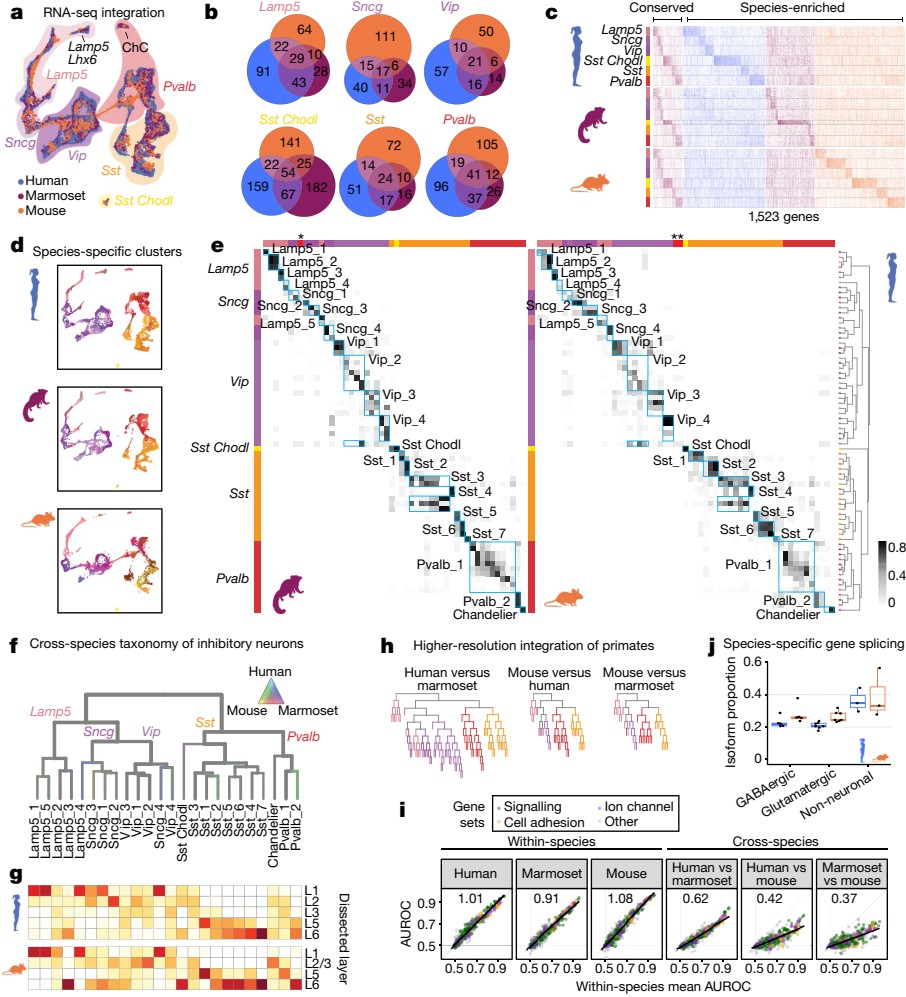

**Fig. 2 | Homology of GABAergic neurons across species. a**, Uniform manifold approximation and projection (UMAP) dimensional reduction of integrated snRNA-seq data. **b**, Venn diagrams showing subclass DEGs shared across species. **c**, Heat map showing expression of conserved and species-enriched DEGs. **d**, UMAP from **a**, separated by species and coloured by within-species clusters. **e**, Proportion of nuclei that overlap between human (rows, ordered as in Fig. 1a) and marmoset or mouse clusters in the integrated space. Asterisks mark the Meis2 subclass. **f**, Dendrogram showing consensus clusters of GABAergic neurons, with branches coloured by species mixture (grey, well mixed). **g**, Consensus cluster layers in humans (top) and mice (bottom). **h**, Dendrograms showing pairwise species integrations, coloured by subclass. **i**, Average classification performance (chance = 0.5) of gene sets for cell types within and between species. Linear regression fits are shown with black lines (slope at top left). **j**, Proportions of isoforms with a change in usage between species (humans, $n = 15$; mice, $n = 15$ cell subclasses). Box plots extend from 25th to 75th percentiles; central lines represent median value; whiskers extend to 1.5 times the interquartile interval.

*APQ4* and *TNC* and corresponds to fibrous astrocytes in white matter. Non-neuronal gene expression diverged with evolutionary distance: ILAs (Astro_1) had 560 differentially expressed genes (DEGs) (Wilcox test; false discovery rate (FDR) less than 0.01; log-transformed fold change greater than 2) between humans and mice, and only 221 DEGs between humans and marmosets (Extended Data Fig. 5e).

Primates had a unique oligodendrocyte population (Oligo *SLC1A3 LOC103793418* in marmosets and Oligo L2-6 *OPALIN MAP6D1* in humans) that was not a distinct cluster in mice (Extended Data Fig. 5c). Surprisingly, this oligodendrocyte population clustered with glutamatergic neurons (Extended Data Fig. 1a, b) and was associated with neuronal transcripts such as *NPTX1*, *OLFM3* and *GRIA1* (Extended Data Fig. 5h). This was not an artefact, as fluorescent in situ hybridization (FISH) for markers of this type (*SOX10* and *ST18*) co-localized with neuronal markers in the nuclei of cells that were sparsely distributed across many layers of human and marmoset M1 (Extended Data Fig. 5i). This type may represent an oligodendrocyte population that has phagocytosed parts of neurons and accompanying transcripts, similar to the reported phagocytic function of some oligodendrocyte precursor cells[26].

To assess the usage of differential isoforms between humans and mice, we used SSv4 data with full transcript coverage and estimated isoform abundance in cell subclasses. Remarkably, 25% of moderately expressed isoforms showed a more than ninefold change in usage between species, and isoform switching was more common in non-neuronal than in neuronal subclasses (Fig. 2j, Extended Data Fig. 3h and Supplementary Table 12). For example, β2-chimaerin (*CHN2*) was highly expressed in L5/6 near-projecting cells, and the short isoform was dominant in mice, while longer isoforms were also expressed in humans (Extended Data Fig. 3i).

## Cell-type-specific epigenetic regulation

Epigenomic profiling of M1 cell types can reveal regulatory mechanisms of transcriptomic identity. To profile the accessible chromatin of RNA-defined cell populations from humans and marmosets, we used SNARE–seq2 (refs. [6,27,28]; Extended Data Fig. 6a, b and Supplementary Table 13). We defined 'RNA-level' clusters by mapping single nuclei to human and marmoset taxonomies (Fig. 1a, b) on the basis of

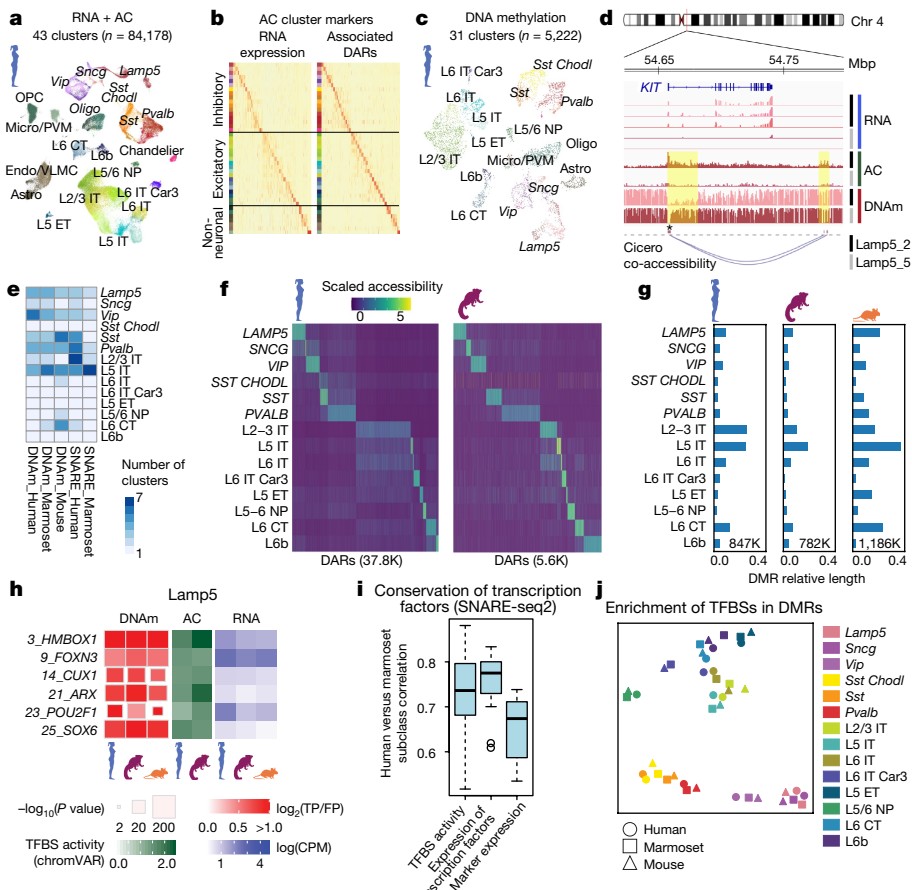

**Fig. 3 | Epigenomic profiling reveals gene-regulatory processes that define M1 cell types. a**, UMAP showing human M1 SNARE–seq2 data, labelled by cell subclass and AC cluster (colours). Astro, astrocyte; Car3, *CAR3* gene; CT, corticothalamic cell; ET, extratelencephalic cell; IT, intratelencephalic cell; micro, microglia; NP, near-projecting; oligo, oligodendrocyte; OPC, oligodendrocyte precursor; PVM, perivascular macrophage. **b**, Heat maps showing the expression of markers of AC clusters and associated DARs. **c**, UMAP showing DNAm data from human M1, labelled by subclass and cluster (colour). **d**, Human genome tracks, showing AC and the hypomethylation (mCG) of DNA (DNAm) near *KIT* selectively in consensus cluster Lamp5_2. Co-accessible chromatin regions were identified by Cicero. **e**, Number of cell types identified for each technology and species varies across subclasses.

**f**, Heat maps showing the activity of human and marmoset subclass DARs (K, thousands). **g**, Barplots showing the relative lengths of hypomethylated DMRs for subclasses across species, normalized by cytosine coverage genome-wide. Total DMRs are shown at the bottom. **h**, Left, conserved enrichment of transcription-factor motifs in DMRs (DNAm); TFBS activities in AC (using chromVAR); and expression of transcription factors, for Lamp5 neurons. CPM, counts per million; FP, false positive; TP, true positive. **i**, Correlations of cell subclasses (*n* = 13) between species for SNARE–Seq2 TFBS activities and expression of transcription factors and markers. Box plots extend from 25th to 75th percentiles; central lines represent medians; whiskers extend over 1.5 times the interquartile interval. **j**, *t*-distributed stochastic neighbour embedding (*t*-SNE) plot showing enrichment of TFBSs in DMRs.

expression similarity; predicted cell-type identities were consistent with independent clustering (Extended Data Fig. 6c–f). Some RNA-level clusters could not be predicted robustly from profiles of accessible chromatin and were iteratively merged (Fig. 3a and Extended Data Fig. 6g–k). Clusters at the level of accessible chromatin had similar coverage across donors, and inferred gene activity was highly correlated with RNA expression (Extended Data Fig. 7a–f). To identify cell-type-specific candidate cis-regulatory elements, we determined differentially accessible regions (DARs) in clusters identified from accessible chromatin (Fig. 3b) and RNA information (Extended Data Fig. 7g, h and Supplementary Table 14). These results highlight the ability of SNARE–seq2 to characterize accessible chromatin at higher cell-type resolution than available from accessible chromatin alone. Distal regulatory elements were linked to marker genes by predicting marker expression on the basis of features of DARs located within 500 kilobases of transcriptional start sites (Fig. 3b, Extended Data Fig. 7i and Supplementary Table 14).

To further characterize the epigenomic landscape of M1 cell types, we profiled DNAm from humans, marmosets and mice[29] using snmC-seq2

(ref. [30]) (Extended Data Fig. 8, Supplementary Table 15). On the basis of DNAm profiles in CpG (CG methylation, or mCG) and non-CpG (CH methylation, or mCH) sites, we grouped single nuclei into 31 DNAm clusters in humans, 36 in marmosets and 42 in mice (Fig. 3c and Extended Data Fig. 8a, b) that correspond to transcriptomic cell types (Extended Data Fig. 8e–g). Notably, we identified more *Vip* neuron types in human M1 by using DNAm rather than accessible chromatin, despite profiling only 5% as many nuclei with snmC-seq2. DNAm clusters could be robustly discriminated and had distinct marker genes based on DNAm signatures for neurons (mCH) or non-neuronal cells (mCG) (Extended Data Fig. 8d and Supplementary Table 16). Differentially methylated regions (DMRs) were determined for each cell type versus all other types, and overlapped only partially with DARs (Extended Data Fig. 8c, i, j)[5]. The intersection of these genomic regions may guide the identification of regulatory elements of marker genes such as *KIT*, which is expressed in the consensus type Lamp5_2 (Fig. 3d) and corresponds to 'rosehip' GABAergic neurons in humans[24].

To gain insight into the evolutionary conservation of regulatory processes that define M1 cell types, we focused on neuronal subclasses

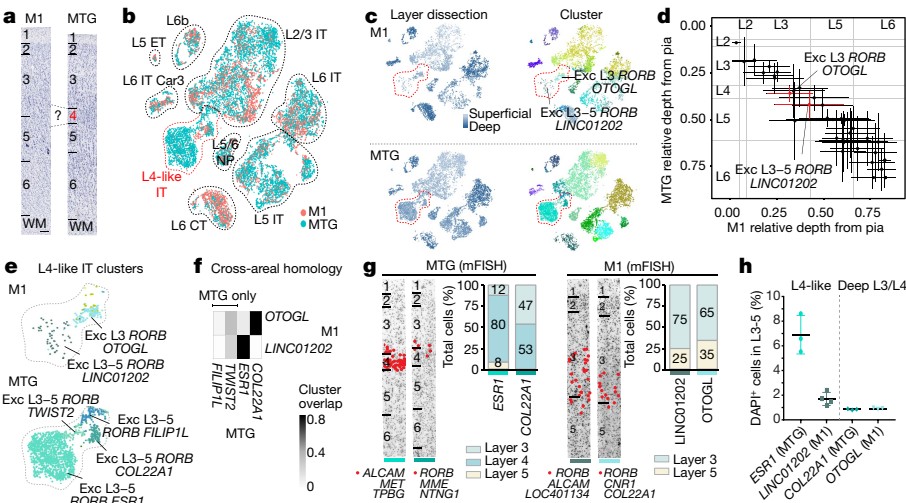

**Fig. 4 | L4-like neurons in M1. a**, L4 is present in human MTG not M1, on the basis of cytoarchitecture in Nissl-stained sections. **b**, *t*-SNE plot of integrated snRNA-seq from M1 and MTG glutamatergic neurons. **c**, Nuclei annotated on the basis of the relative depth of the dissected layer and within-area cluster. Two clusters from superficial layers are labelled (red dotted outline). **d**, Estimated relative depth from pia (mean ± s.d.) of M1 glutamatergic clusters (*n* = 44) and closest matching MTG neurons. Approximate layer boundaries are indicated (grey lines). **e**, Magnified view of L4-like clusters in M1 and MTG.

**f**, Overlap of M1 and MTG clusters in integrated space identifies homologous and MTG-specific clusters. **g**, Multicolour FISH (mFISH) quantifies differences in layer distributions for homologous types between M1 and MTG. Cells (red dots) in each cluster were labelled using the markers listed below each representative inverted image of a DAPI-stained cortical column. DAPI, 4′,6-diamidino-2-phenylindole. **h**, ISH-estimated frequencies (mean ± s.d.) of homologous clusters (*ESR1*, *n* = 3; *LINC01202*, *n* = 4; *COL22A1*, *n* = 3; *OTOGL*, *n* = 3 samples).

(Fig. 3e). Subclass DARs (Fig. 3f) and DMRs (Fig. 3g and Extended Data Fig. 8h) had conserved proportions, although fewer DARs and DMRs were detected for rare subclasses owing to reduced statistical power[5]. DMRs and DARs showed low and variable overlap (median 11%; range 0–32%) across subclasses (Extended Data Fig. 8i, j). Only 5% of human and marmoset subclass DARs were shared between species, compared with 25% of RNA marker genes. To identify transcription factors that may mediate cell subclass identity, we tested for differential activities of transcription-factor-binding sites (TFBSs) in accessible chromatin (Supplementary Table 17) and for significant TFBS enrichments in DMRs (Extended Data Fig. 9 and Supplementary Tables 18, 19). Although many DARs and DMRs were species specific, TFBS enrichments and transcription-factor marker expression were remarkably conserved and distinct between subclasses (Fig. 3h–j and Extended Data Fig. 9). Therefore, evolutionary divergence of expression may be driven partly by genomic relocation of TFBS motifs that are bound by a conserved transcription-factor regulatory network[31].

## L4-like neurons in human M1

M1 lacks L4 as defined by a thin band of densely packed 'granular' neurons that is present in other cortical areas, such as MTG (Fig. 4a). However, prior studies have identified L4-like neurons in M1 on the basis of synaptic properties in mice[32] and cell morphology and lack of SMI-32 labelling[33] and expression of *RORB*[34] (an L4 marker) in primates. To address the potential existence of L4-like neurons in human M1 from a transcriptomic perspective, we integrated snRNA-seq data from agranular M1 and granular MTG, where we previously described multiple L4 glutamatergic neuron types[3]. This alignment revealed a broadly conserved cellular architecture between M1 and MTG (Fig. 4b, c and Extended Data Fig. 10), including M1 neuron types Exc L3 *RORB OTOGL* and Exc L3-5 *RORB LINC01202* that map closely to MTG neurons in deep L3 and L4 (Fig. 4c).

We found transcriptomically similar cell types in similar layers in M1 and MTG across the full cortical depth (Fig. 4d). *OTOGL* and *LINC01202* matched MTG types *COL22A1* and *ESR1*, respectively, whereas there

were no matches for MTG L4 types *FILIP1L* and *TWIST2* (Fig. 4e, f). FISH analysis validated that the M1 *LINC01202* type was sparser and more widely distributed across L3 and L5 than the MTG *ESR1* type, which was restricted to L4 (Fig. 4g, h). By contrast, the M1 *OTOGL* and MTG *COL22A1* types were located in deep L3 and superficial L5 or L4, respectively. Thus, M1 contains cells with L4-like properties, but with less diversity and much sparser representation.

## Core molecular identity of chandelier cells

Canonical features of cell types are likely to be the consequence of conserved transcriptomic and epigenomic features. Focused analysis of *Pvalb*-expressing GABAergic neurons illustrates the power of these data to predict such gene–function relationships. Cortical *Pvalb*-expressing neurons—comprising basket cells and ChCs—share fast-spiking electrical properties but have distinctive morphologies (Fig. 5a), including ChCs that target axon initial segments (AISs). To reveal conserved transcriptomic hallmarks of ChCs, we identified 357 DEGs in ChCs versus basket cells in at least one species. Humans and marmosets shared a significantly (*P* = 0.009; chi-squared test) higher percentage of DEGs (23%) than either species did with mice (average 15%) (Fig. 5b and Supplementary Table 20). Remarkably, only 25 DEGs were conserved across all three species, including *UNC5B* (which encodes a netrin receptor that may contribute to AIS targeting) and three transcription-factor genes (*RORA*, *TRPS1* and *NFIB*) (which were among the top 1% of the most highly expressed transcription-factor genes in ChCs) (Fig. 5c).

To determine whether ChCs had enriched epigenomic signatures for *RORA* and *NFIB* (*TRPS1* lacked motif data), we compared DMRs between ChCs and basket cells. In all species, *RORA* and *NFIB* showed gene-body hypomethylation (mCH) in ChCs but not in basket cells (Fig. 5d), consistent with differential expression. To discern whether these transcription factors may preferentially bind to DNA in ChCs, we tested for the enrichment of transcription-factor motifs in hypomethylated (mCG) DMRs and for transcription-factor activity in sites of accessible chromatin genome-wide. We found that the *RORA* motif was significantly enriched in DMRs in primates (Fig. 5d) and showed

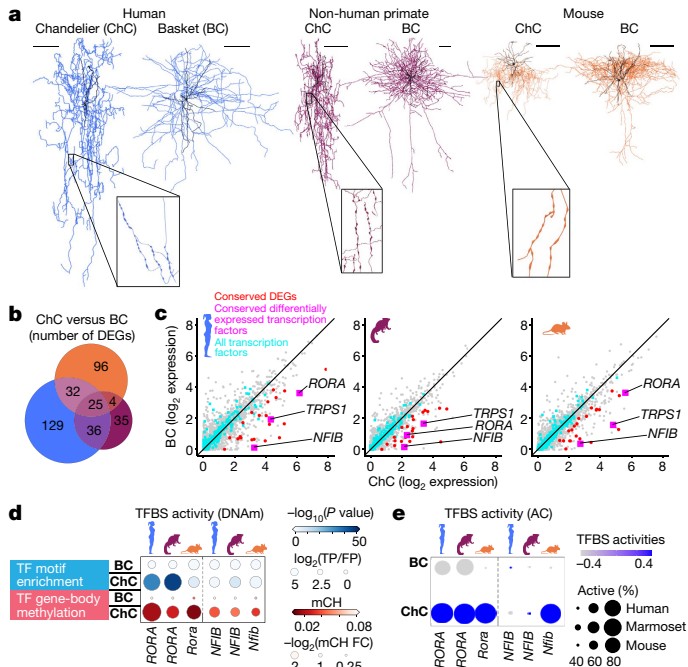

**b** ChC versus BC (number of DEGs)

**c** 
Conserved DEGs
Conserved differentially expressed transcription factors
All transcription factors

**Fig. 5 | Chandelier neurons have a core set of conserved molecular features.** **a**, Representative ultrastructural reconstructions of ChCs and basket cells (BCs) across species. Scale bars, 100 μm. Insets show higher magnifications of unique ChC synapse specializations, axon cartridges. **b**, Venn diagram showing ChC-enriched DEGs shared across species. **c**, Scatter plots showing BC and ChC expression of all genes (grey), all transcription factors (cyan) and conserved ChC markers (non-transcription factors, red; transcription factors, magenta) for each species. **d**, Dot plots showing the enrichment of transcription-factor (TF) motifs in genome-wide mCG DMRs and hypomethylation of transcription-factor gene bodies (mCH) for BCs and ChCs across species. FC, fold change. **e**, Dot plot showing TFBS activities in AC for BCs and ChCs across species.

high activity in accessible-chromatin sites of ChCs in all species (Fig. 5e and Supplementary Table 14). Moreover, 60 of 357 DEGs contained an ROR-binding motif in DMRs and in regions of accessible chromatin in at least one species, further implicating *RORA* in contributing to gene regulatory networks that determine the unique attributes of ChCs.

## Specialization of L5 extratelencephalic neurons

Using snRNA–seq, we found that L5 extratelencephalic and intratelencephalic subclasses of neurons could be aligned across humans, macaques, marmosets and mice in M1 (Extended Data Fig. 11a–d), as previously reported for humans and mice in temporal[3] and fronto-insular cortex[10]. L5 extratelencephalic neurons had more than 250 DEGs distinguishing them from L5 intratelencephalic neurons in each species, and fewer DEGs were shared with greater evolutionary distance (Fig. 6a, b and Supplementary Table 21). Interestingly, many primate-specific extratelencephalic-enriched genes (Fig. 6c) showed gradually increasing extratelencephalic specificity in species that are more closely related to humans. To explore this idea of gradual evolutionary change further, we identified 131 genes with increasing L5 extratelencephalic versus intratelencephalic specificity as a function of evolutionary distance from humans (Fig. 6d, Supplementary Table 22). These genes include canonical axon-guidance genes, which may contribute to maintaining connections between spinal motor neurons that are associated with high dexterity in primates[20]. To investigate whether transcriptomically defined L5 extratelencephalic types include anatomically defined Betz cells, we combined FISH for markers of L5 extratelencephalic subtypes with immunolabelling against SMI-32, a

protein enriched in Betz cells and other long-range-projecting neurons in macaques[35] (Fig. 6e and Extended Data Fig. 11f, g). Cells consistent with the size and shape of Betz cells were identified in two L5 extratelencephalic clusters (Exc L3-5 *FEZF2 ASGR2* and Exc L5 *FEZF2 CSN1S1*), but they also included neurons with pyramidal morphologies.

Conserved and primate-enriched DEGs included ion-channel subunits (Fig. 6b and Extended Data Fig. 11e). Prior studies have established that membrane properties that depend on HCN channels (low input resistance, $R_N$, and a peak resonance, $f_R$, of around 3–9 Hz) distinguish extratelencephalic from intratelencephalic neurons in mice[36]. We found that extratelencephalic neurons expressed high levels of genes encoding proteins related to the HCN channel in all species (*HCN1* and *PEX5L*; Fig. 6b), suggesting conserved HCN-related physiological properties. To facilitate cross-species comparisons of primate extratelencephalic/Betz and mouse extratelencephalic neurons, we made patch-clamp recordings from L5 neurons in acute and cultured slice preparations of mouse (using extratelencephalic-specific *Thy1*–YFP and intratelencephalic-specific *Etv1*–EGFP lines) and macaque M1 and an area of human premotor cortex containing Betz cells (Fig. 6f, g and Extended Data Fig. 12a). For a subset of recordings, we applied patch–seq analysis to identify transcriptomic cell types (Extended Data Fig. 12b). For mouse M1, 91.4% of neurons in the *Thy1*–YFP line had extratelencephalic-like physiology, and 99.2% of neurons in the *Etv1*–EGFP line had non-extratelencephalic-like physiology (Fig. 6h,i). For primate M1, all transcriptomically defined Betz cells (humans, $n = 4$; macaques, $n = 3$) had extratelencephalic-like physiology, whereas all transcriptomically defined non-extratelencephalic neurons (humans, $n = 2$; macaques, $n = 3$) had non-extratelencephalic-like physiology (Fig. 6h, j). The presence of neurons in human premotor cortex with Betz-like morphology and gene expression is consistent with observations that Betz cells may be distributed across motor-related areas that contribute to the corticospinal tract[14].

There were substantial physiological differences between mouse and primate extratelencephalic neurons (Extended Data Fig. 12c–l). The firing rate of primate and mouse non-extratelencephalic neurons decreased to a steady state within the first second of a ten-second depolarizing current injection, whereas the firing rate of mouse extratelencephalic neurons increased moderately over the same time period (Fig. 6k, l and Extended Data Fig. 12d). In primate extratelencephalic/Betz neurons, a distinctive biphasic pattern was characterized by an early cessation of firing followed by a sustained and dramatic increase in firing later in the current injection. Thus, although the acceleration in spike frequency of extratelencephalic neurons was conserved across species, the temporal dynamics and magnitude of the acceleration were distinct in primate extratelencephalic/Betz neurons. Ion-channel-related genes that are differentially expressed between primates and mice are candidates to drive these physiological specializations.

## Discussion

Comparative analysis is a powerful strategy with which to understand brain structure and function. Conservation across species is strong evidence for functional relevance under evolutionary constraints that can help to identify essential molecular and regulatory mechanisms[37,38]. Conversely, divergence indicates adaption or drift, and may be essential to understand the mechanistic underpinnings of human brain function and susceptibility to human-specific diseases. Our integrated transcriptomic and epigenomic analysis of more than 450,000 nuclei in humans, non-human primates and mice has yielded a multimodal, hierarchical classification of approximately 100 cell types in each species, with distinct expression of marker genes and sites of accessible chromatin. This hierarchical organization is highly conserved, although species variation has limited the resolution of alignment to 45 consensus cell types. These types share a core set of molecular

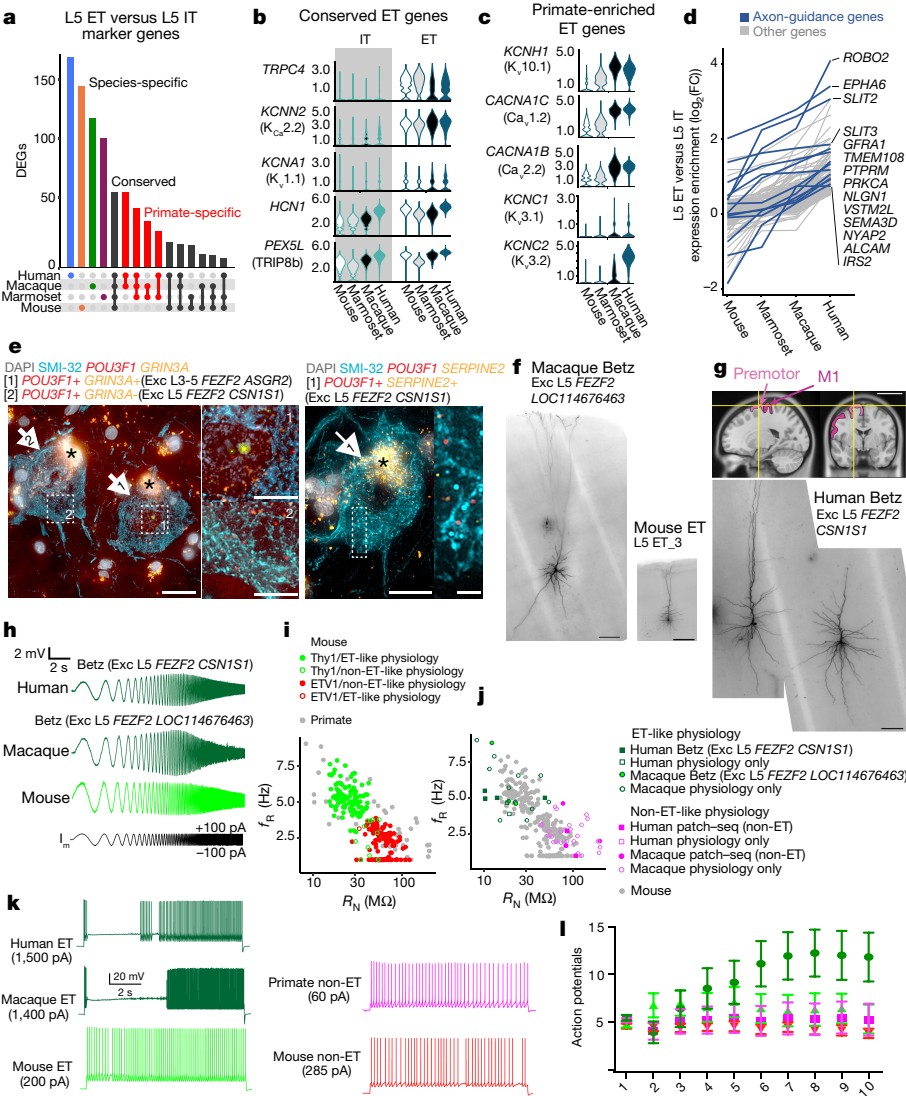

**Fig. 6 | Betz cells have specialized molecular and physiological properties.**
**a**, Upset plot showing marker genes of L5 ET compared with L5 IT across
species. **b**, **c**, Violin plots showing expression of genes related to ion channels
for genes (proteins) that are enriched in ET versus IT neurons (**b**) and in primate
versus mouse ET neurons (**c**). **d**, Genes with decreasing enrichment in L5 ET
versus IT neurons with evolutionary distance from humans. **e**, Example
photomicrographs of ISH-labelled, SMI-32-immunofluorescence-stained cells
with Betz-like morphology in human M1 L5. Cell types are identified on the
basis of marker genes. Insets show higher magnification of ISH in
corresponding cells. Asterisks mark lipofuscin; main panels, scale bars, 25 μm,
inset scale bars, 10 μm. **f**, Exemplar biocytin fills of L5 ET neurons (macaque,
*n* = 1; mice, *n* = 10; humans, *n* = 3) with transcriptomic, morphological and
electrophysiological measurements in brain slices. Scale bars, 200 μm.

**g**, Magnetic resonance images of sagittal and coronal planes, showing the
approximate location of excised premotor cortex tissue (yellow lines) and
adjacent M1. **h**, Voltage responses to a chirp stimulus for the neurons shown in
**f**, **g** (left neuron in **g**). **i**, **j**, Neurons were grouped into putative ET (humans, *n* = 6;
macaques, *n* = 14; mice, *n* = 136) versus non-ET (humans, *n* = 2; macaques, *n* = 28;
mice, *n* = 175) neurons on the basis of resonant frequency ($R_N$) and input
resistance ($f_R$). **k**, Example voltage responses to current injections (10-s step)
for ET and non-ET neurons. The amplitude was adjusted to produce roughly
five spikes during the first second. **l**, Firing rate (mean ± s.e.m.) for 1-s epochs
during the current injection. The firing rates of primate ET neurons (pooled
data from humans and macaques, *n* = 20) decreased and then increased,
whereas the firing rates of other neurons (primate IT neurons, *n* = 30; mouse ET
neurons, *n* = 8; mouse IT neurons, *n* = 12) increased or remained constant.

features, including expression of transcription factors and enrichment
of TFBSs at epigenomic sites. For example, ChCs express a conserved
transcription-factor marker, *RORA*, which has binding sites that are
enriched in regions of accessible chromatin and in hypomethylated
regions around other ChC markers.

Some characteristics of consensus types also diverge with evolution-
ary distance between species. On average, 39% of neuronal subclass
markers are shared between humans and marmosets, and 27% of mark-
ers between humans or marmosets and mice. The composition of M1
circuits shifts dramatically across species. For example, the ratio of
glutamatergic to GABAergic neurons varies from 2:1 in humans to 3:1

in marmosets and 5:1 in mice. The relative proportions of GABAergic
subclasses and types are similar across species, suggesting a global
increase in GABAergic types. As described previously[39], we observed
proportionally more L2 and L3 intratelencephalic neurons in humans,
representing a selective increase in the number of neurons projecting
to other parts of the cortex, presumably to facilitate greater cortico-
cortical communication. Humans and marmosets have proportion-
ally fewer L6 corticothalamic and L5 extratelencephalic neurons (also
observed in MTG[3]), which may reflect dilution of these cells owing to
allometric scaling of the neocortex relative to the subcortical targets
of these cells in primates. These results suggest evolutionary changes

in local and long-range cortical circuit function, and are consistent with developmental shifts in neuronal progenitor pools and changes in the timing of neurogenesis and migration.

We can leverage similarities between cell types across brain regions or species to make inferences about other cellular properties. We identified sparse L4-like cells in M1 that are not aggregated into a distinct layer and are predicted to receive input from thalamic axons. We identified two L5 extratelencephalic clusters that include neurons with Betz morphologies in humans and macaques. Similarly, in a recent study of fronto-insular cortex[10], we identified an extratelencephalic type of neuron that included cells with spindle shapes (von Economo neurons). Surprisingly, these two extratelencephalic types include neurons with non-Betz and non-spindle morphologies, suggesting that there may be graded expression differences associated with these divergent morphologies. Alternatively, distinct markers of Betz neurons may be transiently expressed during the development of long-range connectivity and not maintained in adulthood, as observed for some neurons in flies[40].

A comparative approach can help to elucidate what is different in humans or can be well modelled in closer, non-human primate relatives. In mice and primates, extratelencephalic neurons have a low input resistance and a characteristic peak resonance that reflect their large size and high expression of genes related to the HCN channel, respectively. However, primate Betz/extratelencephalic neurons have distinctive gene-expression and electrophysiological features—including pauses, bursting and spike-frequency acceleration, which have been seen in cats but not in rodents[17,18,41]. The selection of an appropriate model organism is particularly relevant when studying Betz cells and other extratelencephalic neuronal types that are selectively vulnerable in amyotrophic lateral sclerosis, some forms of frontotemporal dementia and other neurodegenerative conditions[42,43].

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

**Trygve E. Bakken**[1 ✉], **Nikolas L. Jorstad**[1], **Qiwen Hu**[2], **Blue B. Lake**[3], **Wei Tian**[4], **Brian E. Kalmbach**[1,5], **Megan Crow**[6], **Rebecca D. Hodge**[1], **Fenna M. Krienen**[7], **Staci A. Sorensen**[1], **Jeroen Eggermont**[8], **Zizhen Yao**[1], **Brian D. Aevermann**[9], **Andrew I. Aldridge**[10], **Anna Bartlett**[10], **Darren Bertagnolli**[1], **Tamara Casper**[1], **Rosa G. Castanon**[10], **Kirsten Crichton**[1], **Tanya L. Daigle**[1], **Rachel Dalley**[1], **Nick Dee**[1],

Nikolai Dembrow[5,11], Dinh Diep[3], Song-Lin Ding[1], Weixiu Dong[3], Rongxin Fang[12], Stephan Fischer[6], Melissa Goldman[7], Jeff Goldy[1], Lucas T. Graybuck[1], Brian R. Herb[13], Xiaomeng Hou[14], Jayaram Kancherla[15], Matthew Kroll[1], Kanan Lathia[1], Baldur van Lew[8], Yang Eric Li[14,16], Christine S. Liu[17,18], Hanqing Liu[10], Jacinta D. Lucero[4], Anup Mahurkar[13], Delissa McMillen[1], Jeremy A. Miller[1], Marmar Moussa[19], Joseph R. Nery[10], Philip R. Nicovich[1], Sheng-Yong Niu[10,20], Joshua Orvis[13], Julia K. Osteen[4], Scott Owen[1], Carter R. Palmer[17,18], Thanh Pham[1], Nongluk Plongthongkum[3], Olivier Poirion[14], Nora M. Reed[7], Christine Rimorin[1], Angeline Rivkin[4], William J. Romanow[17], Adriana E. Sedeño-Cortés[1], Kimberly Siletti[21], Saroja Somasundaram[1], Josef Sulc[1], Michael Tieu[1], Amy Torkelson[1], Herman Tung[1], Xinxin Wang[22], Fangming Xie[23], Anna Marie Yanny[1], Renee Zhang[9], Seth A. Ament[13], M. Margarita Behrens[4], Hector Corrada Bravo[15], Jerold Chun[17], Alexander Dobin[24], Jesse Gillis[6], Ronna Hertzano[25], Patrick R. Hof[26], Thomas Höllt[27], Gregory D. Horwitz[28], C. Dirk Keene[29], Peter V. Kharchenko[2], Andrew L. Ko[30,31], Boudewijn P. Lelieveldt[8,32], Chongyuan Luo[33], Eran A. Mukamel[34], António Pinto-Duarte[4], Sebastian Preissl[14], Aviv Regev[35], Bing Ren[14,16], Richard H. Scheuermann[9,36,37], Kimberly Smith[1], William J. Spain[5,11], Owen R. White[13], Christof Koch[1], Michael Hawrylycz[1], Bosiljka Tasic[1], Evan Z. Macosko[35], Steven A. McCarroll[7,35], Jonathan T. Ting[1,5], Hongkui Zeng[1], Kun Zhang[3], Guoping Feng[38,39,40], Joseph R. Ecker[10,41], Sten Linnarsson[21] & Ed S. Lein[1✉]

[1]Allen Institute for Brain Science, Seattle, WA, USA. [2]Department of Biomedical Informatics, Harvard Medical School, Boston, MA, USA. [3]Department of Bioengineering, University of California, San Diego, La Jolla, CA, USA. [4]The Salk Institute for Biological Studies, La Jolla, CA, USA. [5]Department of Physiology and Biophysics, University of Washington, Seattle, WA, USA. [6]Stanley Institute for Cognitive Genomics, Cold Spring Harbor Laboratory, Cold Spring Harbor, NY, USA. [7]Department of Genetics, Harvard Medical School, Boston, MA, USA. [8]LKEB, Department of Radiology, Leiden University Medical Center, Leiden, The Netherlands. [9]J. Craig Venter Institute, La Jolla, CA, USA. [10]Genomic Analysis Laboratory, The Salk Institute for Biological Studies, La Jolla, CA, USA. [11]Epilepsy Center of Excellence, Department of Veterans Affairs Medical Center, Seattle, WA, USA. [12]Bioinformatics and Systems Biology Graduate Program, University of California, San Diego, La Jolla, CA, USA. [13]Institute for Genomes Sciences, University of Maryland School of Medicine, Baltimore, MD, USA. [14]Center for Epigenomics, Department of Cellular and Molecular Medicine, University of California, San Diego, La Jolla, CA, USA. [15]Department of Computer Science, University of Maryland College Park, College Park, MD, USA. [16]Ludwig Institute for Cancer Research, La Jolla, CA, USA. [17]Sanford Burnham Prebys Medical Discovery Institute, La Jolla, CA, USA. [18]Biomedical Sciences Program, School of Medicine, University of California, San Diego, La Jolla, CA, USA. [19]University of Connecticut, Storrs, CT, USA. [20]Computer Science and Engineering Program, University of California, San Diego, La Jolla, CA, USA. [21]Department of Medical Biochemistry and Biophysics, Karolinska Institutet, Stockholm, Sweden. [22]McDonnell Genome Institute, Washington University School of Medicine, St Louis, MO, USA. [23]Department of Physics, University of California, San Diego, La Jolla, CA, USA. [24]Cold Spring Harbor Laboratory, Cold Spring Harbor, NY, USA. [25]Departments of Otorhinolaryngology, Anatomy and Neurobiology, University of Maryland School of Medicine, Baltimore, MD, USA. [26]Nash Family Department of Neuroscience and Friedman Brain Institute, Icahn School of Medicine at Mount Sinai, New York, NY, USA. [27]Computer Graphics and Visualization Group, Delt University of Technology, Delft, The Netherlands. [28]Department of Physiology and Biophysics, Washington National Primate Research Center, University of Washington, Seattle, WA, USA. [29]Department of Laboratory Medicine and Pathology, University of Washington, Seattle, WA, USA. [30]Department of Neurological Surgery, University of Washington School of Medicine, Seattle, WA, USA. [31]Regional Epilepsy Center, Harborview Medical Center, Seattle, WA, USA. [32]Pattern Recognition and Bioinformatics group, Delft University of Technology, Delft, The Netherlands. [33]Department of Human Genetics, University of California, Los Angeles, Los Angeles, CA, USA. [34]Department of Cognitive Science, University of California, San Diego, La Jolla, CA, USA. [35]Broad Institute of MIT and Harvard, Cambridge, MA, USA. [36]Department of Pathology, University of California, San Diego, CA, USA. [37]Division of Vaccine Discovery, La Jolla Institute for Immunology, La Jolla, CA, USA. [38]McGovern Institute for Brain Research, MIT, Cambridge, MA, USA. [39]Department of Brain and Cognitive Sciences, MIT, Cambridge, MA, USA. [40]Stanley Center for Psychiatric Research, Broad Institute of MIT and Harvard, Cambridge, MA, USA. [41]Howard Hughes Medical Institute, The Salk Institute for Biological Studies, La Jolla, CA, USA. ✉e-mail: trygveb@alleninstitute.org; edl@alleninstitute.org

## Methods

### Statistics and reproducibility

For multiplex fluorescent in situ hybridization (FISH) and immuno-fluorescence staining experiments, each ISH probe combination was repeated with similar results on at least two separate individuals per species, and on at least two sections per individual. The experiments were not randomized and the investigators were not blinded to allocation during experiments and outcome assessment. No statistical methods were used to predetermine sample size.

### Ethical compliance

Postmortem adult human brain tissue was collected after obtaining permission from the decedent's next-of-kin. Postmortem tissue collection was performed in accordance with the provisions of the United States Uniform Anatomical Gift Act of 2006 described in the California Health and Safety Code section 7150 (effective 1 January 2008) and other applicable state and federal laws and regulations. The Western Institutional Review Board reviewed tissue-collection processes and determined that they did not constitute research on human participants that requires assessment by an institutional review board (IRB).

Tissue procurement from a neurosurgical donor was performed outside of the supervision of the Allen Institute at a local hospital, and tissue was provided to the Allen Institute under the authority of the IRB of the participating hospital. A hospital-appointed case coordinator obtained informed consent from the donor before surgery. Tissue specimens were de-identified before receipt by Allen Institute personnel. The specimens collected for this study were apparently non-pathological tissues removed during the normal course of surgery to access underlying pathological tissues. Tissue specimens collected were determined to be non-essential for diagnostic purposes by medical staff, and would have otherwise been discarded.

Mouse experiments were conducted in accordance with the US National Institutes of Health (NIH) Guide for the Care and Use of Laboratory Animals under protocol numbers 0120-09-16, 1115-111-18 or 18-00006, and were approved by the Institutional Animal Care and Use Committee (IACUC) at the University of Washington, the Allen Institute for Brain Science, the Salk Institute, or the Massachusetts Institute of Technology. Marmoset experiments were approved by and in accordance with the Massachusetts Institute of Technology IACUC, protocol number 051705020. Macaque tissue used in this research was obtained from the University of Washington National Primate Resource Center, under a protocol approved by the University of Washington IACUC.

### Postmortem human tissue specimens

Male and female donors 18–68 years of age with no known history of neuropsychiatric or neurological conditions ('control' cases) were considered for inclusion in this study (Extended Data Table 1). Routine serological screening for infectious disease (HIV, hepatitis B and hepatitis C) was conducted using donor blood samples, and only those donors who were negative for all three tests were considered for inclusion. Only those specimens with RNA integrity (RIN) values of 7.0 or more were considered for inclusion. Postmortem brain specimens were processed as described[3]. Briefly, coronal brain slabs were cut at 1 cm intervals and frozen for storage at −80 °C until further use. Putative hand and trunk-lower limb regions of the primary motor cortex were identified, removed from slabs of interest, and subdivided into smaller blocks. One block from each donor was processed for cryosectioning and fluorescent Nissl staining (Neurotrace 500/525, ThermoFisher Scientific). Stained sections were screened for histological hallmarks of primary motor cortex. After verifying that regions of interest contained M1, blocks were processed for nucleus isolation as described below.

### Human RNA-seq, quality control and clustering

**SMART-seq v4.** *Nucleus isolation and sorting*. Vibratome sections were stained with fluorescent Nissl, allowing microdissection of individual cortical layers (https://doi.org/10.17504/protocols.io.7aehibe). Nucleus isolation was performed as described (https://doi.org/10.17504/protocols.io.ztqf6mw). NeuN staining was carried out using mouse anti-NeuN antibody conjugated to phycoerythrin (PE; EMD Millipore, catalogue number FCMAB317PE) at a dilution of 1:500. Control samples were incubated with mouse IgG1k-PE isotype control (BD Biosciences, 555749; 1:250 dilution). DAPI (4′,6-diamidino-2-phenylindole dihydrochloride; ThermoFisher Scientific, D1306) was applied to nucleus samples at a concentration of 0.1 µg ml⁻¹. Single-nucleus sorting was carried out on either a BD FACSAria II SORP or a BD FACSAria Fusion instrument (BD Biosciences) using a 130 µm nozzle and BD Diva software v8.0. A standard gating strategy based on DAPI and NeuN staining was applied to all samples as described[3]. Doublet discrimination gates were used to exclude nucleus aggregates.

*RNA sequencing*. The SMART-Seq v4 ultra low input RNA kit for sequencing (Takara, catalogue number 634894) was used as per the manufacturer's instructions. Standard controls were processed with each batch of experimental samples as described (https://www.protocols.io/view/smarterv4-0-5x-amplification-for-single-cell-or-si-7d5hi86). After reverse transcription, complementary DNA was amplified with 21 polymerase chain reaction (PCR) cycles. The NexteraXT DNA library preparation kit (Illumina, FC-131-1096) with NexteraXT index kit V2 sets A–D (FC-131-2001, 2002, 2003 or 2004) was used for preparation of sequencing libraries. Libraries were sequenced on an Illumina HiSeq 2500 instrument (Illumina HiSeq 2500 System, Research Resource Identifier (RRID) SCR_016383) using Illumina high output V4 chemistry. The following instrumentation software was used during the data-generation workflow: SoftMax Pro v6.5, VWorks v11.3.0.1195 and v13.1.0.1366, Hamilton Run Time Control v4.4.0.7740, Fragment Analyzer v1.2.0.11, and Mantis Control Software v3.9.7.19.

*Quantification of gene expression*. Raw read (fastq) files were aligned to the GRCh38 human genome sequence (Genome Reference Consortium, 2011) with the RefSeq transcriptome version GRCh38.p2 (RefSeq, RRID SCR_003496, current as of 13 April 2015) and updated by removing duplicate Entrez gene entries from the gtf reference file for STAR processing. For alignment, Illumina sequencing adapters were clipped from the reads using the fastqMCF program (from ea-utils). After clipping, the paired-end reads were mapped using spliced transcripts alignment to a reference (STAR v2.7.3a, RRID SCR_015899) with default settings. Reads that did not map to the genome were then aligned to synthetic construct (that is, External RNA Controls Consortium, ERCC) sequences and the *Escherichia coli* genome (version ASM584v2). Quantification was performed using summerizeOverlaps from the R package GenomicAlignments v1.18.0. Expression levels were calculated as counts per million (CPM) of exonic plus intronic reads.

**10× Chromium RNA sequencing.** *Nucleus isolation and sorting*. Nucleus isolation for 10× Chromium RNA sequencing was conducted as described (https://doi.org/10.17504/protocols.io.y6rfzd6). After sorting, single-nucleus suspensions were frozen in a solution of 1× phosphate-buffered saline (PBS), 1% bovine serum albumin (BSA), 10% dimethylsulfoxide (DMSO) and 0.5% RNAsin Plus RNase inhibitor (Promega, N2611), and stored at −80 °C. At the time of use, frozen nuclei were thawed at 37 °C and processed for loading on the 10× Chromium instrument as described (https://doi.org/10.17504/protocols.io.nx3dfqn). Samples were processed using the 10× Chromium single-cell 3′ reagent kit v3. 10× chip loading and sample processing were carried out according to the manufacturer's protocol. Gene expression was quantified using the default 10× Cell Ranger v3 (Cell Ranger, RRID SCR_017344) pipeline, except for substituting of the curated genome annotation used for SMART-seq v4 quantification.

Introns were annotated as 'mRNA', and intronic reads were included to quantify expression.

**Quality control of RNA-seq data.** Nuclei were included for analysis if they passed all quality-control criteria. For SMART-seq v4, criteria were: more than 30% of cDNA was longer than 400 base pairs; more than 500,000 reads were aligned to exonic or intronic sequences; more than 40% of total reads were aligned; more than 50% of reads were unique; the T/A nucleotide ratio was greater than 0.7. For Cv3, criteria were: more than 500 (non-neuronal nuclei) or more than 1,000 (neuronal nuclei) genes were detected; doublet score was less than 0.3.

**Clustering of RNA-seq data.** Nuclei passing quality-control criteria were grouped into transcriptomic cell types using a reported iterative clustering procedure[2,3]. Briefly, intronic and exonic read counts were summed, and $\log_2$-transformed expression was centred and scaled across nuclei. X and Y chromosomes and mitochondrial genes were excluded to avoid nucleus clustering on the basis of sex or nucleus quality. DEGs were selected; principal components analysis (PCA) reduced dimensionality; and a nearest neighbour graph was built using up to 20 principal components. Clusters were identified with Louvain community detection (or Ward's hierarchical clustering if there were fewer than 3,000 nuclei), and pairs of clusters were merged if either cluster lacked marker genes. Clustering was applied iteratively to each subcluster until clusters could not be further split.

Cluster robustness was assessed by repeating iterative clustering 100 times for random subsets of 80% of nuclei. A co-clustering matrix was generated that represented the proportion of clustering iterations in which each pair of nuclei was assigned to the same cluster. We defined consensus clusters by iteratively splitting the co-clustering matrix as described[2,3]. The clustering pipeline is implemented in the R package scratch.hicat v0.0.22 (RRID SCR_018099), with marker genes defined using the limma v3.38.3 package; the clustering method is provided by the 'run_consensus_clust' function (https://github.com/AllenInstitute/scratch.hicat).

Clusters were curated on the basis of quality-control criteria or the expression of markers of cell classes (*GAD1*, *SLC17A7*, *SNAP25*). Clusters were identified as donor specific if they included fewer nuclei sampled from donors than expected by chance. To confirm exclusion, clusters automatically flagged as outliers or donor specific were manually inspected for expression of broad cell-class marker genes, mitochondrial genes related to quality, and known activity-dependent genes.

## Marmoset sample processing and nuclei isolation
Marmoset experiments were approved by, and in accordance with, the Massachusetts Institute of Technology IACUC, protocol number 051705020. Two adult marmosets (2.3 and 3.1 years old; one male, one female; Extended Data Table 2) were deeply sedated by intramuscular injection of ketamine (20–40 mg kg$^{-1}$) or alfaxalone (5–10 mg kg$^{-1}$), followed by intravenous injection of sodium pentobarbital (10–30 mg kg$^{-1}$). When the pedal withdrawal reflex was eliminated and/or the respiratory rate was diminished, animals were transcardially perfused with ice-cold sucrose–HEPES buffer. Whole brains were rapidly extracted into fresh buffer on ice. Sixteen 2-mm coronal blocking cuts were rapidly made using a custom-designed marmoset brain matrix. Coronal slabs were snap-frozen in liquid nitrogen and stored at –80 °C until use.

As for human samples, marmoset M1 was isolated from thawed slabs using fluorescent Nissl staining (Neurotrace 500/525, ThermoFisher Scientific). Stained sections were screened for histological hallmarks of primary motor cortex. Nuclei were isolated from the dissected regions as described (https://www.protocols.io/view/extraction-of-nuclei-from-brain-tissue-2srged6), and were processed using the 10× Chromium single-cell 3′ reagent kit v3. 10× chip loading and sample processing was done according to the manufacturer's protocol.

## Marmoset RNA-seq, quality control and clustering
**RNA-sequencing.** Libraries were sequenced on NovaSeq S2 instruments (Illumina). Raw sequencing reads were aligned to calJac3. Mitochondrial sequence was added into the published reference assembly. Human sequences of *RNR1* and *RNR2* (mitochondrial) and *RNA5S* (ribosomal) were aligned using gmap to the marmoset genome and added to the calJac3 annotation. Reads that mapped to exons or introns of each assembly were assigned to annotated genes. Libraries were sequenced to a median read depth of 5.95 reads per unique molecular index (UMI). The alignment pipeline can be found at https://github.com/broadinstitute/Drop-seq.

**Cell filtering.** Cell barcodes were filtered to distinguish true nuclei barcodes from empty beads and PCR artefacts by assessing proportions of ribosomal and mitochondrial reads, ratio of intronic/exonic reads (greater than 50% of intronic reads), library size (more than 1,000 UMIs) and sequencing efficiency (true cell barcodes have higher reads per UMI). The resulting digital gene-expression matrix (DGE) from each library was carried forward for clustering.

**Clustering.** Clustering analysis proceeded as in ref. [9]. Briefly, independent component analysis (ICA, using the fastICA v1.2-1 package in R; RRID SCR_013110) was performed jointly on all marmoset DGEs after normalization and variable gene selection, as in ref. [44]. The first-round clustering resulted in 15 clusters, corresponding to major cell classes (neurons, glia and endothelial cells). Each cluster was curated as in ref. [44] to remove doublets and outliers. Independent components were partitioned to remove those reflecting artefactual signals (for example, those for which cell loading indicated replicate or batch effects). The remaining independent components were used to determine clustering (Louvain community detection algorithm igraph v1.2.6 package in R); for each cluster, nearest neighbour and resolution parameters were set to optimize 1:1 mapping between each independent component and a cluster.

## Mouse snRNA-seq and snATAC-seq
Single nuclei were isolated from mouse primary motor cortex; gene expression and accessible chromatin were quantified using RNA-seq (Cv3 and SSv4) and snATAC-seq; and transcriptomic cell types, dendrograms and accessible-chromatin profiles were defined as described[5].

## Integrating and clustering human Cv3 and SSv4 snRNA-seq datasets
To establish a set of human consensus cell types, we performed a separate integration of snRNA-seq technologies on the major cell classes (glutamatergic, GABAergic, and non-neuronal). Broadly, this integration is comprised of 6 steps: (1) subsetting the major cell class from each technology (for example, Cv3 GABAergic and SSv4 GABAergic); (2) finding marker genes for all clusters within each technology; (3) integrating both datasets with Seurat's standard workflow using marker genes to guide integration (Seurat v3.1.1)[45]; (4) overclustering the data to a greater number of clusters than were originally identified within a given individual dataset; (5) finding marker genes for all integrated clusters; and (6) merging similar integrated clusters together based on marker genes until all merging criteria were sufficed, resulting in the final human consensus taxonomy.

More specifically, each expression matrix was $\log_2(CPM + 1)$ transformed then placed into a Seurat object with accompanying metadata. Variable genes were determined by downsampling each expression matrix to a maximum of 300 nuclei per scratch.hicat-defined cluster (from a previous step; see scratch.hicat clustering) and running select_markers (scratch.io v0.1.0) with n set to 20, to generate a list of up to 20 marker genes per cluster. The union of the Cv3 and SSv4 gene lists were then used as input for anchor finding, dimensionality

reduction, and Louvain clustering of the full expression matrices. We used 100 dimensions for steps in the workflow, and 100 random starts during clustering. Louvain clustering was performed to overcluster the dataset to identify more integrated clusters than the number of scrattch.hicat-defined clusters. For example, GABAergic neurons had 79 and 37 scrattch.hicat-defined clusters, 225 overclustered integrated clusters, and 72 final human consensus clusters after merging for Cv3 and SSv4 datasets, respectively. To merge the overclustered integrated clusters, up to 20 marker genes were found for each cluster to establish the neighbourhoods of the integrated dataset. Clusters were then merged with their nearest neighbour if there were not a minimum of ten Cv3 and two SSv4 nuclei in a cluster, and a minimum of 4 DEGs that distinguished the query cluster from the nearest neighbour (note: these were the same parameters used to perform the initial scrattch.hicat clustering of each dataset).

### Integrating and clustering

**Human MTG and M1 SSv4 snRNA-seq datasets.** To compare cell types between our M1 human cell type taxonomy and our previously described human MTG taxonomy[3], we used Seurat's standard integration workflow to perform a supervised integration of the M1 and MTG SSv4 datasets. Intronic and exonic reads were summed into a single expression matrix for each dataset, with CPM normalized, and placed into a Seurat object with accompanying metadata. All nuclei from each major cell class were integrated and clustered separately. Up to 100 marker genes for each cluster within each dataset were identified, and the union of these two gene lists was used as input to guide alignment of the two datasets during integration, dimensionality reduction and clustering steps. We used 100 dimensions for all steps in the workflow. To compare laminar positioning in M1 and MTG, we estimated the relative cortical depth from pia for each neuron on the basis of layer dissection and average layer thickness[46].

**Integrating Cv3 snRNA-seq datasets across species.** To identify homologous cell types across species, we used Seurat's SCTransform workflow to perform a separate supervised integration on each cell class across species. Raw expression matrices were reduced to include only those genes with one-to-one orthologues defined in the three species (14,870 genes; downloaded from NCBI Homologene (https://www.ncbi.nlm.nih.gov/homologene) in November 2019; RRID SCR_002924) and placed into Seurat objects with accompanying metadata. To avoid having one species dominate the integrated space and to account for potential differences in each species' clustering resolution, we downsampled the number of nuclei to have similar numbers across species at the subclass level (for example, *Lamp5*, *Sst*, L2/3 IT, L6b, and so on). The species with the largest number of clusters under a given subclass was allowed a maximum of 200 nuclei per cluster. The remaining species then split this theoretical maximum (200 nuclei multiplied by the maximum number of clusters under the subclass) evenly across their clusters. For example, the L2/3 intratelencephalic subclass had eight, four and three clusters for humans, marmosets and mice, respectively. All species were allowed a maximum of 1,600 L2/3 intratelencephalic nuclei in total; or a maximum of 200 human, 400 marmoset and 533 mouse nuclei per cluster. To integrate across species, all Seurat objects were merged and normalized using Seurat's SCTransform function. To better guide the alignment of cell types from each species, we found up to 100 marker genes for each cluster within a given species. We used the union of these gene lists as input for the integration and dimensionality reduction steps, with 30 dimensions used for integration and 100 for dimensionality reduction and clustering. Clustering the human–marmoset–mouse integrated space provided an additional quality-control mechanism, revealing numerous small, species-specific integrated clusters that contained only low-quality nuclei (low UMIs and genes detected). We excluded 4,836 nuclei from the marmoset dataset that constituted low-quality integrated neuronal clusters.

### Estimation of cell-type homology

To identify homologous groups from different species, we applied a tree-based method (https://github.com/AllenInstitute/BICCN_M1_Evo) and package (https://github.com/huqiwen0313/speciesTree). In brief, the approach consists of four steps: first, metacell clustering; second, hierarchical reconstruction of a metacell tree; third, measurements of species mixing and stability of splits; and fourth, dynamic pruning of the hierarchical tree.

First, to reduce noise in single-cell datasets and to remove species-specific batch effects, we clustered cells into small highly similar groups on the basis of the integrated matrix generated by Seurat, as described in the previous section. These cells were further aggregated into metacells, and the expression values of the metacells were calculated by averaging the gene expression of individual cells that belong to each metacell. Correlation was calculated on the basis of the metacell gene-expression matrix to infer the similarity of each metacell cluster. Then hierarchical clustering was performed on the basis of the metacell gene-expression matrix using Ward's method. For each node or corresponding branch in the hierarchical tree, we calculated three measurements, and the hierarchical tree was visualized on the basis of these measurements: first, cluster size was visualized as the thickness of branches in the tree; second, species mixing was calculated on the basis of the entropy of the normalized cell distribution and visualized as the colour of each node and branch; third, the stability of each node. The entropy of cells was calculated as: $H = -\sum_i p_i \log p_i$, where $p_i$ is the probability of cells from one species appearing among all the cells in a node. We assessed the node stability by evaluating the agreement between the original hierarchical tree, and a result on a subsampled dataset was calculated on the basis of the optimal subtree in the subsampled hierarchical trees, derived from subsampling 95% of cells in the original dataset. The entire subsampling process was repeated 100 times and the mean stability score for every node in the original tree was calculated. Finally, we recursively searched each node in the tree. If the heuristic criteria (see below) were not met for any node below the upper node, the entire subtree below the upper node was pruned, and all of the cells belonging to this subtree were merged into one homologous group.

To identify robust homologous groups, we applied criteria in two steps to dynamically search the cross-species tree. First, for each node in the tree, we computed the mixing of cells from three species on the basis of the entropy and set it as a tuning parameter. For each integrated tree, we tuned the entropy parameter to make sure that the tree method generated the highest resolution of homologous clusters without losing the ability to identify potential species-specific clusters. Nodes with entropy larger than 2.9 (for inhibitory neurons) or 2.75 (for excitatory neurons) were considered as well mixed nodes. For example, an entropy of 2.9 corresponded to a mixture of humans, marmosets and mice equal to (0.43, 0.37, 0.2) or (0.38, 0.30, 0.32). We recursively searched all of the nodes in the tree until we found the node nearest the leaves of the tree that was well mixed among species, and this node was defined as a well mixed upper node. Second, we further checked the within-species cell composition for the subtrees below the well mixed upper node to determine whether further splits were needed. For the cells on the subtrees below the well mixed upper node, we measured the purity of within-species cell composition by calculating the percentage of cells that fall into a specific subgroup in each individual species. If the purity for any species was larger than 0.8, we went one step further below the well mixed upper node so that its children were selected. Any branches below these nodes (or well mixed upper node if the within-species cell composition criteria were not met) were pruned, and cells from these nodes were merged into the same homologous groups, and the final integrated tree was generated.

As a final curation step, the homologous groups generated by the tree method were merged to be consistent with within-species clusters. We

defined consensus types by comparing the overlap of within-species clusters between humans and marmosets, and humans and mice, as described[3]. For each pair of human and mouse clusters and human and marmoset clusters, the overlap was defined as the sum of the minimum proportion of nuclei in each cluster that overlapped within each leaf of the pruned tree. This approach identified pairs of clusters that consistently co-clustered within one or more leaves. Cluster overlaps varied from 0 to 1, and were visualized as a heat map with human M1 clusters in rows and mouse or marmoset M1 clusters in columns. Cell-type homologies were identified as one-to-one, one-to-many, or many-to-many so that they were consistent in all three species. For example, the Vip_2 consensus type could be resolved into multiple homologous types between humans and marmosets but not humans and mice, and the coarser homology was retained. Consensus type names were assigned on the basis of the annotations of member clusters from humans and mice, and avoided specific marker gene names owing to the variability of marker expression across species.

To quantify cell-type alignment between pairs of species, we pruned the hierarchical tree described above on the basis of the stability and mixing of two species. We performed this analysis for human–marmoset, human–mouse and marmoset–mouse, and compared the alignment resolution of each subclass. The pruning criteria were tuned to fit the two-species comparison and to remove bias, and we set the same criteria for all comparisons (entropy cutoff 3.0). Specifically, for each subclass and pairwise species comparison, we calculated the number of leaves in the pruned tree. We repeated this analysis on the 100 subsampled datasets, and calculated the mean and standard deviation of the number of leaves in the pruned trees. For each subclass, we tested for significant differences in the average number of leaves across pairs of species using an ANOVA test followed by post hoc Tukey HSD tests.

## Marker determination for cell-type clusters

NS-Forest v2.1 (RRID SCR_018348) was used to determine the minimum set of marker genes whose combined expression identified cells of a given type with maximum classification accuracy[47,48] (https://github.com/JCVenterInstitute/NSForest/releases). Briefly, for each cluster, NS-Forest produces a random-forest model using a one versus all binary classification approach. The top ranked genes from the random forest are then filtered by expression level to retain genes that are expressed in at least 50% of the cells within the target cluster. The selected genes are then reranked by binary score, calculated by first finding median cluster expression values for a given gene and then dividing by the target median cluster expression value. Next, one minus this scaled value is calculated, resulting in 0 for the target cluster and 1 for clusters that have no expression, while negative scaled values are set to 0. These values are then summed and normalized by dividing by the total number of clusters. In the ideal case, where all off-target clusters have no expression, the binary score is 1. Finally, for the top six binary genes, optimal expression level cutoffs are determined and all permutations of genes are evaluated by f-beta score, where the beta is weighted to favour precision. This f-beta score indicates the power of discrimination for a cluster and a given set of marker genes. The gene combination giving the highest f-beta score is selected as the optimal marker gene combination. Marker gene sets for human, mouse and marmoset primary motor cortex are listed in Supplementary Tables 4–6, and were used to construct the semantic cell-type definitions provided in Supplementary Table 1.

## Calculating DEGs

To identify subclass level DEGs that are conserved and divergent across species, we used the integrated Seurat objects from the species integration step. Seurat objects for each major cell class were downsampled to have up to 200 cells per species cell type. Positive DEGs were then found using Seurat's FindAllMarkers function, using the ROC test with default parameters (min.pct = 0.1, AUROC threshold = 0.7). We compared each subclass within species to all remaining nuclei in that class, and used the SCT normalized counts to test for differential expression. For example, human *Sst* nuclei were compared with all other GABAergic human neurons using the ROC test. Venn diagrams were generated using the eulerr package v6.0.0 to visualize the relationships of DEGs across species for a given subclass. Heat maps of DEGs for all subclasses under a given class were generated by downsampling each subclass to 50 random nuclei per species. SCT normalized counts were then scaled and visualized with Seurat's DoHeatmap function.

To identify ChC DEGs that are enriched over basket cells, we used the integrated Seurat objects from the species integration step. The *Pvalb* subclass was subset, and species cell types were then designated as either ChCs or basket cells. Positive DEGs were then found using Seurat's FindAllMarkers function using the ROC test to compare ChCs and basket cells for each species. Venn diagrams were generated using the eulerr package to visualize the relationship of ChC-enriched DEGs across species. Heat maps of conserved DEGs were generated by downsampling the dataset to have 100 randomly selected basket cells and ChCs from each species. SCT normalized counts were then scaled and visualized with Seurat's DoHeatmap function.

We used the four-species (humans, macaques, marmosets and mice) integrated glutamatergic Seurat object from the species integration step for all L5 extratelencephalic DEG figures. L5 extratelencephalic and L5 intratelencephalic subclasses were downsampled to 200 randomly selected nuclei per species. A ROC test was then performed using Seurat's FindAllMarkers function between the two subclasses for each species to identify L5 extratelencephalic-specific marker genes. We then used the UpSetR v1.4.0 package to visualize the intersections of the marker genes across all four species as an upset plot. To determine genes that decrease in expression across evolutionary distance in L5 extratelencephalic neurons, we found the log-transformed fold change between L5 extratelencephalic and L5 intratelencephalic for each species across all genes. We then filtered the gene lists to include only those genes that had a trend of decreasing log-transformed fold change (from humans to macaques to marmosets to mice). Lastly, we excluded any gene that did not have a log-transformed fold change of 0.5 or greater in the human comparison. These 131 genes were then used as input for Gene Ontology (GO) analysis with the PANTHER classification system[49] for the biological process category, with the organism set to *Homo sapiens*. All significant GO terms for this gene list were associated with cell–cell adhesion and axon guidance, and are coloured blue in the line graph of their expression enrichment.

## Differential isoform usage in humans and mice

To assess changes of isoform usage between mice and humans, we used SSv4 data with full transcript coverage and estimated isoform abundance in each cell subclasses. To mitigate low read depth in each cell, we aggregated reads from all cells in each subclass. We estimated the relative isoform usage in each subclass by calculating its genic proportion ($P$), defined as the ratio ($R$) of isoform expression to the gene expression, where $R = (P_{human} - P_{mouse})/(P_{human} + P_{mouse})$. For a common set of transcripts for mice and humans, we used the University of California San Diego (UCSC) browser (RRID SCR_005780) TransMapV5 set of human transcripts (hg38 assembly, Gencode v31 annotations, RRID SCR_014966) mapped to the mouse genome (mm10 assembly) (http://hgdownload.soe.ucsc.edu/gbdb/mm10/transMap/V5/mm10.ensembl.transMapV5.bigPsl). We considered only medium to highly expressed isoforms, which have abundances of greater than 10 transcripts per million (TPM) and $P$ values of greater than 0.2 in either mice or humans, and abundances of greater than 10 TPM in both mice and humans.

To calculate isoform abundance in each cell subclass, we aggregated reads from each subclass; mapped reads to the mouse or human reference genome with STAR 2.7.3a using default parameters; transformed genomic coordinates into transcriptomic coordinates using the STAR parameter –quantMode TranscriptomeSAM; and quantified

isoform and gene expression using the RSEM v1.3.3 parameters (RSEM, RRID:SCR_013027) –bam–seed 12345–paired-end–forward-prob 0.5–single-cell-prior–calc-ci.

To estimate statistical significance, we calculated the standard deviation of isoform genic proportion ($P_{human}$ and $P_{mouse}$) from the RSEM's 95% confidence intervals of isoform expression; calculated the $P$-value using the normal distribution for the ($P_{human} - P_{mouse}$) and the summed (mouse + human) variance; and Bonferroni-adjusted $P$-values by multiplying nominal $P$-values by the number of medium to highly expressed isoforms in each subclass.

## Species cluster dendrograms

DEGs for a given species were identified by using Seurat's FindAllMarkers function with a Wilcox test and comparing each cluster with every other cluster under the same subclass, with logfc.threshold set to 0.7 and min.pct set to 0.5. The union of up to 100 genes per cluster with the highest avg_logFC was used. The average $\log_2$ expression of the DEGs was then used as input for the build_dend function from scrattch.hicat to create the dendrograms. This was carried out with both human and marmoset datasets. For mouse dendrogram methods, see ref. [5].

## Multiplex FISH and immunofluorescence

Fresh-frozen human postmortem brain tissues were sectioned at 14–16 μm onto Superfrost Plus glass slides (Fisher Scientific). Sections were dried for 20 min at −20 °C and then vacuum sealed and stored at −80 °C until use. The RNAscope multiplex fluorescent v1 kit was used per the manufacturer's instructions for fresh-frozen tissue sections (ACD Bio), except that fixation was performed for 60 min in 4% paraformaldehyde in 1× PBS at 4 °C and protease treatment was shortened to 5 min. For combined RNAscope and immunofluorescence, primary antibodies were applied to tissues after completion of FISH staining. Primary antibodies used were mouse anti-GFAP (EMD Millipore, catalogue number MAB360, RRID AB_11212597, 1:500 dilution) and mouse anti-Neurofilament H (SMI-32, BioLegend, catalogue number 801701, RRID AB_2564642, 1:250 dilution). Secondary antibodies were goat anti-mouse IgG (H+L) Alexa Fluor 568 conjugate (ThermoFisher Scientific, catalogue number A-11004, 1:500 dilution), goat anti-mouse IgG (H+L) Alexa Fluor 594 conjugate (ThermoFisher Scientific, catalogue number A-11005, 1:500 dilution) and goat anti-mouse IgG (H+L) Alexa Fluor 647 conjugate (ThermoFisher Scientific, catalogue number A-21235, 1:500 dilution) conjugates (594, 647). Sections were imaged using a 60× oil immersion lens on a Nikon TiE fluorescence microscope equipped with NIS-Elements Advanced Research imaging software (v4.20, RRID SCR_014329). For all RNAscope FISH experiments, positive cells were called by manually counting RNA spots for each gene. Cells were called positive for a gene if they contained three or more RNA spots for that gene. Lipofuscin autofluorescence was distinguished from RNA spot signals on the basis of the larger size of lipofuscin granules and broad fluorescence spectrum of lipofuscin. Staining for each probe combination was repeated with similar results on at least two separate individuals per species and on at least two sections per individual. Experiments examining L5 extratelencephalic neurons in humans were conducted on tissues taken from the dome of the gyrus corresponding to the presumptive trunk-lower limb portion of M1. Images were assessed with FIJI distribution of ImageJ v1.52p and GraphPad Prism v7.04.

## Conservation of gene families

To investigate the conservation and divergence of the coexpression of gene families between primates and mice, we carried out MetaNeighbour analysis[50] using gene groups curated by the HUGO Gene Nomenclature Committee (HGNC, RRID SCR_002827) at the European Bioinformatics Institute (https://www.genenames.org; downloaded January 2020) and by the Synaptic Gene Ontology (SynGO, RRID SCR_017330)[51] (downloaded February 2020). HGNC annotations were propagated via the provided group hierarchy to ensure the comprehensiveness of parent annotations. Only groups containing five or more genes were included in the analysis.

After splitting data by class, we used MetaNeighbour to compare data at the cluster level using labels from cross-species integration with Seurat. Cross-species comparisons were performed at two levels of the phylogeny: first, between the two primate species, marmosets and humans; and second, between mice and primates. In the first case, the data from the two species were each used as the testing and training sets across two folds of cross-validation, reporting the average performance (AUROC) across folds. In the second case, the primate data were used as an aggregate training set, and performance in mice was reported. Results were compared to average within-species performance.

## Replicability of clusters

MetaNeighbour v1.9.1 (RRID SCR_016727) was used to provide a measure of neuronal subclass and cluster replicability within and across species. For this application, we tested all pairs of species (human–marmoset, marmoset–mouse, human–mouse) as well as testing within each species. After splitting the data by class, we identified highly variable genes using the get_variable_genes function from MetaNeighbour, yielding 928 genes for GABAergic and 763 genes for glutamatergic neuron classes, respectively. These were used as input for the MetaNeighbourUS function, which was run using the fast_version and one_vs_best parameters set to TRUE. Using the one_vs_best parameter means that only the two closest neighbouring clusters are tested for their similarity to the training cluster, with results reported as the AUROC for the closest neighbour over the second closest. AUROCs are plotted in heat maps in Extended Data Figs. 2, 3. Data to reproduce these figures can be found in Supplementary Table 9, and scripts are on GitHub (http://github.com/gillislab/MetaNeighbor).

## SNARE–seq2

**Sample preparation.** Human and marmoset primary motor cortex nuclei were isolated for SNARE–seq2 according to the following protocol: https://doi.org/10.17504/protocols.io.8tvhwn6 (ref. [6]). Fluorescence-activated nuclei sorting (FANS) was then performed on a FACSAria Fusion (BD Biosciences, Franklin Lakes, NJ), gating out debris from forward scatter (FSC) and side scatter (SSC) plots and selecting DAPI+ singlets (Extended Data Fig. 6a). Samples were kept on ice until sorting was complete and were used immediately for SNARE–seq2.

**Library preparation and sequencing.** A detailed step-by-step protocol for SNARE–seq2 has been outlined in a companion paper[28] and is available at https://doi.org/10.17504/protocols.io.be5gjg3w. The resulting libraries of accessible chromatin were sequenced on an MiSeq (Illumina, RRID SCR_016379) (read 1, 75 read cycles for the first end of accessible chromatin DNA; read 2, 94 read cycles for cell barcodes and UMIs; read 3, 8 read cycles for i5; read 4, 75 cycles for the second end of accessible chromatin DNA read) for library validation, then on a NovaSeq6000 (Illumina, RRID SCR_016387) using a 300-cycles reagent kit for data generation. RNA libraries were combined at equimolar ratios and sequenced on an MiSeq (Illumina) (read 1, 70 read cycles for cDNA; index 1, 6 read cycles for i7; read 2, 94 cycles for cell barcodes and UMI) for library validation, then on a NovaSeq6000 (Illumina) using a 200-cycles reagent kit for data generation.

**Data processing.** A detailed step-by-step pipeline for processing SNARE–seq2 data is provided elsewhere[28]. For RNA data, this involved the use of dropEst to extract cell barcodes and STAR (v2.5.2b) to align tagged reads to the genome (GRCh38 version 3.0.0 for humans; GCF 000004665.1 Callithrix jacchus-3.2 for marmosets). For data on accessible chromatin, this involved Snaptools v1.4.7 for alignment to the genome (cellranger-atac-GRCh38-1.1.0 for humans, GCF 000004665.1 Callithrix jacchus-3.2 for marmosets) and to generate snap objects for

processing using the R package SnapATAC v2. We generated 84,178 and 9,946 dual-omic single-nucleus RNA and accessible chromatin datasets from human ($n = 2$) and marmoset ($n = 2$) M1, respectively.

**Data analysis.** *Filtering for RNA quality*. For SNARE–seq2 data, quality filtering of cell barcodes and clustering analysis were first performed on transcriptomic (RNA) counts and used to inform subsequent quality filtering and analysis of accessible chromatin. Each cell barcode was tagged by an associated library batch identification code (for example MOP1, MOP2, and so on); RNA read counts associated with dT and n6 adaptor primers were merged; libraries were combined for each sample within each experiment and empty barcodes were removed using the emptyDrops() function of DropletUtils v1.6.1 (ref. [52]); mitochondrial transcripts were removed; and doublets were identified using the DoubletDetection v2.5 software[53] and removed. All samples were combined across experiments within species, and cell barcodes having greater than 200 and fewer than 7,500 genes detected were kept for downstream analyses. To further remove low-quality datasets, we applied a gene UMI ratio filter (gene.vs.molecule.cell.filter) using Pagoda2 v0.1.0 (https://github.com/hms-dbmi/pagoda2).

*Clustering of RNA data*. For human SNARE–seq2 RNA data, clustering analysis was first performed using Pagoda2, where counts were normalized to the total number per nucleus and batch variations were corrected by scaling expression of each gene to the dataset-wide average. After variance normalization, the top 6,000 overdispersed genes were used for principal component analysis. Clustering was performed using an approximate $k$-nearest neighbour graph (with $k$ values between 50 and 500) based on the top 75 principal components, and cluster identities were determined using the infomap community detection algorithm. Major cell types were identified using a common set of broad cell-type marker genes: *GAD1/GAD2* (GABAergic neurons), *SLC17A7/SATB2* (glutamatergic neurons), *PDGFRA* (oligodendrocyte progenitor cells), *AQP4* (astrocytes), *PLP1/MOBP* (oligodendrocytes), *MRC1* (perivascular macrophages), *PTPRC* (T cells), *PDGFRB* (vascular smooth muscle cells), *FLT1* (vascular endothelial cells), *DCN* (vascular fibroblasts) and *APBB1IP* (microglia) (Extended Data Fig. 6c). Low-quality clusters that showed very low gene/UMI detection rates, low marker gene detection and/or mixed cell-type marker profiles were removed. Oligodendrocytes were overrepresented (54,080 in total), possibly reflecting a deeper subcortical sampling than intended; therefore, to ensure a more balanced distribution of cell types, we capped the number of oligodendrocytes at 5,000 and repeated the PAGODA2 clustering as above. To achieve optimal clustering of the different cell types, we used different $k$ values to identify cluster subpopulations for different cell types (L2/3 glutamatergic neurons, $k = 500$; all other glutamatergic neurons, astrocytes, oligodendrocytes and OPCs, $k = 100$; GABAergic neurons, vascular cells and microglia/perivascular macrophages, $k = 50$). To assess the appropriateness of the chosen $k$ values, clusters were compared against SMARTer clustering of data generated on human M1 through correlation of cluster-averaged scaled gene-expression values using the corrplot v0.84 package (https://github.com/taiyun/corrplot) (Extended Data Fig. 6d). For cluster visualization, UMAP dimensional reduction was performed in Seurat (v3.1.0, RRID SCR_007322) using the top 75 principal components identified using Pagoda2 (RRID SCR_017094). For marmosets, clustering was initially performed using Seurat, where the top 2,000 variable features were selected from the mean variance plot using the 'vst' method and used for principal component analysis. UMAP embeddings were generated using the top 75 principal components. To harmonize cellular populations across platforms and modalities, snRNA-seq within-species cluster identities were predicted from both human and marmoset data. We used an iterative nearest-centroid classifier algorithm (see Methods subsection 'Mapping of samples to reference taxonomies') to generate probability scores for each SNARE–seq2 nucleus mapping to their respective species' snRNA-seq reference cluster (Cv3 for marmoset

and SMART-Seqv4 for human). Comparing the predicted RNA cluster assignment of each nucleus with Pagoda2-identified clusters showed highly consistent cluster membership using a Jaccard similarity index (Extended Data Fig. 6e), confirming the robustness of these cell identities discovered using different analysis platforms.

*Filtering for quality of accessible chromatin data, and peak calling*. Initial analysis of corresponding SNARE–seq2 data on chromatin accessibility was performed using SnapATAC v2 software (https://github.com/r3fang/SnapATAC) (https://doi.org/10.1101/615179). Snap objects were generated by combining individual snap files across libraries within each species. Cell barcodes were included for downstream analyses only if cell barcodes passed RNA quality filtering (see above) and showed more than 1,000 read fragments and 500 UMIs. Read fragments were then binned to 5,000-bp windows of the genome, and only those cell barcodes that showed a fraction of binned reads within promoters of greater than 10% (15% for marmosets) and less than 80% were kept for downstream analysis. For peak calling, pseudo bulk aggregates of reads were generated for each of the consensus RNA taxonomies, subclasses and classes using Snaptools. Given that comparable sequencing and sampling depths were achieved (Supplementary Table 14), pseudo bulk aggregates for peak calling included all within-species samples. Peaks were called using MACS2 v2.1.2 software (https://github.com/taoliu/MACS) using the runMACS function in SnapATAC and with the following options '–nomodel–shift 100–ext 200–qval 5e-2 –B–SPMR'. Peak counts by cell barcodes were then computed using the 'createPmat' function of SnapATAC.

*Clustering of accessible chromatin data*. The matrices for peak counts were filtered to keep only locations from chromosomes 1–22, X or Y, and processed using Seurat (v3.1.0) and Signac (v0.1.4) software[45] (https://satijalab.org). All peaks having at least 100 counts (20 for marmosets) across cells were used for dimensionality reduction using latent semantic indexing ('RunLSI' function) and visualized by UMAP using the first 50 dimensions (40 for marmosets).

*Calculating gene-activity scores*. For a gene-activity matrix from accessibility data, cis co-accessible sites and gene-activity scores were calculated using Cicero software (v1.2.0)[54] (https://cole-trapnell-lab.github.io/cicero-release/). The binary peak matrix was used as input, with the expression-family variable set to 'binomialff' to make the aggregated input Cicero CDS object using the UMAP coordinates derived from accessible chromatin peaks, and setting 50 cells to aggregate per bin. Co-accessible sites were then identified using the 'run_cicero' function using default settings, and modules of cis co-accessible sites were identified using the 'generate_ccans' function. Co-accessible sites were annotated to a gene if they fell within a region spanning 10,000 bp upstream and downstream of the gene's transcription start site (human) or within 5,000 bp of the gene body (marmoset). The Cicero gene activity matrix was then calculated using the 'build_gene_activity_matrix' function using a co-accessibility cutoff of 0.25 and added to a separate assay of the Seurat object.

*Integrating data on RNA and accessible chromatin*. To reconcile the differing resolutions achievable from RNA and accessible chromatin (Extended Data Fig. 6f–k), we carried out an integrative analysis using Seurat. Transfer anchors were identified between the activity and RNA matrices using the 'FindTransferAnchors' function. For human data, transfer anchors were generated using an intersected list of variable genes identified from Pagoda2 analysis of RNA clusters (top 2,000 genes) and marker genes for clusters identified from SSv4 data (2,492 genes having β-scores greater than 0.4), together with canonical correlation analysis (CCA) for dimension reduction. For marmoset data, transfer anchors were generated using an intersected list of variable genes identified using Seurat (top 2,000 genes) and DEGs identified between marmoset RNA clusters (Cv3 snRNA-seq data, $P < 0.05$, top 100 markers per cluster). Imputed RNA expression values were then calculated using the 'TransferData' function from the Cicero gene activity matrix using normalized RNA expression values for reference and

LSI for dimension reduction. RNA and imputed expression data were merged, and a UMAP co-embedding and shared nearest neighbour (SNN) graph was generated using the top 50 principal components (40 for marmoset) and clusters identified ('FindClusters') using a resolution of 4. The resulting integrated clusters were compared with RNA clusters by calculating jaccard similarity scores using scratch.hicat software. Cell populations identified as T cells from Pagoda2 analysis (humans only) and those representing low-quality integrated clusters, showing a mixture of disparate cell types, were removed from these analyses. RNA clusters were assigned to co-embedded clusters on the basis of the highest jaccard similarity score and frequency, and then merged to generate the best matched co-embedded clusters, taking into account cell type and subclass to ensure more accurate merging of ambiguous populations. This produced clusters based on accessible chromatin that directly match the RNA-defined populations (Extended Data Fig. 6k). For RNA cluster and subclass level predictions (Extended Data Fig. 6g), we used the Seurat 'TransferData' function to transfer RNA cluster or subclass labels to accessible-chromatin data using the precomputed transfer anchors and LSI dimensionality reduction.

*Final peaks of accessible chromatin and gene-activity matrices.* A final combined list of peak regions was generated using MACS2, as detailed above, for all cell populations corresponding to RNA consensus taxonomies (more than 100 nuclei), accessibility level, subclass level (more than 50 nuclei) and class level barcode groupings. The final peak by cell barcode count matrix was generated by SnapATAC and used to establish a Seurat object as outlined above, with peak counts, Cicero gene activity scores and RNA expression values for matched cell barcodes contained within different assay slots. To confirm the appropriateness of calling peaks on cell barcode groupings that included both samples, we found that 93% of peak regions called by MACS2 on clusters at the accessible-chromatin level for the H18.30.001 sample overlapped with peak regions called independently for H18.30.002. Clusters at the accessible-chromatin level also showed similar coverage across individual samples (Extended Data Fig. 7a–d), and peak counts were highly correlated across experiments (mean Pearson's correlation coefficient ($r$) of 0.99 for humans and 0.98 for marmosets). Furthermore, gene activity estimates based on cis-regulatory interactions predicted from co-accessible promoter and distal peak regions using Cicero[54] were highly correlated with RNA expression values (Extended Data Fig. 7e, f). Dimensionality reduction using LSI on peak counts for final visualization (Extended Data Fig. 7a–c) was performed as above.

*Dual-omic data.* Following quality-control filtering for RNA and accessible chromatin (including limiting the representation of oligodendrocytes for humans) and modality integration, we obtained 84,178 and 9,946 dual-omic single-nucleus RNA and accessible-chromatin datasets from human ($n = 2$) and marmoset ($n = 2$) M1, respectively (Extended Data Fig. 6a, b and Supplementary Table 13). On average, 2,242 genes were detected per nucleus for humans and 3,858 genes per nucleus for marmosets, owing to the more than fourfold greater sequencing depth for marmosets (average 17,576 reads per nucleus for humans and 77,816 reads per nucleus for marmosets). In human and marmoset cells, we identified a total of 273,103 and 134,769 regions of accessible chromatin, and an average of 1,527 or 1,322 unique peaks of accessible chromatin per nucleus, respectively.

*Analysis of transcription-factor motifs.* Jaspar motifs (JASPAR2020, all vertebrate) were used to generate a motif matrix and motif object that was added to the Seurat object using Signac ('CreateMotifMatrix', 'CreateMotifObject', 'AddMotifObject'); and GC content, region lengths and dinucleotide base frequencies were calculated using the 'RegionStats' function. For motif activity scores, chromVAR v1.8.0 (https://greenleaflab.github.io/chromVAR)[55] was carried out according to default parameters (marmosets) or using the Signac 'RunChromVAR'" function on the peak count matrix (humans). The chromVAR deviation score matrix was then added to a separate assay slot of the Seurat object, and differential activities (or deviation scores) of TFBSs

between different populations were assessed using the 'Find[All]Markers' function through logistic regression and using the number of peak counts as a latent variable.

To examine non-redundant TFBS families, we downloaded motif collections generated by matrix clustering[56] from the JASPAR database (http://jaspar.genereg.net/matrix-clusters/), and used them to generate averaged chromVAR TFBS activities by subclass. Select motif clusters were visualized using ggHeat plotting function (SWNE package v0.5.7, https://github.com/yanwu2014/swne).

*Identification of DARs.* To compare DARs between cell populations (Fig. 3b and Extended Data Fig. 7g, h), we identified DARs that are significantly enriched within each cell grouping against a selection of background cells having best matched total peak counts. In this way, we identified DARs for each cell population, while taking into account any technical artefacts associated with the total accessibility for each cell. This involved calculating the total peaks in each cell on the basis of the accessibility matrix, estimating the distribution of total peaks (depth distribution) for the cells belonging to the test cluster, and randomly sampling cells (10,000 for humans and 2,000 for marmosets) from the rest of the clusters in a weighted way to select cells that have similar depth distribution to the test cluster. DARs were then identified as significantly enriched in the positive cells over selected background cells using the 'CalcDiffAccess' function, chromfunks v0.3.0 (https://github.com/yanwu2014/chromfunks), where $P$-values were calculated using a Fisher's exact test on a hypergeometric distribution[6], and adjusted $P$ values (or $q$ values) were calculated using the Benjamini–Hochberg method. To compare DAR proportions across subclasses and species, we subsampled subclasses (maximum of 500 for humans and 200 for marmosets) and identified DARs using the 'CalcDiffAccess' function as above. AUC values, testing the separation power of a specific DAR among different major clusters, were then calculated from the term frequency–inverse document frequency (TF–IDF) normalized peak by cell matrix using getDifferentialGenes and auc functions from the pagoda2 and pROC v1.16.2 packages. To visualize subsampled subclass DARs, we selected significant human ($q < 0.005$ and log-transformed fold change > 1) and marmoset ($q < 0.05$ and log-transformed fold change > 1) DARs passing an upper quantile AUC cutoff. For clusters of accessible chromatin and RNA in humans, we selected up to the top 100 DARs on the basis of log-transformed fold change values (accessible chromatin, $q < 0.01$ and log-transformed fold change > 1; RNA, $q < 0.05$ and log-transformed fold change > 1). Averaged accessibility values by cell grouping were then calculated, scaled (trimming values to a minimum of 0 and a maximum of 5), and visualized using the ggHeat plotting function (SWNE package).

To identify conservation of DARs between humans and marmosets, we found that 97% of marmoset DARs could be aligned to the human genome on the basis of at least 10% of matched bases using the LiftOver tool (https://genome.ucsc.edu/cgi-bin/hgLiftOver). DARs in each subclass were considered conserved if they were located within 1 kb of the aligned genomic location based on the overlap of genomic locations between species using the 'findOverlaps' function in the *GenomicRanges* v1.38.0 package.

*Linking DARs to marker genes.* We identified marker genes for the clusters of accessible chromatin by comparing the gene expression from cells in each cluster with a weighted sampling of background cells from the remaining clusters. Wilcoxon tests were used to calculate the $z$-scores and adjusted $P$ values for individual genes using 'getDifferentialGenes' function from the pagoda2 package. Genes were ranked by calculating AUC values, and DARs for the corresponding clusters were identified using the method described above. For each identified DAR, we assigned it to the nearest gene. The top expressed genes and associated DARs that were located within 500 kb of the gene region in each cluster of accessible chromatin were considered as associated targets. To further identify targets that have a direct link between DARs and gene expression, we trained a random forest regression model to

predict changes in gene expression in each cluster of accessible chromatin on the basis of the features extracted from its assigned DARs. The significant targets were then identified by comparing the regression model and a background model generated by random permutation. The union of the top predictive targets and identified marker genes and their associated DARs was selected and visualized.

*Correlation analyses*. To correlate RNA expression and associated accessible-chromatin activities for clusters at the levels of RNA and accessible chromatin (Extended Data Fig. 7e, f), we generated average scaled expression values and carried out pairwise correlations for marker genes identified from an intersected list of variable genes from Pagoda2 analysis of RNA clusters (top 2,000 genes) and marker genes for clusters identified from SSv4 data (2,492 genes having β-scores of greater than 0.4). For correlation of TFBS activities across species (Fig. 3), chromVAR TFBS activity scores for all Jaspar motifs found to be differentially active across marmoset or human subclasses ($P < 0.05$) were averaged, scaled for each species separately, and then correlated. Averaged scaled gene-expression values for the corresponding transcription factor were also correlated. Variable genes identified from both human and marmoset SNARE–seq2 RNA data using Seurat FindVariableFeatures function (selection.method = 'vst', nfeatures = 3,000) were used to generate averaged scaled expression values and correlated. Correlations between human and marmoset cell subclasses were visualized as boxplots for TFBS activities, expression of transcription factors, and variable genes using the R package *ggplot2* v3.3.2 (ref. [57]).

*Plots and figures*. All UMAP, feature, dot, and violin plots were generated using Seurat. Correlation plots were generated using the corrplot package. Connection plots were generated using Cicero and visualized using Gviz v1.30.3 (ref. [58]). To generate BigWig files for genome browser tracks, we compiled bam files for each cluster and normalized fragments using trimmed mean of *M*-values (TMM) to better account for differences in size (total fragments) and signal-to-noise ratios between clusters. For this, inverse scale factors were calculated using EdgeR v3.28.1 (ref. [59]) for each cluster on the basis of a subset of fragments that overlap chromosome 22. BigWig files were then generated using deepTools v3.4.2 bamCoverage[60] with the following options: (–ignoreDuplicates–minFragmentLength 0–maxFragmentLength 1000–binSize 50–scaleFactor). Genome browser tracks were generated using the Integrative Genomics Viewer (IGV v2.7.0).

## Single-cell methylome data (snmC-seq2)

**Sequencing and quantification.** *Library preparation and Illumina sequencing*. Single nuclei were isolated from human and marmoset M1 tissue as described above for RNA-seq profiling, and for mouse tissue as detailed in ref. [5]. Single nuclei were labelled with an anti-NeuN antibody and isolated by fluorescence-activated cell sorting (FACS), and neurons were enriched (90% NeuN+ nuclei) to increase detection of rare types. Mouse experiments were approved by the Salk IACUC under protocol number 18-00006. Detailed methods for bisulfite conversion and library preparation were previously described for snmC-seq2 (refs. [4,30]). The snmC-seq2 libraries generated from mouse brain tissues were sequenced using an Illumina Novaseq 6000 instrument with S4 flowcells and 150 bp paired-end mode. We generated 6,095, 6,090 and 9,876 single-nucleus methylcytosine datasets from M1 of humans ($n = 2$), marmosets ($n = 2$), and mice, respectively.

*Mapping and feature-count pipeline*. We implemented a versatile mapping pipeline (http://cemba-data.rtfd.io) for all the single-cell methylome-based technologies developed by our group[4,30,61]. The main steps of this pipeline included: first, demultiplexing FASTQ files into single cells; second read-level quality control; third, mapping; fourth BAM file processing and quality control; and fifth, final generation of molecular profiles. The details of the five steps for snmC-seq2 were described previously[30]. We mapped all of the reads from the three corresponding species onto the human hg19 genome, the marmoset ASM275486v1 genome, and the mouse mm10 genome. After mapping, we calculated the methyl-cytosine counts and total cytosine counts for two sets of genome regions in each cell: the non-overlapping chromosome 100-kb bins of each genome (the methylation levels of which were used for clustering analysis) and the gene-body regions (the methylation levels of which were used for cluster annotation and integration with RNA expression data). On average, 5.5% of human, 5.6% of marmoset and 6.2% of mouse genomes were covered by stringently filtered reads per cell, with $3.4 \times 10^4$, $1.8 \times 10^4$ and $4.5 \times 10^4$ genes detected per cell in the three species.

**Quality control and preprocessing.** *Cell filtering*. We filtered the cells on the basis of the following main mapping metrics: first, an mCCC rate of less than 0.03 (the mCCC rate reliably estimates the upper bound of the bisulfite non-conversion rate[4]); second, an overall mCG rate of 0.5; third, an overall mCH rate of less than 0.2; fourth, total final reads of more than 500,000; and fifth, a bismark mapping rate of more than 0.5. Other metrics such as genome coverage, rate of PCR duplicates, and index ratio were also generated and evaluated during filtering. However, after removing outliers with the main metrics 1–5, few additional outliers could be found.

*Feature filtering*. We filtered 100-kb genomic bin features by removing bins with mean total cytosine base calls of less than 250 or more than 3,000. We also excluded regions that overlap with the ENCODE blacklist[62] from further analysis.

*Computation and normalization of the methylation rate*. For CG and CH methylation, the computation of methylation rate from the methyl-cytosine and total cytosine matrices contains two steps: first, prior estimation for the beta-binomial distribution; and second, posterior rate calculation and normalization per cell.

In step 1, for each cell we calculated the sample mean, $m$, and variance, $v$, of the raw methylcytosine rate for each sequence context (CG, CH). The shape parameters ($\alpha, \beta$) of the beta distribution were then estimated using the method of moments:

$$a = m(m(1 - m)/v - 1)$$

$$\beta = (1 - m)(m(1 - m)/v - 1)$$

This approach used different priors for different methylation types for each cell and used weaker priors for cells with more information (higher raw variance).

In step 2, we calculated the posterior $\hat{m}c = \alpha + mc/\alpha + \beta + cov$, where cov is the total read number and mc is the number of reads supporting methylation. We normalized this rate by the cell's global mean methylation, $m = \alpha/(\alpha + \beta)$. Thus, all the posterior $\hat{m}c$ with 0 cov will be a constant 1 after normalization. The resulting normalized mc rate matrix contains no 'not available' (NA) values, and features with less cov tend to have a mean value close to 1.

*Selection of highly variable features*. Highly variable methylation features were selected on the basis of a modified approach using the scanpy v1.4.4 package scanpy.pp.highly_variable_genes function[63]. In brief, the scanpy.pp.highly_variable_genes function normalized the dispersion of a gene by scaling with the mean and standard deviation of the dispersions for genes falling into a given bin for mean expression of genes. In our modified approach, we reasoned that both the mean methylation level and the mean cov of a feature (100-kb bin or gene) could impact the dispersion of the mc rate. We grouped features that fall into a combined bin of mean and cov, and then normalized the dispersion within each mean–cov group. After dispersion normalization, we selected the top 3,000 features based on normalized dispersion for clustering analysis.

*Dimension reduction and combination of different mC types*. For each selected feature, mc rates were scaled to unit variance, and zero mean. Principal component analysis (PCA) was then performed on the scaled mc rate matrix. The number of important principal components was

selected by inspecting the variance ratio of each principal component using the elbow method. The CH and CG principal components were then concatenated together for further analysis in clustering and manifold learning.

**Data analysis.** *Consensus clustering on concatenated principal components.* We used a consensus clustering approach based on multiple Leiden-clustering[64] over a *k*-nearest neighbour (KNN) graph to account for the randomness of the Leiden clustering algorithms. After selecting dominant principal components from PCA in both mCH and mCG matrices, we concatenated the principal components together to construct a KNN graph using scanpy.pp.neighbours with Euclidean distance. Given fixed resolution parameters, we repeated the Leiden clustering 300 times on the KNN graph with different random starts, and combined these cluster assignments as a new feature matrix, where each single Leiden result is a feature. We then used the outlier-aware DBSCAN algorithm from the scikit-learn v0.21.3 package (RRID SCR_002577) to perform consensus clustering over the Leiden feature matrix using the hamming distance. Different epsilon parameters of DBSCAN are traversed to generate consensus cluster versions with the number of clusters that range from the minimum to the maximum number of clusters observed in the multiple Leiden runs. Each version contained a few outliers that usually fall into three categories: first, cells located between two clusters that had gradient differences instead of clear borders; second, cells with a low number of reads that potentially lack information in essential features to determine the specific cluster; and third, cells with a high number of reads that were potential doublets. The amount of the first and second types of outliers depends on the resolution parameter and is discussed in the 'Choice of resolution parameter' section below. The third type of outliers were very rare after cell filtering. The supervised model evaluation then determined the final consensus cluster version.

*Supervised model evaluation of the clustering assignment.* For each consensus clustering version, we performed a recursive feature elimination with cross-validation (RFECV)[65] process from the scikit-learn package to evaluate clustering reproducibility. We first removed the outliers from this process, and then we held out 10% of the cells as the final testing dataset. For the remaining 90% of the cells, we used tenfold cross-validation to train a multiclass prediction model using the input principal components as features and sklearn.metrics.balanced_accuracy_score[66] as an evaluation score. The multiclass prediction model is based on BalancedRandomForestClassifier from the imblearn v0.0 package that accounts for imbalanced classification problems[67]. After training, we used the 10% testing dataset to test the model performance using the balanced_accuracy_score score. We kept the best model and corresponding clustering assignments as the final clustering version. Finally, we used this prediction model to predict outliers' cluster assignments, and rescued those with a prediction probability of more than 0.3, otherwise labelling them as outliers.

*Choice of resolution parameter.* Choosing the resolution parameter of the Leiden algorithm is essential for determining the final number of clusters. We selected the resolution parameter according to three criteria: first, the portion of outliers is less than 0.05 in the final consensus clustering version; second, the ultimate accuracy of the model's prediction is more than 0.95; and third, the average number of cells per cluster is 30 or more, thereby controlling the cluster size to reach the minimum coverage required for further epigenome analysis such as DMR calls. All three criteria prevented the oversplitting of the clusters; thus, we selected the maximum resolution parameter to meet the criteria using a grid search.

*Three levels of iterative clustering analysis.* We used an iterative approach to cluster the data into three levels of categories with the consensus clustering procedure described above. In the first level, termed CellClass, clustering analysis is done on all cells. The resulting clusters are then manually merged into three canonical classes,

glutamatergic neurons, GABAergic neurons and non-neurons, based on marker genes. The same clustering procedure is then conducted within each CellClass to obtain clusters as the MajorType level. Within each MajorType, we obtain the final clusters as the SubTypes in the same way. *Integrating cell clusters identified from snmC-seq2 and from Cv3.* We identified gene markers on the basis of gene-body hypomethylation for each level of clustering of snmC-seq2 data using our in-house analysis utilities (https://github.com/lhqing/cemba_data), and identified gene markers for cell classes and subclasses from Cv3 analysis using scanpy[63]. We then used Scanorama v1.0 (ref. [68]) to integrate the two modalities with the markers identified (Supplementary Table 23). Methylome tracks at the subclass level can be found at http://neomorph.salk.edu/aj2/pages/cross-species-M1/.

*Calling CG DMRs.* We identified CG DMRs using methylpy v1.4.0 (https://github.com/yupenghe/methylpy) as described[69]. Briefly, we first called CG differentially methylated sites and then merged them into blocks if they both showed similar sample-specific methylation patterns and were within 250 bp. Normalized relative lengths of DMRs (Fig. 4d) were calculated by summing the lengths of DMRs and the surrounding 250 bp, and dividing by the numbers of cytosines covered in sequencing.

*Analysis of enriched TFBS motifs.* For each cell subclass (cluster), we analysed enriched TFBS motifs for hypomethylated DMRs compared with the hypomethylated DMRs from other cell subclasses (clusters) using software AME[70]. DMRs and surrounding 250-bp regions were used in the analysis. Enrichment results are reported as significance (*P* values) and effect sizes ($\log_2$(true positives/false positives).

## Characterization of chandelier cells

**Morphology.** Morphological reconstructions of *Pvalb*-expressing ChC and basket cells were obtained from human donors using the patch–seq protocol described below for L5 extratelencephalic neurons. Macaque reconstructions were from source data available in Neuromorpho[71,72]. Mouse cells also appear in ref. [73].

**Mouse ATAC-seq: data acquisition and analysis.** Chandelier cells are rare in mouse cortex and were enriched by isolating individual neurons from transgenically labelled mouse primary visual cortex (VISp). Many of the transgenic mouse lines have previously been characterized by single-cell RNA-seq[2]. Single-cell suspensions of cortical neurons were generated as described[2] and subjected to tagmentation (ATAC-seq)[74,75]. Mixed libraries containing 60–96 samples were sequenced on an Illumina MiSeq. In total, 4,275 single cells were collected from 36 driver-reporter combinations in 67 mice. After sequencing, raw FASTQ files were aligned to the GRCm38 (mm10) mouse genome using Bowtie v1.1.0 (RRID SCR_005476) as described[76]. Following alignment, duplicate reads were removed using samtools v1.9 rmdup, which yielded only single copies of uniquely mapped paired reads in BAM format. Quality-control filtering was applied to select samples with more than 10,000 uniquely mapped paired-end fragments, more than 10% of which were longer than 250 base pairs and with more than 25% of their fragments overlapping high-depth cortical DNase-seq peaks from ENCODE[77]. The resulting dataset contained a total of 2,799 samples.

To increase the cell-type resolution of chromatin-accessibility profiles beyond that provided by driver lines, we used a feature-free method for computation of pairwise distances (Jaccard). Using Jaccard distances, we carried out PCA and *t*-SNE, followed by Phenograph v1.5.2 (RRID SCR_016919) clustering[78]. This clustering method grouped cells from class-specific driver lines together, but also segregated them into multiple clusters. Phenograph-defined neighbourhoods were assigned to cell subclasses and clusters by comparing accessibility near transcription start sites (TSS ± 20 kb) to median expression values of scRNA-seq clusters at the cell-type and subclass levels from mouse primary visual cortex[79]. From this analysis, we assigned a total of 226 samples to *Pvalb* and 124 samples to *Pvalb Vipr2* (ChC) clusters. The

sequence data for these samples were grouped together and further processed through the Snap-ATAC pipeline.

Mouse scATAC-seq peak counts for *Pvalb* and ChC were used to generate a Seurat object as outlined above for human and marmoset SNARE–seq2 data on accessible chromatin. Cicero cis co-accessible sites were identified, gene-activity scores calculated, and motif-enrichment analyses performed as above. Genes used for motif enrichment were ChC markers identified from differential expression analysis between *PVALB*-positive clusters in mouse Cv3 scRNA-seq data (with an adjusted *P* value of less than 0.05).

### Patch–seq
**Participants.** The human neurosurgical specimen was obtained from a 61-year-old female patient who underwent deep tumour resection (glioblastoma) from the frontal lobe at a local hospital. The patient provided informed consent and experimental procedures were approved by the hospital institute review board before commencing the study. Post hoc analysis revealed that the neocortical tissue obtained from this patient was from a premotor region near the confluence of the superior frontal gyrus and the precentral gyrus (Fig. 6g). Betz cells are enriched in the primary motor cortex, but they are also present in premotor cortex (area 6; refs. [14,80,81]; Allen Institute Human Brain Reference Atlas). These neurons have several histological hallmarks of Betz cells (including gigantocellular somata, horizontally emanating dendrites and abundant rough endoplasmic reticulum[14]). In addition, as can be seen in the biocytin images in Fig. 6, the recorded neurons possessed large somata with many perisomatic dendrites. Additional histological hallmarks of Betz cells cannot be assessed in biocytin-filled neurons.

All procedures involving macaques and mice were approved by the IACUC at either the University of Washington or the Allen Institute for Brain Science. Macaque M1 tissue was obtained from male ($n$ = 4) and female ($n$ = 5) animals (mean age = 10 ± 2.21 years) designated for euthanasia from the University of Washington National Primate Resource Center, under a protocol approved by the University of Washington UACUC. Mouse M1 tissue was obtained from 4–12-week-old male and female mice from the following transgenic lines: *Thy1h*–eyfp (B6.Cg-Tg(*Thy1*–YFP)-HJrs/J; RRID IMSR_JAX:003782); *Etv1*–egfp (Tg(*Etv1*–EGFP)BZ192Gsat/Mmucd; RRID MMRRC_011152-UCD) (animals maintained on an outbred Charles River Swiss Webster background (Crl:CFW(SW; RRID IMSR_CRL:024)); and C57BL/6-Tg(*Pvalb*–tdTomato)15Gfng/J; RRID IMSR_JAX:027395). Mice were provided food and water ad libitum and were maintained on a regular 12-h day/night cycle with no more than five adult animals per cage.

**Preparation of brain slices.** Brain slices were prepared in a similar way for *Pvalb*–TdTomato mice and macaque and human samples. Upon resection, human neurosurgical tissue was immediately placed in a chilled and oxygenated solution formulated to prevent excitotoxicity and preserve neural function[82]. This artificial cerebrospinal fluid (NMDG aCSF) consisted of (in mM): 92 *N*-methyl-D-glucamine (NMDG), 2.5 KCl, 1.25 NaH$_2$PO$_4$, 30 NaHCO$_3$, 20 4-(2-hydroxyethyl)-1-piperazineethanesulfonic acid (HEPES), 25 glucose, 2 thiourea, 5 sodium ascorbate, 3 sodium pyruvate, 0.5 CaCl$_2$·4H$_2$O and 10 MgSO$_4$·7H$_2$O. The pH of the NMDG aCSF was titrated to 7.3–7.4 with concentrated hydrochloric acid, and the osmolality was 300–305 mOsmoles per kilogram. The solution was prechilled to 2–4 °C and thoroughly bubbled with carbogen (95% O$_2$/5% CO$_2$) before collection. Macaques were anaesthetized with sevoflurane gas, during which the entire cerebrum was extracted and placed in the protective solution described above. After extraction, macaques were euthanized with sodium-pentobarbital. We dissected the trunk/limb area of the primary motor cortex to prepare brain slices. *Pvalb*–TdTomato mice were deeply anaesthetized by intraperitoneal administration of Avertin (20 mg kg$^{-1}$) and were perfused through the heart with NMDG aCSF (bubbled with carbogen).

Brains were sliced at 300-µm thickness on a vibratome using the NMDG protective recovery method and a zirconium ceramic blade[83,84]. Mouse brains were sectioned coronally, and human and macaque brains were sectioned such that the angle of slicing was perpendicular to the pial surface. After sections were obtained, slices were transferred to a warmed (32–34 °C) initial recovery chamber filled with NMDG aCSF under constant carbogenation. After 12 min, slices were transferred to a chamber containing an aCSF solution consisting of (in mM): 92 NaCl, 2.5 KCl, 1.25 NaH$_2$PO$_4$, 30 NaHCO$_3$, 20 HEPES, 25 glucose, 2 thiourea, 5 sodium ascorbate, 3 sodium pyruvate, 2 CaCl$_2$·4H$_2$O and 2 MgSO$_4$·7H$_2$O, continuously bubbled with 95% O$_2$/5% CO$_2$. Slices were held in this chamber for use in acute recordings or transferred to a six-well plate for long-term culture and viral transduction. Cultured slices were placed on membrane inserts and wells were filled with culture medium consisting of 8.4 g l$^{-1}$ MEM Eagle medium, 20% heat-inactivated horse serum, 30 mM HEPES, 13 mM D-glucose, 15 mM NaHCO$_3$, 1 mM ascorbic acid, 2 mM MgSO$_4$·7H$_2$O, 1 mM CaCl$_2$.4H$_2$O, 0.5 mM GlutaMAX-I and 1 mg l$^{-1}$ insulin[83]. The slice culture medium was carefully adjusted to pH 7.2–7.3, an osmolality of 300–310 mOsmoles per kilogram by addition of pure H$_2$O, sterile-filtered and stored at 4 °C for up to 2 weeks. Culture plates were placed in a humidified 5% CO$_2$ incubator at 35 °C, and the slice culture medium was replaced every two to three days until endpoint analysis. One to three hours after brain slices were plated on cell culture inserts, brain slices were infected by direct application of concentrated AAV viral particles over the slice surface[80].

For mouse M1, the extratelencephalic-specific *Thy1*–YFP-H[41,84] and intratelencephalic-specific *Etv1*–EGFP[85] lines preferentially labelled physiologically defined extratelencephalic and non-extratelencephalic neurons, respectively (Fig. 6h, i). Thy1 and Etv1 mice were deeply anaesthetized by intraperitoneal administration of a ketamine (130 mg kg$^{-1}$) and xylazine (8.8 mg kg$^{-1}$) mix and were perfused through the heart with chilled (2–4 °C) sodium-free aCSF consisting of (in mM): 210 sucrose, 7 D-glucose, 25 NaHCO$_3$, 2.5 KCl, 1.25 NaH$_2$PO$_4$, 7 MgCl$_2$, 0.5 CaCl$_2$ 1.3 sodium ascorbate and 3 sodium pyruvate, bubbled with carbogen (95% O$_2$/5% CO$_2$). Near-coronal slices, 300-µm thick, were generated using a Leica vibratome (VT1200) in the same sodium-free aCSF, and were transferred to warmed (35 °C) holding solution (in mM): 125 NaCl, 2.5 KCl, 1.25 NaH$_2$PO$_4$, 26 NaHCO$_3$, 2 CaCl$_2$, 2 MgCl$_2$, 17 dextrose and 1.3 sodium pyruvate, bubbled with carbogen (95% O$_2$/5% CO$_2$). After 30 min of recovery, the chamber holding the slices was allowed to cool to room temperature.

**Patch-clamp electrophysiology.** Macaque, human and *Pvalb*–TdTomato mouse brain slices were placed in a submerged, heated (32–34 °C) recording chamber that was continually perfused (at a rate of 3–4 ml min$^{-1}$) with aCSF under constant carbogenation and containing (in mM) 1): 119 NaCl, 2.5 KCl, 1.25 NaH$_2$PO$_4$, 24 NaHCO$_3$, 12.5 glucose, 2 CaCl$_2$·4H$_2$O and 2 MgSO$_4$·7H$_2$O (pH 7.3–7.4). Slices were viewed with an Olympus BX51WI microscope using infrared differential interference contrast (IR-DIC) optics and a 40× water-immersion objective. The infragranular layers of macaque primary motor cortex and human premotor cortex are heavily myelinated, which makes visualization of neurons under IR-DIC almost impossible. To overcome this challenge, we labelled neurons using various viral constructs in organotypic slice cultures (Extended Data Fig. 12a). We were unable to use some classic histological markers of Betz cells (prominent rough endoplasmic reticulum, conspicuous nucleolus, intensity of anti-Nissl staining) for selection of neurons during patch-clamp experiments. Thus, we used the size of soma (greater than 40 µm in height or width) as the primary criterion, because somatic volume and/or height/width reasonably separates Betz cells from other pyramidal neurons[14,86,87]. Occasionally in the fluorescent image we observed additional hallmarks of Betz cells, namely large tap-root dendrites[88,89] and horizontal dendrites emanating directly from the somatic compartment. In many of these neurons, substantial lipofuscin could be observed. Finally, the size of

the biocytin-filled neuron in the example (Fig. 6) is at the upper end of the range in corticospinal neurons in macaque area 4 (20–60 μm)[90]. The conservative size criterion resulted in soma sizes that are consistent with the more than threefold enhancement of the volume in Betz cells compared with other pyramidal neurons, and match the size range of these neurons in adult macaques[86,87].

Patch pipettes (2–6 MΩ) were filled with an internal solution containing (in mM): 110.0 potassium gluconate, 10.0 HEPES, 0.2 EGTA, 4 KCl, 0.3 sodium GTP, 10 phosphocreatine disodium salt hydrate, 1 Mg-ATP, 20 μg ml⁻¹ glycogen, 0.5 U μl⁻¹ RNase inhibitor (Takara, catalogue number 2313A) and 0.5% biocytin (Sigma, B4261), pH 7.3. Fluorescently labelled neurons from *Thy1* or *Etv1* mice were visualized through a 40× objective using either Dodt contrast with a CCD camera (Hamamatsu) and/or a two-photon imaging/uncaging system from Prairie (Bruker) Technologies. Recordings were made in aCSF containing (in mM): 125 NaCl, 3.0 KCl, 1.25 NaH$_2$PO$_4$, 26 NaHCO$_3$, 2 CaCl$_2$, 1 MgCl$_2$, 17 dextrose and 1.3 sodium pyruvate bubbled with carbogen (95% O$_2$/5% CO$_2$) at 32–35 °C, with synaptic inhibition blocked using 100 μM picrotoxin. Sylgard-coated patch pipettes (3–6 MΩ) were filled with an internal solution containing (in mM): 135 potassium gluconate, 12 KCl, 11 HEPES, 4 MgATP, 0.3 NaGTP, 7 potassium phosphocreatine and 4 sodium phophocreatine (pH 7.42 with KOH) with neurobiotin (0.1–0.2%), Alexa 594 (40 μM) and Oregon Green BAPTA 6F (100 μM).

Whole-cell somatic recordings were acquired using either a Multiclamp 700B amplifier or an AxoClamp 2B amplifier (Molecular Devices), and were digitized using an ITC-18 (HEKA). Data-acquisition software was either MIES (https://github.com/AllenInstitute/MIES/; RRID SCR_016443) or custom software written in Igor Pro. Electrical signals were digitized at 20–50 kHz and filtered at 2–10 kHz. Upon attaining whole-cell current-clamp mode, the pipette capacitance was compensated and the bridge was balanced. Access resistance was monitored throughout the recording and was 8–25 MΩ.

**Data analysis.** Data were analysed using custom analysis software written in Igor Pro (RRID SCR_000325). All measurements were made at resting membrane potential. The input resistance ($R_N$) was measured from a series of 1-s hyperpolarizing steps from −150 pA to +50 pA in +20 pA increments. For neurons with low input resistance (for example, the Betz cells), this current-injection series was scaled by four times or more. The input resistance was calculated from the linear portion of the current/steady-state-voltage relationship generated in response to these current injections. The resonance ($f_R$) was determined from the voltage response to a constant-amplitude sinusoidal current injection (Chirp stimulus). The chirp stimulus increased in frequency either linearly from 1–20 Hz over 20 s, or logarithmically from 0.2–40 Hz over 20 s. The amplitude of the chirp stimulus was adjusted in each cell to produce a peak-to-peak voltage deflection of roughly 10 mV. The impedance amplitude profile (ZAP) was constructed from the ratio of the fast Fourier transform of the voltage response to the fast Fourier transform of the current injection. ZAPs were produced by averaging at least three presentations of the chirp stimulus, and were smoothed using a running median smoothing function. The frequency corresponding to the peak impedance ($Z_{max}$) was defined as the resonant frequency. Spike input/output curves were constructed in response to 1-s current injections (50–500 pA in 50-pA steps). For a subset of experiments, this current-injection series was extended to 3 nA in 600-pA steps to probe the full dynamic range of low-$R_N$ neurons. Analysis of the acceleration of spike frequency was performed for current injections that produced roughly ten spikes during the 1-s step. The acceleration ratio was defined as the ratio of the second to the last interspike interval. To examine the dynamics of spike timing over longer periods, we also measured spiking in response to current injections with 10-s steps, in which the amplitude of the current was adjusted to produce roughly five spikes in the first second. Properties of action potentials were measured for currents near rheobase. The threshold of action potentials was defined

as the voltage at which the first derivative of the voltage response exceeded 20 V s⁻¹. The width of action potentials was measured at half the amplitude between threshold and the peak voltage. The faster after-hyperpolarization was defined relative to threshold. We clustered mouse, macaque and human pyramidal neurons into two broad groups on the basis of their $R_N$ and $f_R$ values using Ward's algorithm. Macaque and human extratelencephalilc neurons were grouped for physiological analysis because their intrinsic properties were not substantially different, and because there is evidence that Betz cells can be found in premotor cortex as well as in M1[80,81].

**Biocytin histology.** We used a horseradish peroxidase (HRP)-based reaction−with diaminobenzidine (DAB) as the chromogen−to visualize filled cells after electrophysiological recording, and DAPI staining to identify cortical layers as described[91].

**Microscopy.** Mounted sections were imaged as described[91]. Briefly, operators captured images on an upright AxioImager Z2 microscope (Zeiss, Germany) equipped with an Axiocam 506 monochrome camera and 0.63× optivar. Two-dimensional tiled overview images were captured with a 20× objective lens (Zeiss Plan NEOFLUAR 20×/0.5) in brightfield transmission and fluorescence channels. Tiled image stacks of individual cells were acquired at higher resolution in the transmission channel only for the purpose of automated and manual reconstruction. Light was transmitted using an oil-immersion condenser (numerical aperture 1.4). High-resolution stacks were captured with a 63× objective lens (Zeiss Plan-Apochromat 63×/1.4 oil or Zeiss LD LCI Plan-Apochromat 63×/1.2 imm corr) at an interval of 0.28 μm (numerical aperture 1.4 NA; mouse specimens) or 0.44 μm (numerical aperture 1.2; human and non-human primate specimens) along the z-axis. Tiled images were stitched in ZEN 2012 SP2 software and exported as single-plane TIFF files.

**Morphological reconstruction.** Reconstructions of the dendrites and the full axon were generated based on a three-dimensional image stack that was run through a Vaa3D-based (v3.475) image processing and reconstruction pipeline as described[91].

**Production and transduction of viral vectors.** Recombinant AAV vectors were produced by triple transfection of enhancer plasmids containing inverted terminal repeats (ITRs) along with AAV helper and rep/cap plasmids using the HEK 293T/17 cell line (ATCC, CRL-11268), followed by harvest, purification and concentration of the viral particles. The plasmid supplying the helper function is available from a commercial source (Cell Biolabs). The PHP.eB capsid variant was generated by V. Gradinaru at the California Institute of Technology[92], and the DNA plasmid for AAV packaging is available from Addgene (RRID Addgene_103005). Quality control of the packaged AAV was determined by viral titring to determine that an adequate concentration was achieved (more than $5 \times 10^{12}$ viral genomes per millilitre), and by sequencing the AAV genome to confirm the identity of the viral vector that was packaged. Human and NHP L5 extratelencephalic neurons, including Betz cells, were targeted to cultured slices by transducing the slices with viral vectors that either generically label neurons (AAV−hSyn1−tdTomato), or that enrich for L5 extratelencephalic neurons by expressing reporter transgene under the control of the msCRE4 enhancer[79].

**Processing of patch−seq samples.** For a subset of experiments, the nucleus was extracted at the end of the recording and processed for RNA-seq. Before collecting data for these experiments, we thoroughly cleaned all surfaces with RNase Zap. The contents of the pipette were expelled into a PCR tube containing lysis buffer (Takara, 634894). cDNA libraries were produced using the SMART-Seq v4 ultra low input RNA kit for sequencing according to the manufacturer's instructions. We performed reverse transcription and cDNA amplification for 20 PCR

cycles. Sample proceeded through Nextera NT DNA library preparation using Nextera XT Index Kit V2 Set A (FC-131-2001).

**Isolating of macaque nuclei, RNA-seq and clustering.** Tissue was obtained from three macaque animals (aged 3–17 years, male and female; Extended Data Table 2) as above. As described for humans, M1 was isolated from thawed slabs using fluorescent Nissl staining (Neurotrace 500/525, ThermoFisher Scientific). Stained sections were screened for histological hallmarks of primary motor cortex, and L5 was dissected. Nuclei were isolated from the dissected layer; gene expression was quantified with 10× Chromium v3 using the Mmul_10 genome annotation; nuclei that passed quality-control criteria were clustered; and a taxonomy of glutamatergic types was defined. To identify which clusters in our three-species taxonomy aligned with macaque clusters from our L5 dissected Cv3 dataset, we carried out an identical integration workflow on glutamatergic neurons to that used for the three-species integration. Macaque clusters were assigned subclass labels on the basis of their corresponding alignment with subclasses from the other species.

**Mapping of samples to reference taxonomies.** To identify which cell type a given patch–seq nuclei mapped to, we used our previously described nearest-centroid classifier[2]. Briefly, a centroid classifier was constructed for glutamatergic reference data (human SSv4 or macaque Cv3) using marker genes for each cluster. Patch–seq nuclei were then mapped to the appropriate species reference 100 times, using 80% of randomly sampled marker genes during each iteration. Probabilities for each nucleus mapping to each cluster were computed over the 100 iterations, resulting in a confidence score ranging from 0 to 100. We identified four human patch–seq nuclei that mapped with greater than 85% confidence, and four macaque nuclei that mapped with greater than 93% confidence, to a cluster in the L5 extratelencephalic subclass.

### Reporting summary

Further information on research design is available in the Nature Research Reporting Summary linked to this paper.

### Data availability

Raw sequence data produced as part of the BRAIN Initiative Cell Census Network (BICCN; RRID SCR_015820) are available for download from the Neuroscience Multi-omics Archive (RRID SCR_016152; https://assets.nemoarchive.org/dat-ek5dbmu) and the Brain Cell Data Center (RRID SCR_017266; https://biccn.org/data). Visualization and analysis tools are available at NeMO Analytics (RRID SCR_018164; individual species, https://nemoanalytics.org//index.html?layout_id=ac9863bf; integrated species, https://nemoanalytics.org//index.html?layout_id=34603c2b) and Cytosplore Viewer (RRID SCR_018330; https://viewer.cytosplore.org/). These tools allow users to compare cross-species datasets and consensus clusters via genome and cell browsers and to calculate differential expression within and among species. Subclass-level methylome tracks are available at http://neomorph.salk.edu/aj2/pages/cross-species-M1/. A semantic representation of the cell types defined through these studies is available in the provisional Cell Ontology (RRID SCR_018332; https://bioportal.bioontology.org/ontologies/PCL; Supplementary Table 1).

The following publicly available datasets were used for analysis: Jaspar motifs database (JASPAR2020, all vertebrate, http://jaspar.genereg.net/matrix-clusters/), HUGO Gene Nomenclature Committee (HGNC) at the European Bioinformatics Institute (https://www.genenames.org; downloaded January 2020), Synaptic Gene Ontology (SynGO; downloaded February 2020), and orthologous genes across species from NCBI Homologene (downloaded November 2019). Macaque reconstructions were from source data available in Neuromorpho (chandelier cells, NeuroMorpho.org, NMO_01873; basket

cells, NeuroMorpho.org, NMO_01851). Mouse ATAC-seq data are available from https://assets.nemoarchive.org/dat-7qjdj84; MTG human SMARTseq v4 data from https://portal.brain-map.org/atlases-and-data/rnaseq/human-mtg-smart-seq and https://assets.nemoarchive.org/dat-swzf4kc); and ENCODE blacklist regions from http://mitra.stanford.edu/kundaje/akundaje/release/blacklists/hg38-human/hg38.blacklist.bed.gz.

### Code availability

Code to reproduce figures is available for download from https://github.com/AllenInstitute/BICCN_M1_Evo.

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

**Acknowledgements** We thank the tissue procurement, tissue processing and facilities teams at the Allen Institute for Brain Science for assistance with the transport and processing of postmortem and neurosurgical brain specimens; the technology team at the Allen Institute for assistance with data management; M. Vawter, J. Davis and the San Diego Medical Examiner's Office for assistance with postmortem tissue donations; X. Opitz-Araya and the Allen Institute for Brain Science viral technology team for AAV packaging; K. Brouner, A. Ruiz, T. Egdorf, A. Gary, M. Maxwell, A. Pom and J. Bomben for biocytin staining; N. Dotson, R. Enstrom, M. Hupp, L. Potekhina and S. Ransford for imaging biocytin-filled cells; L. Ng, D. Hill and R. Rajanbabu for patching the human and mouse cells in Figure 5 describing chandelier neurons; and S. Kebede, A. Mukora and G. Willams for reconstructing these cells. This work was funded by the Allen Institute for Brain Science and by US National Institutes of Health (NIH) grant U01 MH114812-02 to E.S.L. Support for the development of NS-Forest v.2 and the provisional cell ontology was provided by the Chan–Zuckerberg Initiative Donor-Advised Fund (DAF), an advised fund of the Silicon Valley Community Foundation (2018-182730). G.Q. is supported by National Science Foundation (NSF) CAREER award 1846559. S.O. is supported by the NARSAD Young Investigator Award. This work was partially supported by a Dutch Research Council (NWO) Gravitation project, BRAINSCAPES: A Roadmap from Neurogenetics to Neurobiology (NWO grant 024.004.012) and NWO TTW project 3DOMICS (NWO grant 17126). This project was supported in part by NIH grants P51OD010425 from the Office of Research Infrastructure Programs (ORIP) and UL1TR000423 from the National Center for Advancing Translational Sciences (NCATS). Its contents are solely the responsibility of the authors and do not necessarily represent the official view of NIH, ORIP, NCATS, the Institute of Translational Health Sciences or the University of Washington National Primate Research Center. This work is supported in part by NIH BRAIN Initiative award RF1MH114126 from the National Institute of Mental Health to E.S.L., J.T.T. and B.P.L.; NIH BRAIN Initiative awards U01 MH121282 to J.R.E and M.M.B, U19 MH114831 to J.R.E. and E.M.C., U19 MH114830 to H.Z., U01 MH114819 to G.F., 1U01MH114828 to K.Z. and J.C., RF1MH123220 to M.H. and R.H.S., and U19 MH114821. NIH awards R01DC019370 to R.H., R24MH114815 to R.H. and O.R.W., and R24 MH114788 to O.R.W. Nancy and Buster Alvord Endowment to C.D.K.; National Institute on Drug Abuse award R01DA036909 to B.T.; National Institute of Neurological Disorders and Stroke award R01NS044163 to W.J.S.; and the California Institute for Regenerative Medicine (GC1R-06673-B) and the Chan–Zuckerberg Initiative DAF, an advised fund of the Silicon Valley Community Foundation (2018–182730), to R.H.S. J.R.E. is an Investigator of the Howard Hughes Medical Institute. The authors thank the founder of the Allen Institute, Paul G. Allen, for his vision, encouragement and support.

**Author contributions** Generation of RNA data: A.M.Y., A.Re., A.T., B.B.L., B.T., C.D.K., C.R., C.R.P., C.S.L., D.B., D.D., D.M., E.S.L., E.Z.M., F.M.K., G.F., H.T., H.Z., J.C., J.Go., J.S., K.C., K.L., K.Si., K.Sm., K.Z., M.G., M.K., M.T., N.Dee, N.M.R., N.P., R.D.H., S.A.M., S.D., S.L., T.C., T.E.B., T.P., W.J.R. Generation of mC data: A.B., A.P., A.I.A., A.Ri., C.L., H.L., J.D.L., J.K.O., J.R.E., J.R.N., M.M.B., R.G.C. Generation of ATAC data: A.E.S., A.P., B.B.L., B.R., B.T., C.R.P., C.S.L., D.D., J.C., J.D.L., J.K.O., K.Z., L.T.G., M.M.B., N.P., S.P., W.J.R., X.H., X.W. Electrophysiology, morphology and generation of patch–seq data: A.L.K., B.E.K., D.M., E.S.L., G.D.H., J.Go., J.T.T., K.Sm., M.T., N.Dem., P.R.N., R.D., S.A.S., S.O., T.L.D., T.P., W.J.S. Data archive and infrastructure: A.E.S., A.M., B.R.H., H.C.B., J.A.M., J.Go., J.K., J.O., M.M., O.R.W., R.H., S.A.A., S.S., Z.Y. Cytosplore Viewer software: B.P.L., B.V.L., J.E., T.H. Data analysis: A.D., B.B.L., B.D.A., B.E.K., B.P.L., B.V.L., D.D., E.A.M., E.S.L., F.M.K., F.X., H.L., J.E., J.Gi., J.Go., J.R.E., J.T.T., K.Sm., M.C., N.Dem., N.L.J., O.P., P.V.K., Q.H., R.F., R.H.S., R.Z., S.F., S.N., S.O., T.E.B., T.H., W.D., W.T., Y.E.L., Z.Y. Data interpretation: A.D., A.Re., B.B.L., B.E.K., B.T., C.K., C.L., E.S.L., F.X., H.L., H.Z., J.Gi., J.Go., J.R.E., J.T.T., M.C., M.H., M.M.B., N.Dem., N.L.J., P.R.H., P.V.K., Q.H., R.D.H., R.H.S., R.Z., S.D., S.O., T.E.B., W.T., Y.E.L., Z.Y. Writing manuscript: A.D., B.B.L., B.E.K., C.K., E.S.L., F.M.K., M.C., N.Dem., N.L.J., P.R.H., Q.H., R.H.S., T.E.B., W.J.S., W.T.

**Competing interests** A.Re. is an equity holder and founder of Celsius Therapeutics, a founder of Immunitas, and a member of the Scientific Advisory Board at Syros Pharmaceuticals, Neogene Therapeutics, Asimov and Thermo Fisher Scientific. B.R. is a shareholder of Arima Genomics, Inc. K.Z. is a co-founder and equity holder and serves on the Scientific Advisory Board of Singlera Genomics. P.V.K. serves on the Scientific Advisory Board to Celsius Therapeutics Inc.

**Additional information**
**Correspondence and requests for materials** should be addressed to T.E.B. or E.S.L.

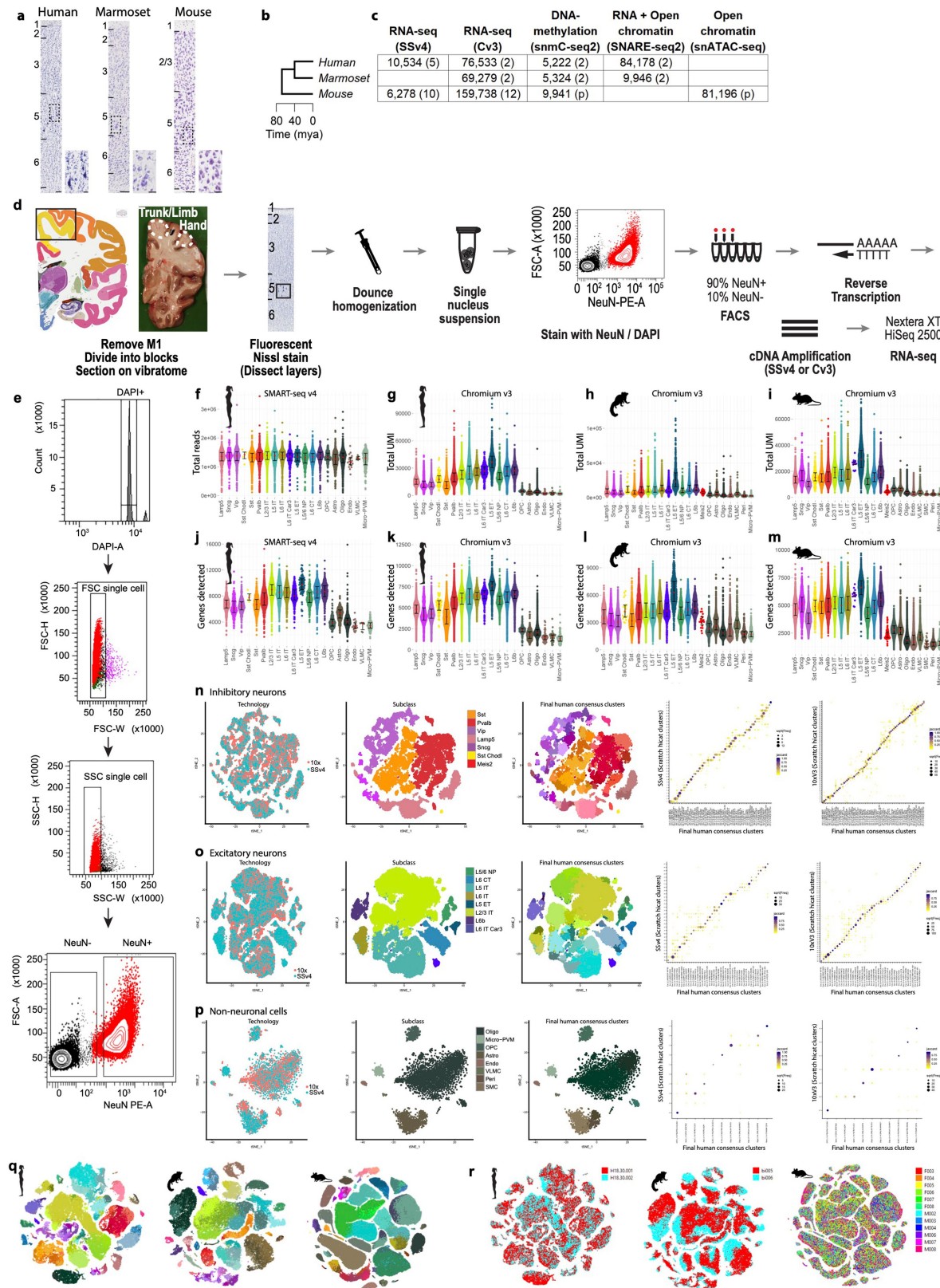

**Extended Data Fig. 1** | See next page for caption.

**Extended Data Fig. 1 | Metrics of RNA-seq quality and integration of human datasets. a**, Nissl-stained sections of M1, annotated with layers and showing the relative expansion of cortical thickness (particularly of L2 and L3 in primates) and large pyramidal neurons or 'Betz' cells in human L5 (black dotted outline, with high-magnification adjacent panel). Scale bars, 100 μm (low magnification), 20 μm (high magnification). M1 was identified in each species from its cortical location and histological features. **b**, Phylogeny of species; mya, millions of years ago. **c**, Number of nuclei included for analysis in each molecular assay. Numbers of donors are in parentheses; 'p' indicates pooled biological replicates. All assays used nuclei isolated from the same donors in humans and marmosets. 15,842 nuclei were also profiled from L5 in macaques ($n$ = 3) using Cv3. **d**, Workflow showing the isolation of single nuclei from M1 of post-mortem human brain and profiling with RNA-seq. The black outline in the Nissl image highlights a cluster of Betz cells in L5. **e**, FACS gating scheme for sorting nuclei. **f**, Using SSv4, we sequenced more than one million total reads across all subclasses in humans. **g**–**i**, Cv3 analysis shows that total unique molecular identifiers (UMIs) vary between subclasses, and that these differences are shared across species. For each subclass, single nuclei are plotted together with median values and interquartile intervals. **j**–**m**, Gene detection (expression level greater than 0) is highest in human when using SSv4 (**j**) and lowest for marmosets when using Cv3 (**l**). Note that the average read depth used for SSv4 was approximately 20-fold greater than that for Cv3 (target 60,000 reads per nucleus). For each subclass, single nuclei plus medians and interquartile intervals are plotted. **n**–**p**, Integration of SSv4 and Cv3 RNA-seq datasets from human single nuclei isolated from GABAergic (**n**) and glutamatergic (**o**) neurons and non-neuronal cells (**p**). Left three panels, UMAP visualizations, coloured by RNA-seq technology, cell subclass, and unsupervised consensus clusters. Right two panels, confusion matrices show membership of SSv4 and Cv3 nuclei within 127 integrated consensus clusters. **q**, **r**, *t*-SNE projections of single nuclei, based on expression of several thousand genes with variable gene expression and coloured by cluster label (**q**) or donor (**r**). Clusters are well separated in all species, and nuclei from different donors are well mixed within clusters, with some donor-specific technical effects in marmosets.

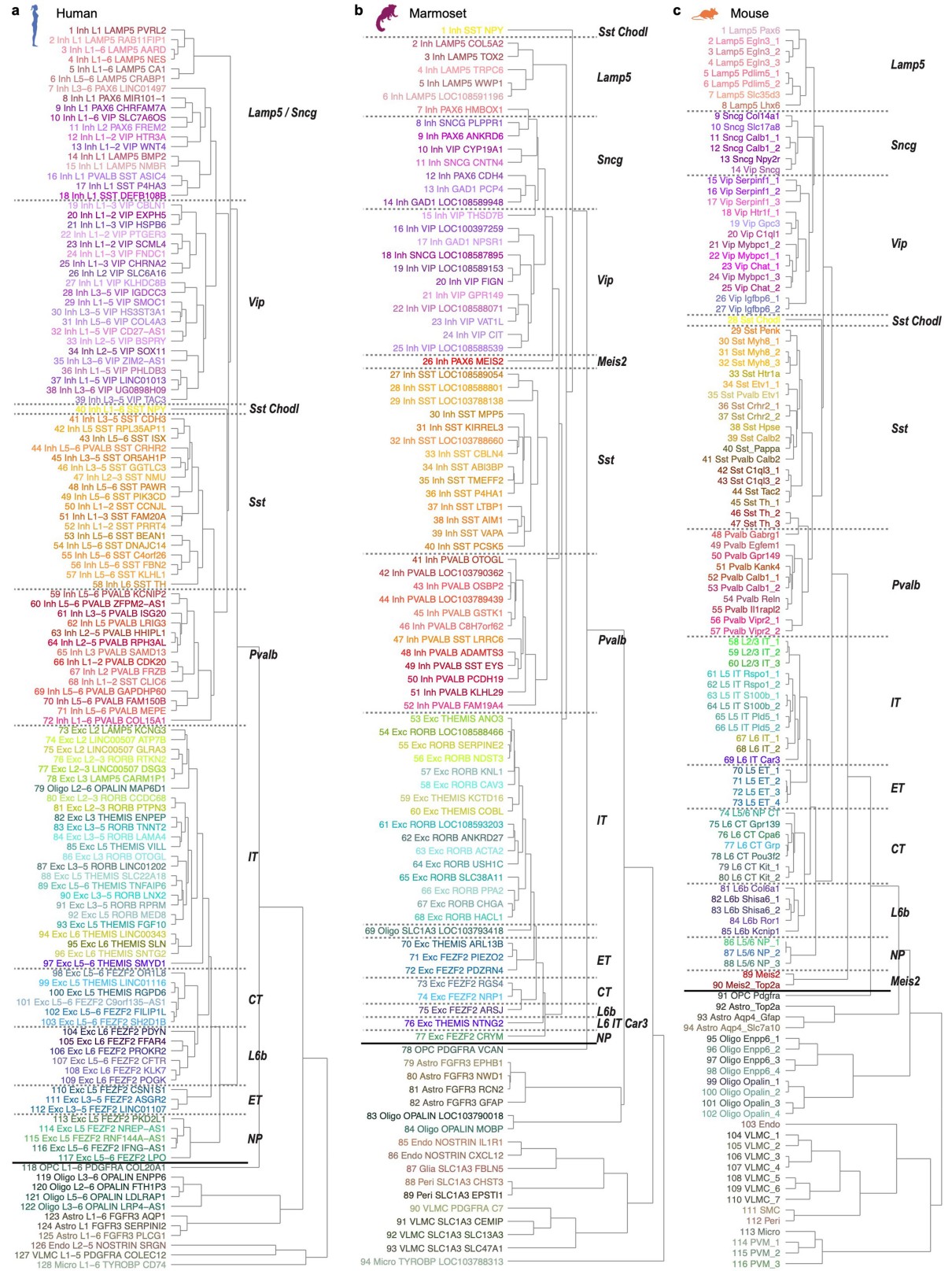

**Extended Data Fig. 2 | Taxonomies of M1 cell types in humans, marmosets and mice. a–c**, Taxonomies are reproduced from Fig. 1. Leaves are labelled with species-specific clusters, and branches are labelled with major subclasses of neuronal types. We defined 127 human clusters on the basis of Cv3 and SSv4 data, 94 marmoset clusters from Cv3 data, and 116 mouse clusters in a companion paper[5] by integrating 7 RNA datasets. These apparent differences in cellular diversity are likely to be driven by sampling depth, data quality and statistical criteria. For example, more non-neuronal nuclei were sampled in mice (58,098) and marmosets (21,189) than in humans (4,005), and more non-neuronal types were identified in those species.

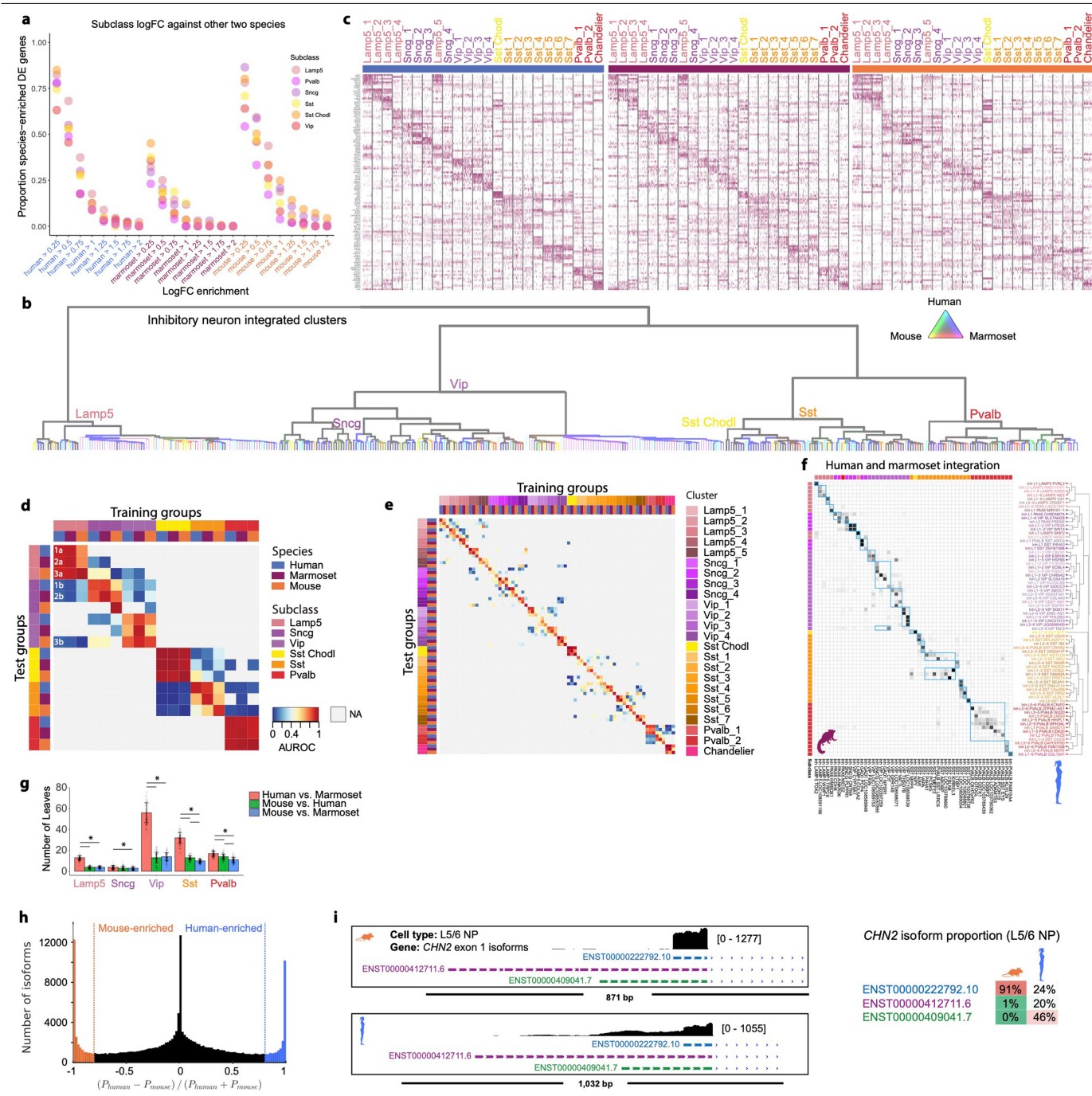

**Extended Data Fig. 3** | See next page for caption.

**Extended Data Fig. 3 | RNA-seq integration of GABAergic neurons across species. a**, Dot plot showing the proportion of species-enriched subclass marker genes (from Fig. 2c, d) that show log-transformed fold change (logFC) enrichment over the same subclass from the other two species. **b**, Dendrogram showing clusters of GABAergic (inhibitory) neurons from unsupervised clustering of integrated RNA-seq data from humans, marmosets and mice. The branch thickness indicates the relative number of nuclei, and the branch colour indicates species mixing (grey is well mixed). Major branches are labelled by subclass. The dendrogram in Fig. 2f was derived from this tree by pruning species-specific branches. **c**, Heat maps showing scaled expression of the top five marker genes for each GABAergic cross-species cluster, and five marker genes for *Lamp5* and *Sst* clusters. Initial genes were identified by performing a Wilcox test of every integrated cluster against all other GABAergic nuclei. Additional DEGs were identified for *Lamp5* and *Sst* cross-species clusters, by comparing one of the cross-species clusters with all other related nuclei (for example, Sst_1 against all other *Sst* clusters). **d**, **e**, Heat map showing 'one versus best MetaNeighbour' scores for GABAergic subclasses (**d**) and clusters (**e**). Each column shows the performance of a single training group across the three test datasets. AUROCs are computed between the two closest neighbours in the test dataset, where the closer neighbour will have the higher score, and all others are shown in grey (NA). For example, in **d** the first column contains results of training on human *Lamp5*, labelled with numbers to indicate test

datasets, where 1 is human, 2 is marmoset and 3 is mouse, and letters to indicate closest (a) and second-closest (b) neighbouring groups. Dark red three-by-three blocks along the diagonal indicate high transcriptomic similarity across all three species. **f**, Heat map showing cluster overlaps obtained from pairwise human–marmoset Seurat integration, indicating the proportion of within-species clusters that coalesce within integrated clusters. Columns and rows are ordered as in Fig. 2e, with cross-species consensus clusters indicated by blue boxes. The top and left colour bars indicate subclasses of within-species clusters. **g**, Bar plots quantifying the number of well mixed leaf nodes (mean ± s.d.; $n = 100$ subsamples) in dendrograms of pairwise species integrations from Fig. 2h. ANOVA tests for each subclass were followed by two-sided Tukey's HSD tests with Bonferroni correction for multiple comparisons; degrees of freedom = 297; *$P < 0.0001$. **h**, Histogram showing the relative difference in isoform genic proportion (*P*) between humans and mice for all subclass comparisons. All moderately to highly expressed isoforms were included (gene TPM greater than 10 in both species; isoform TPM greater than 10 and proportion greater than 0.2 in either species). Vertical lines indicate a more than ninefold change in mice or humans. **i**, Genome-browser tracks of RNA-seq (SSv4) reads in human and mouse L5/6 NP neurons at the *CHN2* locus for the three most common isoforms. The short isoform of *CHN2* is predominantly expressed in mouse neurons; longer isoforms are also expressed in human neurons.

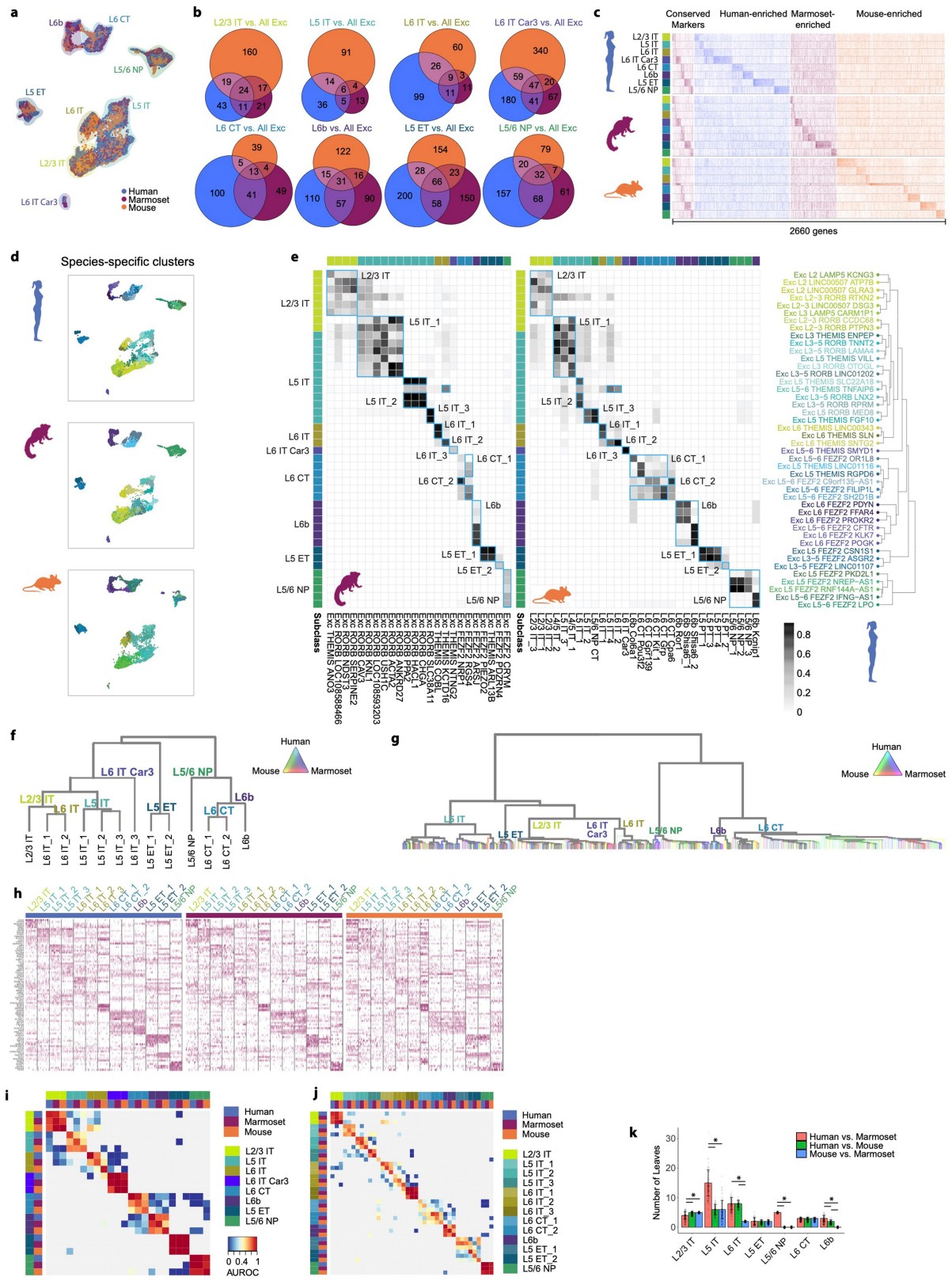

**Extended Data Fig. 4** | See next page for caption.

**Extended Data Fig. 4 | Homology of glutamatergic neurons across species.**
**a**, UMAP visualization of integrated snRNA-seq data from human, marmoset and mouse glutamatergic neurons. The highlighted colours indicate subclasses. **b**, Venn diagrams indicating the number of DEGs shared across species by subclass. DEGs were determined by ROC tests of a subclass against all other glutamatergic subclasses within a species. **c**, Heat map of conserved and species-enriched DEGs from **b**, ordered by subclass and species enrichment. The heat map shows expression, scaled by column, for up to 50 randomly sampled nuclei from each subclass for each species. **d**, UMAP visualization of integrated snRNA-seq data, with projected nuclei split by species. Colours indicate different within-species clusters. **e**, Cluster overlap heat map showing the proportion of nuclei in each pair of species clusters that are mixed in the cross-species integrated space. Cross-species consensus clusters are indicated by labelled blue boxes. The top and left axes indicate the subclass of a given within-species cluster by colour. The bottom axis indicates marmoset (left) and mouse (right) within-species clusters. The right axis shows the glutamatergic branch of the human dendrogram from Fig. 1a.
**f**, Dendrogram showing cross-species clusters of glutamatergic neurons, with branches coloured by species mixture (grey, well mixed). **g**, Unpruned dendrogram of clusters of glutamatergic neurons, from unsupervised clustering of integrated RNA-seq data. The branch thickness indicates the relative number of nuclei, and the branch colour indicates species mixing. Major branches are labelled by subclass. **h**, Heat maps showing scaled expression of marker genes for each glutamatergic cross-species cluster. The top five marker genes for each cross-species cluster are shown, with an additional five genes for L5 extratelencephalic, L5 intratelencephalic and L6 intratelencephalic neurons. Initial genes were identified by performing a Wilcox test of every integrated cluster against all other glutamatergic nuclei. Additional DEGs were identified for L5 extratelencephalic, L5 intratelencephalic and L6 intratelencephalic cross-species clusters, by comparing one of the cross-species clusters with all other related nuclei (for example, L5 IT_1 against all other L5 IT neurons). **i**, **j**, Heat map of 'one versus best MetaNeighbour' scores for glutamatergic subclasses (**i**) and clusters (**j**).
**k**, Bar plots quantifying the number of well mixed leaf nodes (mean ± s.d.; $n = 100$ subsamples) from unsupervised clustering of pairwise species integrations. ANOVA tests for each subclass were followed by two-sided Tukey's HSD tests with Bonferroni correction for multiple comparisons; $*P < 0.005$.

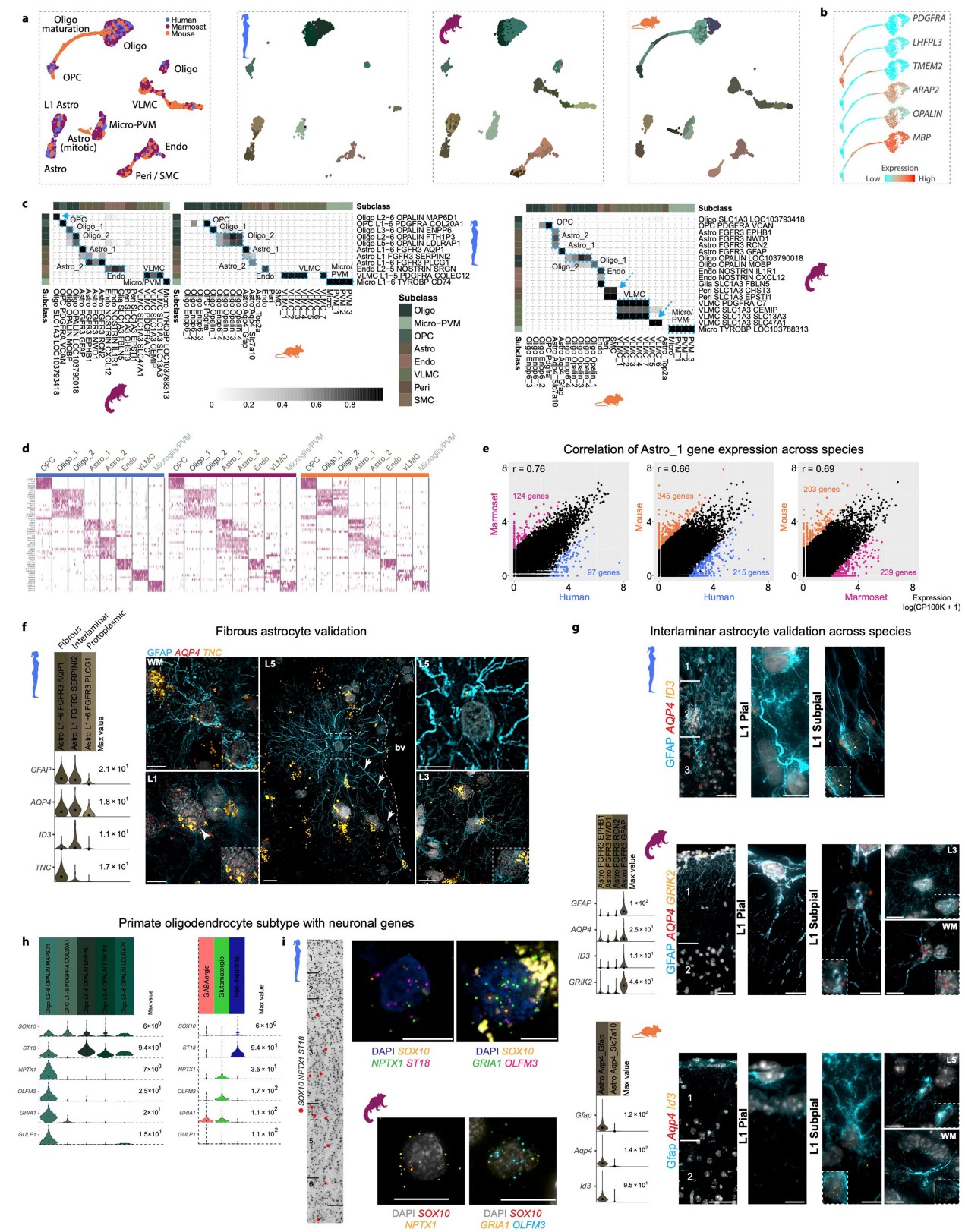

**Extended Data Fig. 5 |** See next page for caption.

**Extended Data Fig. 5 | Homology of non-neuronal cell types across species.**
**a**, UMAP plots of integrated RNA-seq data for non-neuronal nuclei, coloured by species and within-species clusters. Note that some cell types are present in only one or two species. **b**, UMAP plot showing maturation lineage between oligodendrocyte precursor cells (OPCs) and oligodendrocytes on the basis of reported marker genes[93]; this lineage was present in mice but not primates, probably reflecting the younger age of mouse tissues used. **c**, Heat maps showing the proportion of nuclei in each species-specific cluster that overlap in the integrated clusters. Blue boxes define homologous cell types that can be resolved across all three species. Arrows highlight clusters that overlap between two species and are not detected in the third species, owing to differences in the sampling depth of non-neuronal cells, the relative abundances of cell types between species, or evolutionary divergence. Pericytes, smooth muscle cells (SMCs) and some subtypes of vascular and leptomeningeal cells (VLMCs) were present in marmoset and mouse and not human datasets, although these cells are present in human cortex[94]. Mitotic astrocytes (Astro_Top2a) were present in mice only, and represented 0.1% of non-neuronal cells. **d**, Conserved marker genes from homologous cell types across species. **e**, Pairwise comparisons between species of log-transformed gene expression (counts per 100,000 transcripts) of the Astro_1 type. Coloured points correspond to significantly differentially expressed genes (FDR less than 0.01; log-transformed fold change greater than 2). r, Spearman correlation. **f**, Validation of fibrous astrocytes in situ. Violin plots show marker genes from clusters of human astrocytes that correspond to fibrous, interlaminar and protoplasmic types on the basis of in situ labelling of types. Left ISH images show fibrous astrocytes located in the white matter (WM, top), and a subset of L1 astrocytes (bottom) that express the Astro L1-6 *FGFR3 AQP1* marker gene *TNC*. The centre ISH image shows a putative varicose projection astrocyte located in cortical L5 adjacent to a blood vessel (bv) and extending long processes labelled with glial fibrillary acidic protein (GFAP; white arrows); this astrocyte does not express the marker gene *TNC*. The white dashed box

indicates the area shown at higher magnification in the top right panel. Likewise, the L3 protoplasmic astrocyte shown in the bottom right panel does not express *TNC*. Scale bars, 15 μm. **g**, Combined GFAP immunohistochemistry and RNAscope FISH for markers of L1 astrocytes in humans, mice and marmosets. In humans (top panels), pial and subpial interlaminar astrocytes are labelled with *AQP4* and *ID3* and extend long processes from L1 down to L3. In marmosets (centre panels), both pial and subpial L1 astrocytes express *AQP4* and *GRIK2* and extend GFAP-labelled processes through L1 that terminate before reaching L2. An image of a marmoset protoplasmic astrocyte located in L3 (top right) shows that this astrocyte type does not express the marker gene *GRIK2*. A subset of marmoset fibrous astrocytes located in the white matter (bottom right) express *GRIK2*, suggesting that fibrous and L1 astrocytes have a shared gene-expression signature, as also seen in humans[3]. L1 astrocytes in mice (bottom panels) consist of pial and subpial types that differ morphologically but are characterized by their expression of the genes *Aqp4* and *Id3*. Pial astrocytes in mice extend short GFAP-labelled processes that terminate within L1, whereas subpial astrocytes appear to extend processes predominantly towards the pial surface. Protoplasmic astrocytes (an example is shown in L5) do not express *Id3*, whereas fibrous astrocytes share expression of *Id3* with L1 astrocyte types. In each image, a higher magnification of the cell is shown in white dashed boxes to demonstrate RNAscope spots for each gene labelled. Scale bars, 20 μm. **h**, Violin plots showing marker genes from clusters of oligodendrocyte lineages in humans. Transcripts detected in the Oligo L2–6 *OPALIN MAP6D1* cluster include genes that are expressed almost exclusively in neuronal cells. **i**, Left, Inverted DAPI image showing a column of cortex labelled with markers of the human Oligo L2-6 *OPALIN MAP6D1* type. Red dots show cells triple labelled for *SOX10*, *NPTX1* and *ST18*. Top right, examples of cells labelled with combinations of marker genes specific for the human Oligo L2-6 *OPALIN MAP6D1* type. Bottom right, example of a marmoset cell labelled with the marker genes *OLIG2* and *NRXN3*. Scale bars, 20 μm.

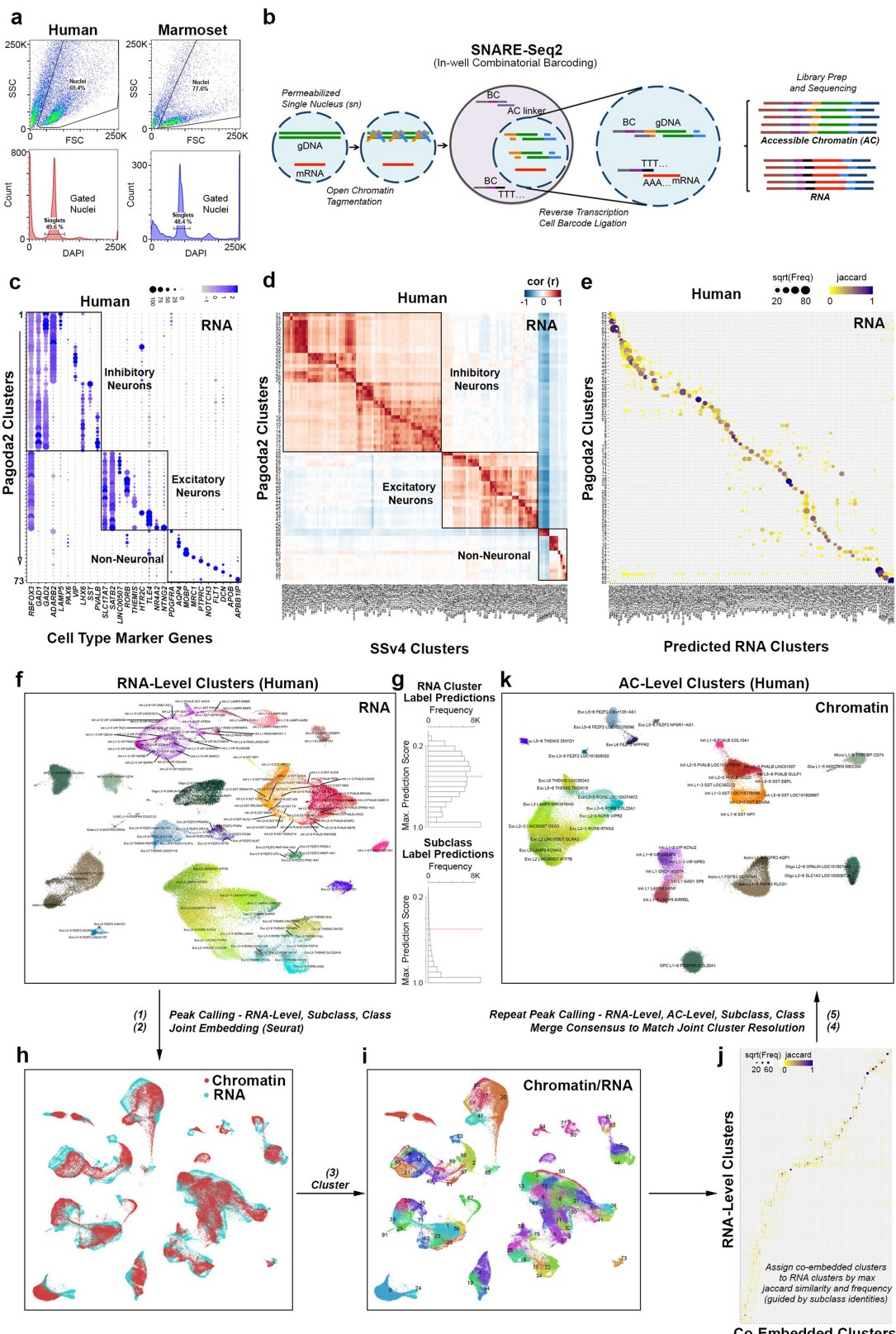

**Extended Data Fig. 6** | See next page for caption.

**Extended Data Fig. 6 | SNARE–seq2 transcriptomic profiling resolves M1 cell types. a**, **b**, FACS gating parameters used for sorting human and marmoset single nuclei (**a**) and for SNARE–seq2 (**b**), to generate libraries of RNA and accessible chromatin (AC) that have the same cell barcodes (BC). gDNA, guide DNA. **c**, Dot plot showing averaged values for the expression of marker genes (blue shading; log scale) and the proportion of nuclei with expression (black circles) for clusters identified from analysis of SNARE–seq2 RNA using Pagoda2. **d**, Correlation heat map of averaged scaled gene-expression values for Pagoda2 clusters against SSv4 clusters from the same M1 region. **e**, Jaccard similarity plot for cell barcodes grouped according to Pagoda2 clustering and compared against the predicted SSv4 clustering. **f**–**k**, Overview of cluster assignments at the level of accessible chromatin using RNA-defined clusters, indicating the five main steps of the process. **f**, RNA clusters visualized by UMAP on RNA expression data, which were used to independently call peaks from data on accessible chromatin. **g**, Histograms showing maximum prediction scores for RNA cluster (top) and subclass (bottom) labels from RNA data to corresponding accessibility data (Cicero gene activities). **h**, Peak regions called from barcode groupings at the level of RNA cluster, subclass and class were combined, and the corresponding peak by cell barcode matrix was used to predict gene-activity scores by using Cicero for integrative analyses of RNA and accessible chromatin. The UMAP shows joint embedding of RNA and imputed AC expression values using Seurat/Signac. **i**, UMAP showing clusters identified from the joint embedding (**h**). **j**, Jaccard similarity plot comparing cell barcodes grouped either according to RNA clustering or by joint clustering of RNA and accessible chromatin (**i**). RNA clusters were merged to best match the cluster resolution achieved from co-embedded clusters. Chromatin peak counts generated from peak calling on barcode groupings from RNA, accessible chromatin, subclass and class were used to generate a final peak by cell barcode matrix. **k**, Final clusters at the level of accessible chromatin visualized using UMAP.

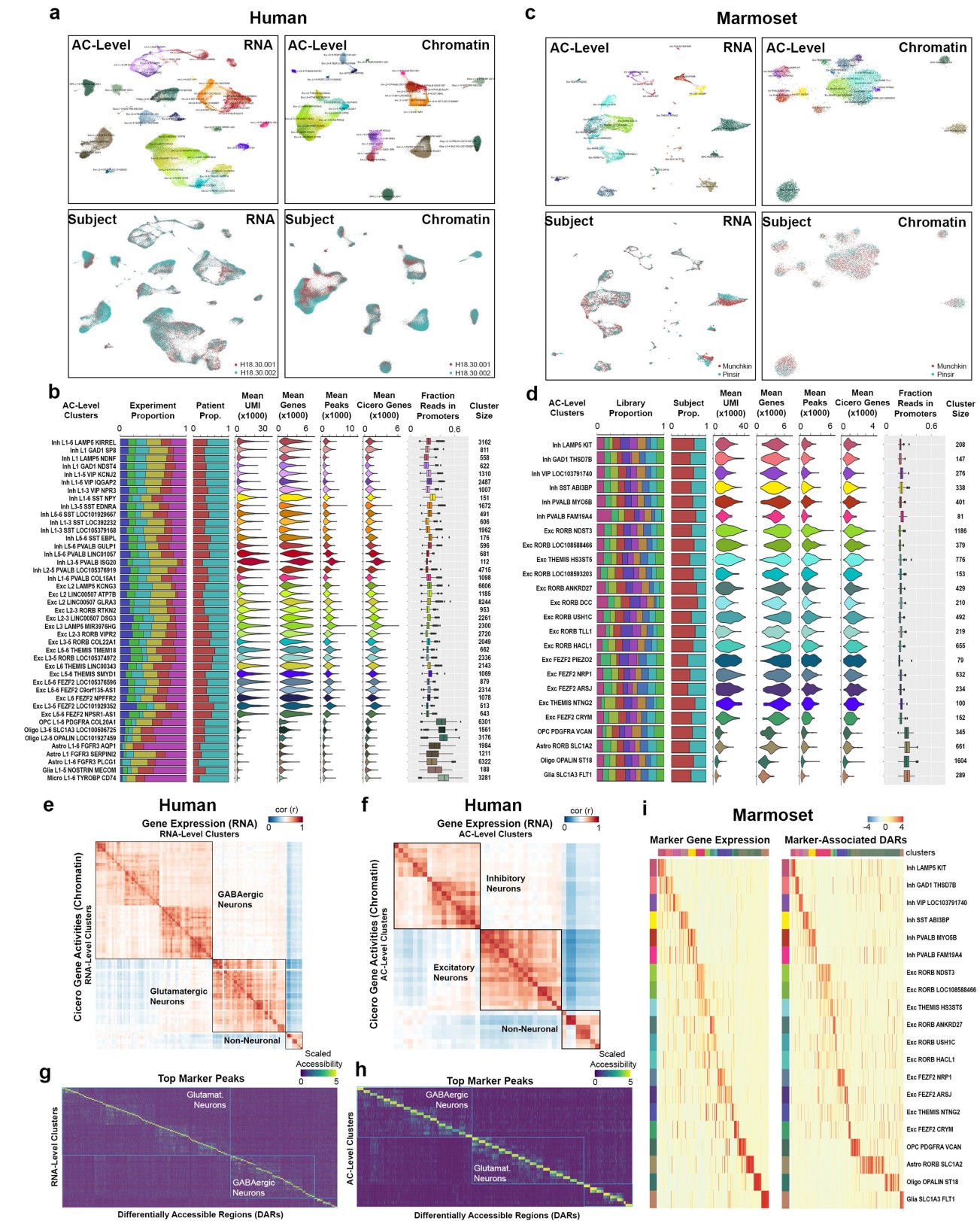

**Extended Data Fig. 7 |** See next page for caption.

**Extended Data Fig. 7 | SNARE–seq2 quality statistics. a**, UMAP plots showing human clusters at the level of accessible chromatin and corresponding participant identities for both RNA and chromatin embeddings. **b**, Bar, violin and box plots for human AC-level clusters, showing the proportion contributed by each experiment or patient, mean UMI and genes detected from the RNA data, the mean peaks and Cicero active genes detected from AC data, the fraction of reads found in promoters for AC data, and the number of nuclei making up each of the clusters. Box plots extend from 25th to 75th percentiles; central lines represent medians; and whiskers extend up to 1.5 times the interquartile interval. **c**, UMAP plots showing marmoset AC-level clusters and corresponding subject identities for both RNA and chromatin embeddings. **d**, Bar, violin and box plots for marmoset AC-level clusters, showing the proportion contributed by each library or subject, mean UMI and genes detected from the RNA data, the mean peaks and cicero active genes detected from AC data, the fraction of reads found in promoters for AC data, and the number of nuclei making up each of the clusters. Box plots extend from 25th to 75th percentiles; central lines represent medians; and whiskers extend up to 1.5 times the interquartile interval. **e**, **f**, Correlation heat maps of average scaled gene-expression values against average scaled Cicero gene activity values for RNA clusters (**e**) and AC-level clusters (**f**). **g**, **h**, Heat maps showing top averaged scaled chromatin accessibility values for DARs (Supplementary Table 14) identified for clusters at the level of RNA (**g**) and accessible chromatin (**f**). **i**, Heat maps showing the expression of marmoset AC-cluster markers and associated DARs, as shown for humans in Fig. 3b.

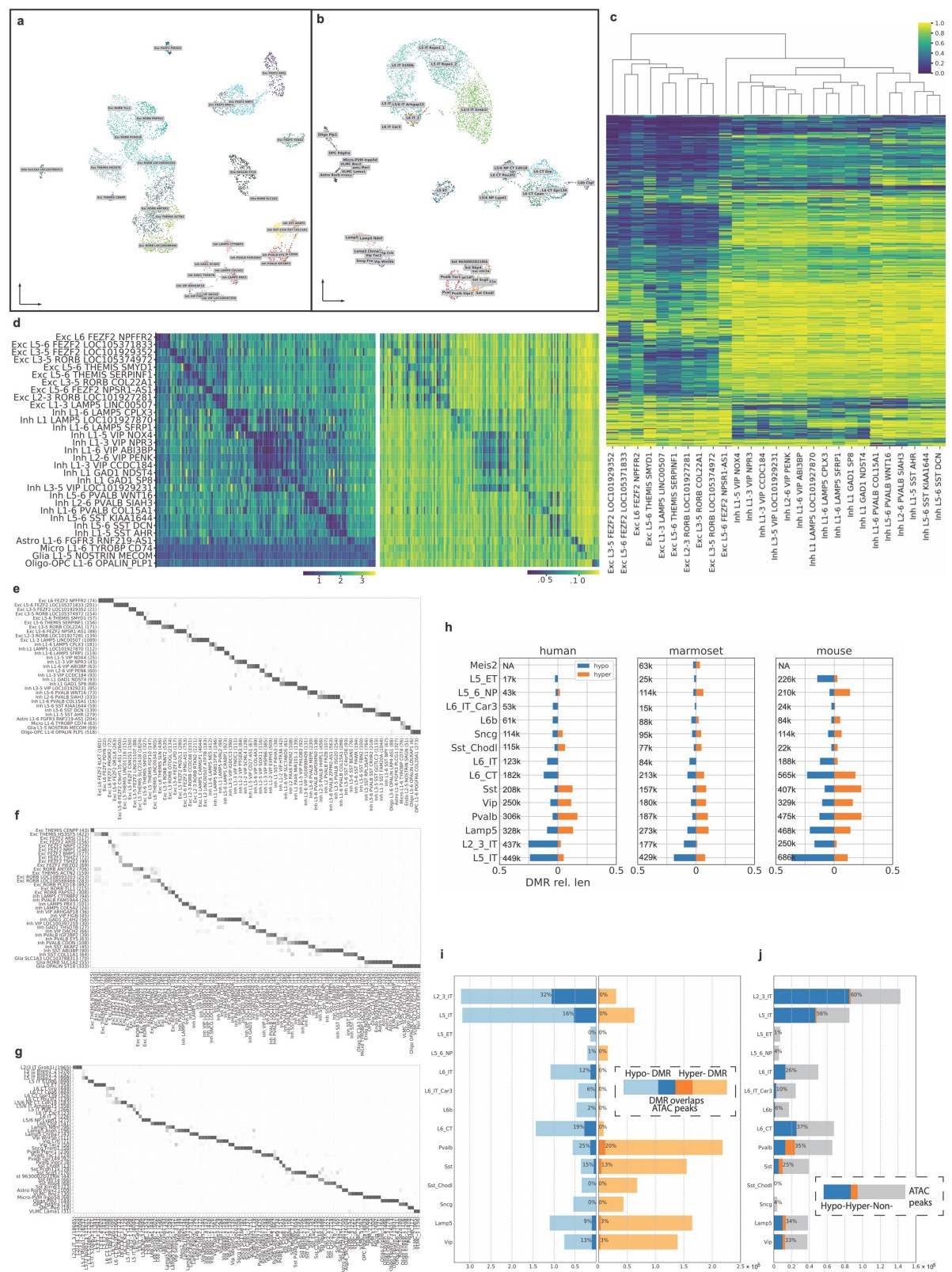

**Extended Data Fig. 8 | Cell types identified by DNA methylation and integration with RNA-seq data. a, b,** UMAP visualizations of marmoset (**a**) and mouse (**b**) data on DNA methylation (snmC-seq2) and cell clusters. **c,** Cell-type DMRs (mCG) across human neuronal clusters. Only those DMRs with at least 20 differentially methylated cytosine sites are shown. **d,** Hypomethylation of CG (left) and CH (right) in the gene bodies of cluster markers in humans. **e–g,** Mapping between DNAm-seq and RNA-seq clusters from humans (**e**), marmosets (**f**) and mice (**g**). The numbers of nuclei in each cluster are listed in parentheses. **h,** Barplots showing the relative lengths of hypomethylated and hypermethylated DMRs among cell subclasses across three species, normalized by genome-wide cytosine coverage (see Methods). The total numbers of DMRs for each subclass are listed (k, thousands). **i,** Numbers of hypomethylated and hypermethylated DMRs and overlap with chromatin accessible peaks in each subclass of human. **j,** Numbers of AC peaks and overlap with DMRs in each subclass in humans.

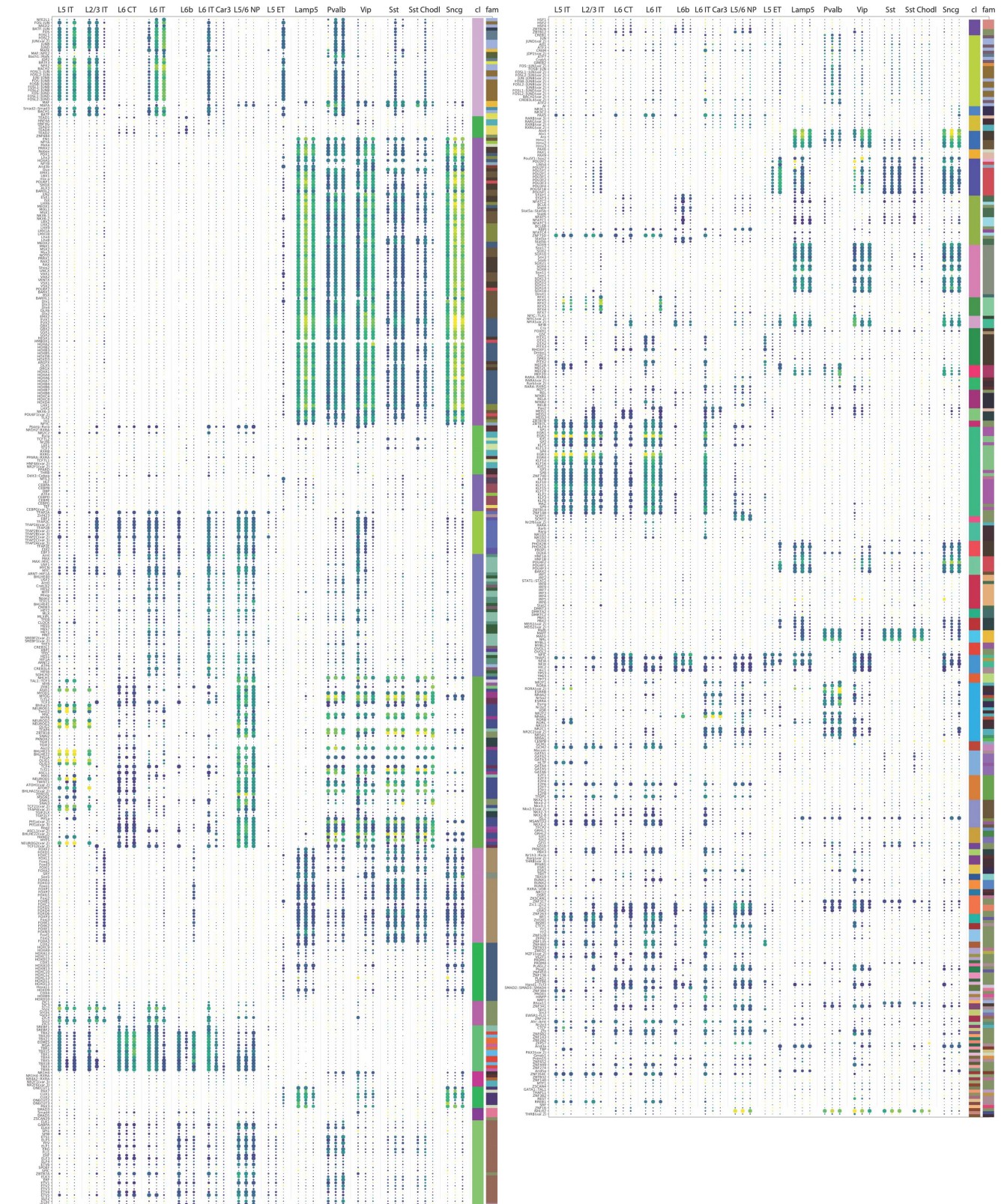

**Extended Data Fig. 9 | Analysis of TFBS enrichment on hypomethylated DMRs shows that gene regulation is distinct across subclasses and conserved across species.** Analyses of the enrichment of TFBS motifs were conducted using JASPAR's non-redundant core vertebrate transcription-factor motifs for neuronal subclasses in each species. Each subclass tri-column shows, from left to right, the results from humans, marmosets and mice. The size of a dot denotes the *P* value of the corresponding motif, while the colour denotes the fold change. The rightmost two columns show clusters of transcription factors (cl) identified from motif profiles and families of transcription factors (fam) identified from the structures of transcription factors as defined in the JASPAR database.

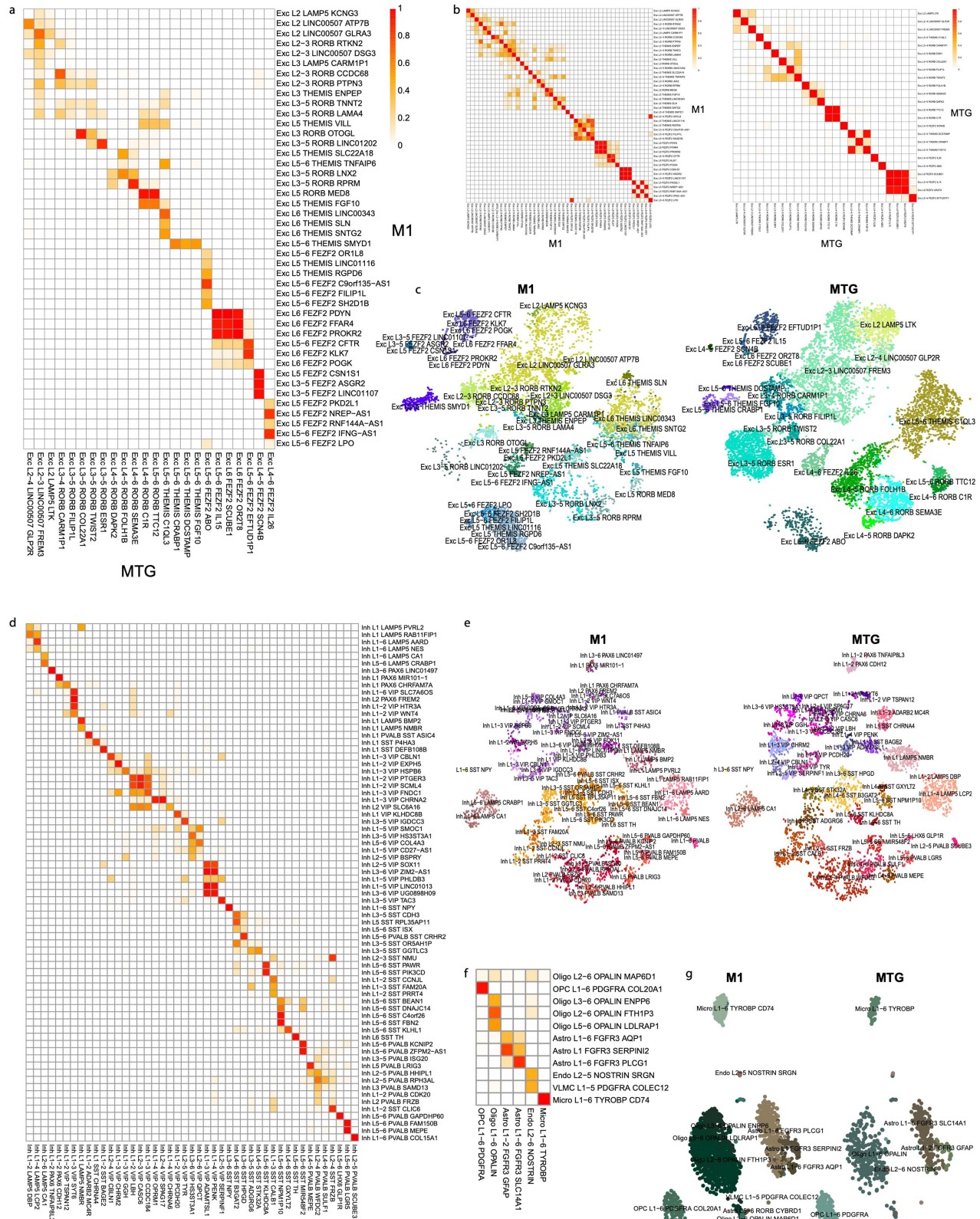

**Extended Data Fig. 10 | Homologies of cell types in human cortical areas based on RNA-seq integration. a**, Heat map showing the overlap of clusters of glutamatergic neurons between M1 and MTG. Interestingly, four MTG L2/3 intratelencephalic types (*LTK*, *GLP2R*, *FREM3* and *CARM1P1*) with distinct physiology and morphology[23] had less clear homology in M1, indicating more areal variation in supragranular neurons. **b**, Heat maps showing the overlap of clusters of glutamatergic neurons for M1 and MTG test datasets. Clusters were split in half, and the two datasets were integrated using the same analysis pipeline as for the M1 and MTG integration. Most clusters mapped correctly (along the diagonal) with some loss in resolution between closely related clusters (red blocks). **c**, *t*-SNE plots of integrated glutamatergic neurons labelled with M1 and MTG clusters. **d**–**g**, Heat maps of cluster overlaps and *t*-SNE plots of integrations for GABAergic neurons (**d**, **e**) and non-neuronal cells (**f**, **g**), as described in **a**–**c** for glutamatergic neurons.

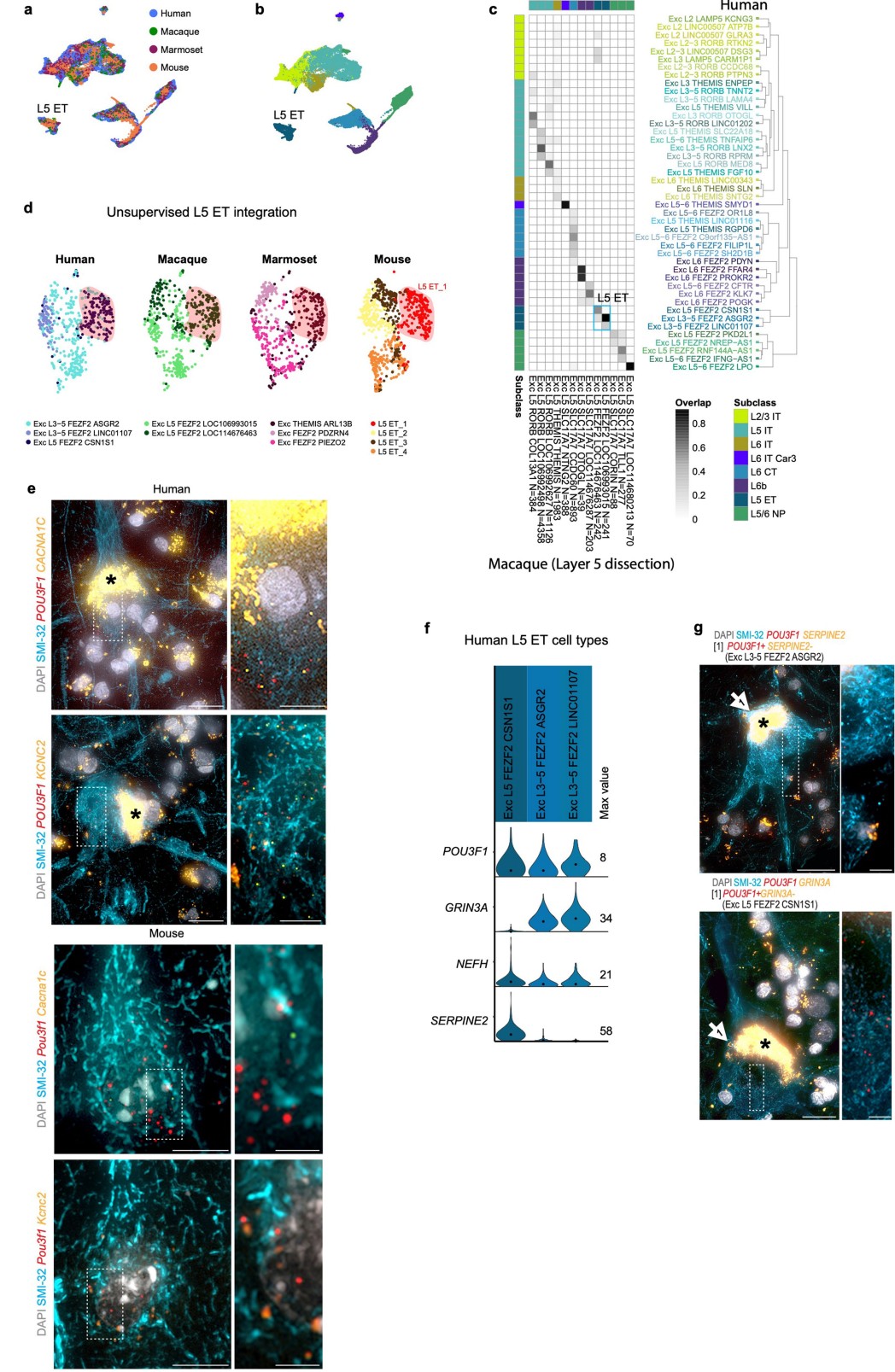

**Extended Data Fig. 11 | Cross-species alignment of L5 glutamatergic neurons, and conservation and divergence of transcriptomic properties. a**, **b**, UMAP visualizations of cross-species integration of snRNA-seq data for glutamatergic neurons isolated from humans, macaques (L5 dissection only), marmosets and mice. Colours indicate species (**a**) or cell subclass (**b**). **c**, Heat map of cluster overlaps, showing the proportion of nuclei from within-species clusters that are mixed within the same integrated clusters. Human clusters (rows) are ordered according to the dendrogram reproduced from Fig. 1a. Macaque clusters (columns) are ordered to align with human clusters. Colour bars at the top and left indicate subclasses of within-species clusters. The blue outline denotes the L5 extratelencephalic subclass. **d**, UMAP visualizations of cross-species integration of L5 extratelencephalic neurons. There is good correspondence across species to the mouse L5 ET_1 subtype that projects to medulla[5]. **e**, Examples of cells labelled by ISH and stained with anti-SMI-32 immunofluorescence in L5 of human and mouse M1. Cells are labelled with the extratelencephalic marker *POU3F1*/*Pou3f1* and the ion-channel genes *CACNA1C*/*Cacna1c* or *KCNC2*/*Kcnc2*. Consistent with snRNA-seq data, human L5 extratelencephalic M1 neurons appear to express higher levels of *CACNA1C* and *KCNC2* than do mouse L5 extratelencephalic M1 neurons. Scale bars, main images, 25 μm (humans), 15 μm (mice); insets, 10 μm (humans), 5 μm (mice). **f**, Violin plot showing the expression of marker genes for subtypes of human L5 extratelencephalic neurons. **g**, Two examples of cells with Betz morphology, labelled by ISH and stained with anti-SMI-32 immunofluorescence, in L5 of human M1 that correspond to the L5 extratelencephalic clusters Exc L3-5 *FEZF2 ASGR2* and EXC L5 *CSN1S1*. Insets show higher magnification of ISH-labelled transcripts in corresponding cells. Scale bars, 25 μm, insets 10 μm. Asterisks mark lipofuscin.

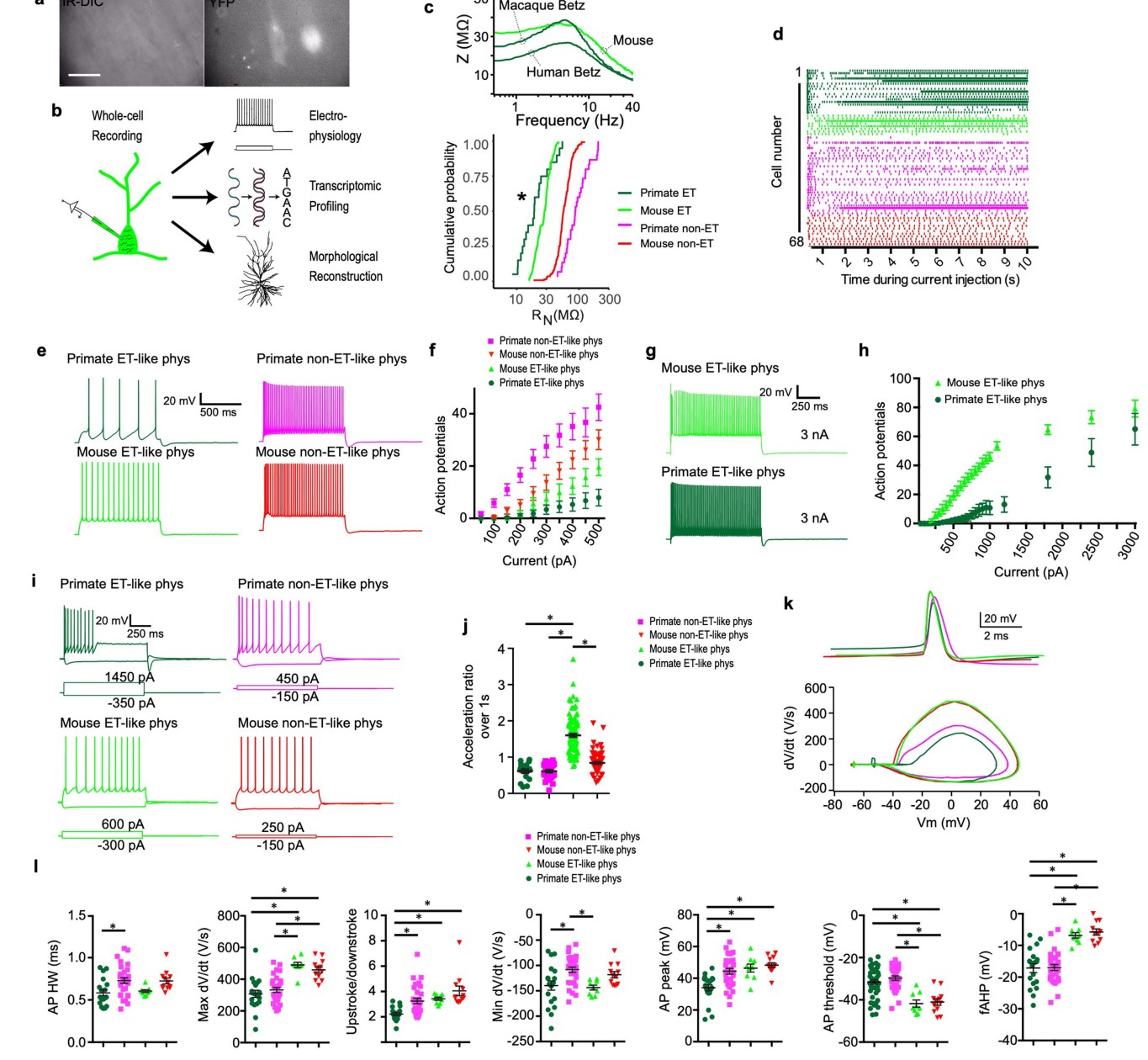

**Extended Data Fig. 12** | See next page for caption.

**Extended Data Fig. 12 | Differences in spike trains produced by L5 glutamatergic neurons and single spike properties across species.**
**a**, Example IR-DIC (left) and fluorescence (right) images obtained from a macaque organotypic slice culture. Note the inability to visualize the fluorescently labelled neurons in IR-DIC because of dense myelination. All human and macaque recordings were from labelled neurons. Scale bar, 50 μm. **b**, Patch–seq involves the collection of morphological, physiological and transcriptomic data from the same neuron. Following electrophysiological recording and cell filling with biocytin via whole-cell patch clamp, the contents of the cell are aspirated and processed for RNA sequencing. This permits a transcriptomic cell type to be pinned to the physiologically probed neuron. **c**, Top, example ZAP profiles for the neurons shown in Fig. 6f–h. Bottom, cumulative probability distribution showing input resistance for physiologically defined L5 neuron types from primates versus mice. *$P$ = 0.0064, Kolmogorov–Smirnov test between mouse and primate extratelencephalic neurons. **d**, Raster plot of spike times during 1-s epochs of a 10-s injection of DC current, with colour coding as in **c**. Primate extratelencephalic neurons (pooled data from humans and macaques, $n$ = 20) displayed a distinctive decrease followed by a pronounced increase in firing rate over the course of the current injection, whereas other neuron types did not (primate intratelencephalic neurons, $n$ = 30; mouse extratelencephalic neurons, $n$ = 8; mouse intratelencephalic neurons, $n$ = 12). Notably, a similar biphasic-firing pattern is observed in macaque corticospinal neurons in vivo during prolonged motor movements[95,96], suggesting that the firing pattern of these neurons during behaviour is intimately tied to their intrinsic membrane properties. The acceleration in spike times of rodent extratelencephalic neurons has been attributed to the expression of Kv1-containing voltage-gated K⁺ channels, encoded by genes such as the conserved extratelencephalic gene *KCNA1* (ref. [41]). **e**, Example voltage responses to a 1-s, 500-pA current injection. **f**, Action potentials (mean ± s.e.m.) as a function of the amplitude of injected current. Primate extratelencephalic neurons display the shallowest relationship between action potential and injected current, perhaps partially because of their exceptionally low input resistance (primate extratelencephalic neurons, $n$ = 20; primate intratelencephalic neurons, $n$ = 30; mouse extratelencephalic neurons, $n$ = 9; mouse intratelencephalic neurons, $n$ = 12). **g**, Voltage responses to a current injection with a 1-s, 3-nA step. **h**, Action potentials (mean ± s.e.m.) as a function of injected current for a subset of experiments in which the amplitude of injected current was increased incrementally to 3 nA. Although both mouse ($n$ = 9) and primate ($n$ = 10) extratelencephalic neurons could sustain high firing rates, primate neurons required 3 nA of current over 1 s to reach similar average firing rates as mouse extratelencephalic neurons. **i**, Example voltage responses to current injections with 1-s depolarizing steps. The amplitude of the current injection was adjusted to produce roughly ten spikes. Also shown are voltage responses to a hyperpolarizing current injection. **j**, The firing rate (mean ± s.e.m.) of primate extratelencephalic ($n$ = 18), primate intratelencephalic ($n$ = 30) and mouse intratelencephalic ($n$ = 86) neurons decreased during the 1-s step current injection, whereas the firing rate of mouse extratelencephalic neurons increased ($n$ = 110). The acceleration ratio is the ratio of the second to the last interspike interval. *$P$ < 0.05, Bonferroni-corrected two-sided $t$-test. **k**, Example single action potentials (above) and phase plane plots (below). **l**, Various features of action potentials (mean ± s.e.m.) are plotted as a function of cell type (primate extratelencphalic, $n$ = 20; primate intratelencephalic, $n$ = 30; mouse extratelencephalic, $n$ = 9; mouse intratelencephalic, $n$ = 12). Notably, action potentials in primate extratelencephalic neurons were reminiscent of fast spiking interneurons, in that they were shorter and more symmetrical compared with action potentials in other neuron types/species. Intriguingly, the K⁺-channel subunits $K_v3.1$ and $K_v3.2$, which are implicated in fast-spiking physiology[97], are encoded by highly expressed genes (*KCNC1* and *KCNC2*) in primate extratelencephalic neurons (Fig. 6c). *$P$ < 0.05, Bonferroni-corrected two-sided $t$-test.

**Extended Data Table 1 | Summary of human donors of postmortem tissue**

| Specimen ID | Age | Sex | Race | Cause of Death | PMI (hr) | Tissue RIN | Hemi-sphere Sampled | Data Type |
|---|---|---|---|---|---|---|---|---|
| H200.1023 | 43 | F | Iranian descent | Mitral valve prolapse | 18.5 | 7.4 ± 0.7 | L | SSv4 |
| H200.1025 | 50 | M | Caucasian | Cardiovascular | 24.5 | 7.6 ± 1.0 | L | SSv4 |
| H200.1030 | 54 | M | Caucasian | Cardiovascular | 25 | 7.7 ± 0.8 | L | SSv4 |
| H18.30.001 | 60 | F | Unknown | Car accident | 18 | 7.9 ± 2.5 | R | SSv4, Cv3, SNARE-seq2, snmC-seq2 |
| H18.30.002 | 50 | M | Unknown | Cardiovascular | 10 | 8.2 ± 0.4 | R | SSv4, Cv3, SNARE-seq2, snmC-seq2 |

PMI, postmorten interval; RIN, RNA integrity number. Data types: SSv4, SMART-Seqv4; Cv3, 10×Genomics Chromium single-cell 3' kit v3; SNARE–seq2, single-nucleus chromatin accessibility and mRNA expression sequencing; snmC-seq2, single-nucleus methyl cytosine sequencing.

**Extended Data Table 2 | Summary of tissue specimens obtained from postmortem of non-human primates**

| Species | Specimen ID | Age (years) | Sex | Body mass (kg) | Data Type |
|---|---|---|---|---|---|
| Common marmoset (*Callithrix jacchus*) | bi005 | 2.3 | M | | Cv3 |
| | bi006 | 3.1 | F | | Cv3 |
| | bi003 | 1.9 | M | | FISH |
| Pig-tailed macaque (*Macaca nemestrina*) | Q19.26.002 | 12.3 | M | 15.8 | Cv3 |
| | Q19.26.003 | 3.9 | F | 4.2 | Cv3 |
| | Q19.26.008 | 17.2 | M | 16.3 | Cv3 |

Data types: Cv3, 10× Genomics Chromium single-cell 3' kit v3; FISH, ACD Bio multiplex fluorescent in situ hybridization.

# Reporting Summary

Nature Research wishes to improve the reproducibility of the work that we publish. This form provides structure for consistency and transparency in reporting. For further information on Nature Research policies, see Authors & Referees and the Editorial Policy Checklist.

## Statistics

For all statistical analyses, confirm that the following items are present in the figure legend, table legend, main text, or Methods section.

| n/a | Confirmed | |
|---|---|---|
| ☐ | ☒ | The exact sample size (*n*) for each experimental group/condition, given as a discrete number and unit of measurement |
| ☐ | ☒ | A statement on whether measurements were taken from distinct samples or whether the same sample was measured repeatedly |
| ☐ | ☒ | The statistical test(s) used AND whether they are one- or two-sided *Only common tests should be described solely by name; describe more complex techniques in the Methods section.* |
| ☒ | ☐ | A description of all covariates tested |
| ☐ | ☒ | A description of any assumptions or corrections, such as tests of normality and adjustment for multiple comparisons |
| ☐ | ☒ | A full description of the statistical parameters including central tendency (e.g. means) or other basic estimates (e.g. regression coefficient) AND variation (e.g. standard deviation) or associated estimates of uncertainty (e.g. confidence intervals) |
| ☐ | ☒ | For null hypothesis testing, the test statistic (e.g. *F*, *t*, *r*) with confidence intervals, effect sizes, degrees of freedom and *P* value noted *Give P values as exact values whenever suitable.* |
| ☒ | ☐ | For Bayesian analysis, information on the choice of priors and Markov chain Monte Carlo settings |
| ☒ | ☐ | For hierarchical and complex designs, identification of the appropriate level for tests and full reporting of outcomes |
| ☐ | ☒ | Estimates of effect sizes (e.g. Cohen's *d*, Pearson's *r*), indicating how they were calculated |

*Our web collection on statistics for biologists contains articles on many of the points above.*

## Software and code

Policy information about availability of computer code

| | |
|---|---|
| Data collection | BD Diva software v8.0, Nikon NIS-Elements Advanced Research imaging software v4.20, SoftMax Pro v6.5; VWorks v11.3.0.1195 and v13.1.0.1366; Hamilton Run Time Control v4.4.0.7740; Fragment Analyzer v1.2.0.11; Mantis Control Software v3.9.7.19; 10x Chromium v3 and Illumina MiSeq, HiSeq 2500, and Novaseq 6000 instrument control software. Physiology data acquisition software was either MIES (https://github.com/AllenInstitute/MIES/) or custom software written in Igor Pro. |
| Data analysis | Smart-seq v4 paired-end reads were clipped using ea-utils, then mapped using Spliced Transcripts Alignment to a Reference (STAR v2.7.3a). Reads were quantified using the R package, GenomicAlignments v1.18.0. For 10x Cv3 datasets, gene expression was quantified using 10x Cell Ranger v3 (https://support.10xgenomics.com/single-cell-gene-expression/software/downloads/latest). The clustering pipeline is implemented in the R package, scrattch.hicat v0.0.22 (https://github.com/AllenInstitute/scrattch.hicat). Custom R code written for clustering, and marker gene analysis and using open source R packages is available from https://github.com/AllenInstitute/ BICCN_M1_Evo. This includes R packages fastICA v1.2-1, limma v3.38.3, MetaNeighbor v1.9.1 (https://github.com/gillislab/ MetaNeighbor), scrattch.io v0.1.0, ggplot2 v3.3.2, Pagoda2 v0.1.0 (https://github.com/hms-dbmi/pagoda2), corrplot v0.84 (https:// github.com/taiyun/corrplot), igraph v1.2.6, Seurat v3.1.1 (https://satijalab.org/seurat/), eulerr v6.0.0, UpSetR v1.4.0, RSEM v1.3.3. Python packages DoubletDetection v2.5 and NSforest v2.1 (https://github.com/JCVenterInstitute/NSForest). Cross-species tree merging algorithm available at https://github.com/huqiwen0313/speciesTree. DNA-methylation and RNA-seq integration used custom code at https://github.com/lhqing/cemba_data.

ATAC-seq data analysis using R packages SnapATAC v2 (https://github.com/r3fang/SnapATAC), DropletUtils v1.6.1, Cicero v1.2.0 (https:// cole-trapnell-lab.github.io/cicero-release/), chromVAR v1.8.0 (https://greenleaflab.github.io/chromVAR), chromfunks v0.3.0 (https:// github.com/yanwu2014/chromfunks), SWNE v0.5.7 (https://github.com/yanwu2014/swne), Signac v0.1.4 (https://satijalab.org), pROC v1.16.2, GenomicRanges v1.38.0, Gviz v1.30.3, and EdgeR v3.28.1. Python 3.70 packages Bowtie v1.1.0, samtools v1.9, Phenograph v1.5.2, Snaptools v1.4.7, MACS2 v2.1.2 (https://github.com/taoliu/MACS), and deepTools v3.4.2.

SnmC-seq2-seq data analysis using Python software scanpy v1.4.4, scikit-learn v0.21.3, imblearn v0.0, Scanorama v1.0, methylpy v1.4.0 (https://github.com/yupenghe/methylpy). |

Genome browser tracks were generated using the Integrative Genomics Viewer (IGV v2.7.0). Neuron morphology reconstruction used ZEN 2012 SP2 software and Vaa3D v3.475. FIJI distribution of ImageJ v1.52p, GraphPad Prism v7.04.

For manuscripts utilizing custom algorithms or software that are central to the research but not yet described in published literature, software must be made available to editors/reviewers. We strongly encourage code deposition in a community repository (e.g. GitHub). See the Nature Research guidelines for submitting code & software for further information.

## Data

Policy information about availability of data

All manuscripts must include a data availability statement. This statement should provide the following information, where applicable:

- Accession codes, unique identifiers, or web links for publicly available datasets
- A list of figures that have associated raw data
- A description of any restrictions on data availability

Raw sequence data are available for download from the Neuroscience Multi-omics Archive (https://nemoarchive.org/) under accession number 'dat-ek5dbmu' and the Brain Cell Data Center (https://biccn.org/data). Visualization and analysis tools are available at NeMO Analytics (Individual species: https://nemoanalytics.org// index.html?layout_id=ac9863bf; Integrated species: https://nemoanalytics.org//index.html?layout_id=34603c2b) and Cytosplore Viewer (https:// viewer.cytosplore.org/). These tools allow users to compare cross-species datasets and consensus clusters via genome and cell browsers and calculate differential expression within and among species. Subclass level methylome tracks can be found at http://neomorph.salk.edu/aj2/pages/cross-species-M1/.  A semantic representation of the cell types defined through these studies is available in the provisional Cell Ontology (https://bioportal.bioontology.org/ontologies/PCL; Supplementary Table 1).

The following publicly available datasets were used for analysis: Jaspar motifs database (JASPAR2020, all vertebrate, http://jaspar.genereg.net/matrix-clusters/), HUGO Gene Nomenclature Committee (HGNC) at the European Bioinformatics Institute (https://www.genenames.org; downloaded January 2020), Synaptic Gene Ontology (SynGO; downloaded February 2020), and orthologous genes across species from NCBI Homologene (downloaded November 2019). Macaque reconstructions were from source data available in Neuromorpho (chandelier cell NeuroMorpho.org ID: NMO_01873, basket cell NeuroMorpho.org ID: NMO_01851). Mouse ATAC-seq available from https://assets.nemoarchive.org/dat-7qjdj84. MTG human SMARTseq v4 data (https://portal.brain-map.org/atlases-and-data/rnaseq/human-mtg-smart-seq, https://assets.nemoarchive.org/dat-swzf4kc). ENCODE blacklist regions (http://mitra.stanford.edu/kundaje/akundaje/ release/blacklists/hg38-human/hg38.blacklist.bed.gz)

# Field-specific reporting

Please select the one below that is the best fit for your research. If you are not sure, read the appropriate sections before making your selection.

☒ Life sciences  ☐ Behavioural & social sciences  ☐ Ecological, evolutionary & environmental sciences

For a reference copy of the document with all sections, see nature.com/documents/nr-reporting-summary-flat.pdf

# Life sciences study design

All studies must disclose on these points even when the disclosure is negative.

| Sample size | Sample size was not pre-determined. For RNA-seq, snmC-seq2, SNARE-seq2, snATAC-seq, and mFISH, single nuclei were isolated from postmortem brains of human (n = 5), macaque (n = 3), marmoset (n = 3), and mouse (n = 12-24). For human and marmoset, this allowed us to collect nuclei from high quality specimens that met stringent quality control metrics while also confirming that transcriptomic and epigenomic clusters were consistent between donors and not driven by technical artifacts.<br><br>For mice, sample size (number of animals) was determined by the experimental requirements for collection of sufficient tissue for each assay. In no case were differences between individual animals or batches similar in magnitude to the reported cell type differences. The number of cells collected was determined by specific limitations of each data modality. |
|---|---|
| Data exclusions | Low-quality nuclei were included for analysis if they met the following pre-established quality control (QC) thresholds.<br><br>Human RNA-seq (SMART-seq v4):<br>> 30% cDNA longer than 400 base pairs<br>> 500,000 reads aligned to exonic or intronic sequence<br>> 40% of total reads aligned<br>> 50% unique reads<br>> 0.7 TA nucleotide ratio<br><br>Human and Macaque RNA-seq (10x v3):<br>> 500 (non-neuronal nuclei) or > 1000 (neuronal nuclei) genes detected<br>< 0.3 doublet score<br><br>Marmoset RNA-seq (10x v3):<br>Cell barcodes were filtered to distinguish true nuclei barcodes from empty beads and PCR artifacts by assessing proportions of ribosomal and mitochondrial reads, ratio of intronic/exonic reads (> 50% of intronic reads), library size (> 1000 UMIs) and sequencing efficiency (true cell barcodes have higher reads/UMI). |

Mouse RNA-seq (SMART-seq v4 and 10x v3):
< 100,000 total reads, < 1,000 detected genes (CPM > 0), < 75%  of reads aligned to genome, or CG dinucleotide odds ratio > 0.5. Cells were classified into broad classes of excitatory, inhibitory, and non-neuronal based on known markers, and cells with ambiguous identities were removed as doublets.

snmC-seq2:
1) mCCC rate < 0.03. mCCC rate reliably estimates the upper bound of bisulfite non-conversion rate 5; 2) overall mCG rate > 0.5; 3) overall mCH rate < 0.2; 4) total final reads > 500,000; and 5) bismark mapping rate > 0.5.

SNARE-seq2:
RNA quality filtering. Empty barcodes were removed using the emptyDrops() function of DropletUtils 80, mitochondrial transcripts were removed, doublets were identified using the DoubletDetection software 81 and removed. All samples were combined across experiments within species and cell barcodes having greater than 200 and less than 7500 genes detected were kept for downstream analyses. To further remove low quality datasets, a gene UMI ratio filter (gene.vs.molecule.cell.filter) was applied using Pagoda2 (https://github.com/hms-dbmi/pagoda2). AC quality filtering. Cell barcodes were included if they showed greater than 1000 read fragments and 500 UMI. Read fragments were then binned to 5000 bp windows of the genome and only cell barcodes showing the fraction of binned reads within promoters greater than 10% (15% for marmoset) and less than 80% were kept.

Patch-seq:
Patch-seq nuclei were mapped to each species glutamatergic reference cell types and were retained for analysis if they mapped with > 85% confidence to a cluster in the L5 ET subclass.

| | |
|---|---|
| Replication | Flow cytometry data were reproducible across human tissue specimens from the 5 donors used in the study and across different nuclei isolations from individual tissue donors.

RNA-seq: Clustering reproducibility was measured by performing clustering analysis 100 times using a randomly-selected 80% of nuclei.

snmC-seq2: Leiden clustering resolution parameter was selected by three criteria: 1. The portion of outliers < 0.05 in the final consensus clustering version. 2. The ultimate prediction model accuracy > 0.95. 3. The average cell per cluster ≥ 30, which controls the cluster size to reach the minimum coverage required for further epigenome analysis such as DMR calls. All three criteria prevented the over-splitting of the clusters.

SNARE-seq2: Clustering of RNA data using the Pagoda2 package was highly similar to results from mapping to RNA-seq clusters using a centroid-based classifier.

For in situ hybridization and immunohistochemistry experiments, the number of times an experiment was repeated with similar results is listed in relevant figure legends. In general, experiments using human tissues were repeated on at least 2 independent donor tissues. |
| Randomization | All species specimens were controls and were therefore allocated into the same experimental group. Randomization was not used. |
| Blinding | Human specimens were de-identified and assigned a unique numerical code. Researchers responsible for data generation and analysis were not blinded and had access to basic information about donors (age, sex, ethnicity), as well as the unique numerical code assigned to each donor.

For experiments other than those involving human specimens, similar donor information was available to researchers involved in data generation and analysis.

Blinding was not relevant to these experiments because species information was necessary for sequence alignment to correct reference genomes, integration pipelines, and was a primary analytical endpoint. Similarly, donor information was necessary for study design and cluster curation steps during analysis. |

# Reporting for specific materials, systems and methods

We require information from authors about some types of materials, experimental systems and methods used in many studies. Here, indicate whether each material, system or method listed is relevant to your study. If you are not sure if a list item applies to your research, read the appropriate section before selecting a response.

## Materials & experimental systems

| n/a | Involved in the study |
|---|---|
| ☐ | ☒ Antibodies |
| ☐ | ☒ Eukaryotic cell lines |
| ☒ | ☐ Palaeontology |
| ☐ | ☒ Animals and other organisms |
| ☐ | ☒ Human research participants |
| ☒ | ☐ Clinical data |

## Methods

| n/a | Involved in the study |
|---|---|
| ☒ | ☐ ChIP-seq |
| ☐ | ☒ Flow cytometry |
| ☒ | ☐ MRI-based neuroimaging |

# Antibodies

| Antibodies used | 1. Mouse anti-NeuN-PE conjugated, EMD Millipore, Milli-Mark, clone A60, #FCMAB317PE, 1:500<br>2. Mouse anti-GFAP, EMD Millipore, #MAB360, clone GA5, 1:500<br>3. Mouse anti-neurofilament-H, nonphosphorylated (NF-H, clone SMI 32), Biolegend, #801701, 1:250<br>4. Goat anti-mouse IgG(H+L) Alexa Fluor 568 conjugate, ThermoFisher Scientific, #A-11004, 1:500<br>5. Goat anti-mouse IgG(H+L) Alexa Fluor 594 conjugate, ThermoFisher Scientific, #A-11005, 1:500<br>6. Goat anti-mouse IgG(H+L) Alexa Fluor 647 conjugate, ThermoFisher Scientific, #A-21235, 1:500<br>7. Mouse IgG1,k PE Isotype control, clone MOPC-21,#555749, BD Pharmingen, 1:250 |
|---|---|
| Validation | 1. mouse anti-NeuN-PE conjugated EMD Millipore, Milli-Mark, clone A60, #FCMAB317PE: From the manufacturer's website: This Milli-Mark Anti-NeuN-PE Antibody, clone A60 is validated for use in flow cytometry for the detection of NeuN. Quality is evaluated by flow cytometry using U251 cells. The immunogen is purified cell nuclei from mouse brain. Species reactivity – human.<br><br>2. Mouse anti-GFAP, EMD Millipore, #MAB360, clone GA5, 1:500: routinely evaluated by Western Blot on Mouse brain lysates.<br><br>3. Mouse anti-neurofilament-H, nonphosphorylated (NF-H, clone SMI 32), Biolegend, 1:250: From the manufacturer's website: Each lot of this antibody is quality control tested by formalin-fixed paraffin-embedded immunohistochemical staining. Species reactivity - Human, Mouse, Rat, Other mammalian. This antibody reacts with a nonphosphorylated epitope in neurofilament H of most mammalian species. The manufacturer provides IHC, IF, and western blot validation data for the antibody on their website. For immunohistochemistry, a concentrationcrange of 1.0 - 5.0 μg/ml is suggested.<br><br>4. Goat anti-mouse IgG(H+L) Alexa Fluor 568 conjugate, ThermoFisher Scientific, #A-11004, 1:500: each lot of antibody is quality control tested using immunocytochemistry.<br><br>5. Goat anti-mouse IgG(H+L) Alexa Fluor 647 conjugate, ThermoFisher Scientific, #A-21235, 1:500: each lot of antibody is quality control tested using immunocytochemistry.<br><br>6. Mouse IgG1,k PE Isotype control, clone MOPC-21,#555749, BD Pharmingen, 1:250: From the manufacturer's website: The MOPC-21 immunoglobulin is a mouse myeloma protein. The MOPC-21 immunoglobulin was selected as an isotype control following screening for low background on a variety of mouse and human tissues. The monoclonal antibody was purified from tissue culture supernatant or ascites by affinity chromatography. The antibody was conjugated with R-PE under optimum conditions, and unconjugated antibody and free PE were removed. The manufacturer states that the antibody is routinely tested by flow cytometry. |

# Eukaryotic cell lines

Policy information about cell lines

| Cell line source(s) | HEK 293T/17 cell line was sourced from ATCC. Product page: https://www.atcc.org/products/all/CRL-11268.aspx# |
|---|---|
| Authentication | None of the cell lines used were authenticated. |
| Mycoplasma contamination | The cell lines were not tested for mycoplasma contamination. |
| Commonly misidentified lines<br>(See ICLAC register) | *Name any commonly misidentified cell lines used in the study and provide a rationale for their use.* |

# Animals and other organisms

Policy information about studies involving animals; ARRIVE guidelines recommended for reporting animal research

| Laboratory animals | Pig-tailed macaque (Macaca nemestrina): adult 12.3 year male, 3.9 year female, 17.2 year male<br>Marmoset (Callithrix jacchus): adult (1.9-3.1 years), male and female<br>Mouse (Mus musculus): adult (P56 +/- 3 days) wildtype C57Bl/6J, male and female<br><br>For patch-seq, mouse (Mus musculus) M1 tissue was obtained from 4-12 week old male and female mice from the following transgenic lines: Thy1h-eyfp (B6.Cg-Tg(Thy1-YFP)-HJrs/J, RRID:IMSR_JAX:003782), Etv1-egfp Tg(Etv1-EGFP)BZ192Gsat/Mmucd, RRID:MMRRC_011152-UCD, (etv1) mice maintained with the outbred Charles River Swiss Webster background (Crl:CFW(SW, RRID:IMSR_CRL:024), and C57BL/6-Tg(Pvalb-tdTomato)15Gfng/J, RRID:IMSR_JAX:027395.<br><br>Mice were provided food and water ad libitum and were maintained on a regular 12-h day/night cycle at no more than five adult animals per cage. |
|---|---|
| Wild animals | No wild animals were used in this study. |
| Field-collected samples | No field-collected samples were used in this study. |
| Ethics oversight | Mouse experiments were conducted in accordance with the US National Institutes of Health Guide for the Care and Use of |

| Ethics oversight | Laboratory Animals under protocol numbers 0120-09-16, 1115-111-18, or 18-00006 and were approved by the Institutional Animal Care and Use Committee at University of Washington, Allen Institute for Brain Science, Salk Institute, or Massachusetts Institute of Technology. Marmoset experiments were approved by and in accordance with Massachusetts Institute of Technology IACUC protocol number 051705020. Macaque tissue used in this research was obtained from the University of Washington National Primate Resource Center, under a protocol approved by the University of Washington Institutional Animal Care and Use Committee. |
|---|---|

Note that full information on the approval of the study protocol must also be provided in the manuscript.

# Human research participants

Policy information about studies involving human research participants

| Population characteristics | Postmortem tissue donors used in the study:<br><br>Specimen ID Age Sex Race Cause of Death PMI (hr) Tissue RIN Hemisphere Sampled Data Type<br>H200.1023  43  F  Iranian descent  Mitral valve prolapse  18.5  7.4 ± 0.7  L  SSv4<br>H200.1025  50  M  Caucasian Cardiovascular 24.5  7.6 ± 1.0  L  SSv4<br>H200.1030  54  M  Caucasian Cardiovascular  25  7.7 ± 0.8  L  SSv4<br>H18.30.001 60 F Unknown Car accident 18 7.9 ± 2.5 R SSv4, Cv3, SNARE-seq2, sn-methlyome<br>H18.30.002 50 M Unknown Cardiovascular 10 8.2 ± 0.4 R SSv4, Cv3, SNARE-seq2, snmC-seq2<br><br>RIN, RNA integrity number. Data type: SMART-Seqv4 (SSv4), 10x Genomics Chromium Single Cell 3' Kit v3 (Cv3), Single-Nucleus Chromatin Accessibility and mRNA Expression sequencing (SNARE-seq2), Single nucleus methyl cytosine sequencing (snmC-seq2).<br><br>Neurosurgical tissue donor used in the study:<br>Donor was 61 F Unknown Glioblastoma grade IV, Patch-seq |
|---|---|
| Recruitment | Postmortem tissue specimens from males and females between 18 – 68 years of age with no known history of neuropsychiatric or neurological conditions ('control' cases) were considered for inclusion in this study of cell transcriptional profiles. Key conditions for exclusion were:<br><br>• Known brain injury, cancer or disease<br>• Known neuropsychiatric or neuropathological history<br>• Epilepsy or other seizure history<br>• Drug/alcohol dependency<br>• > 1 hour on ventilator<br>• Positive for infectious disease<br>• Prion disease<br>• Chronic renal failure<br>• Death from homicide or suicide<br>• Sleep apnea<br>• Time since death (postmortem interval, PMI) > 25 hours<br><br>Neurosurgical specimens: Tissue procurement from neurosurgical donor was performed outside of the supervision of the Allen Institute at a local hospital, and tissue was provided to the Allen Institute under the authority of the IRB of the participating hospital. A hospital-appointed case coordinator obtained informed consent from donor prior to surgery. |
| Ethics oversight | Postmortem adult human brain tissue was collected after obtaining permission from decedent next-of-kin. Postmortem tissue collection was performed in accordance with the provisions of the United States Uniform Anatomical Gift Act of 2006 described in the California Health and Safety Code section 7150 (effective 1/1/2008) and other applicable state and federal laws and regulations. The Western Institutional Review Board reviewed tissue collection processes and determined that they did not constitute human subjects research requiring institutional review board (IRB) review.<br><br>Tissue procurement from neurosurgical donor was performed outside of the supervision of the Allen Institute at a local hospital, and tissue was provided to the Allen Institute under the authority of the institutional review board of the participating hospital. A hospital-appointed case coordinator obtained informed consent from donor before surgery. Tissue specimens were de-identified before receipt by Allen Institute personnel. The specimens collected for this study were apparently non-pathological tissues removed during the normal course of surgery to access underlying pathological tissues. Tissue specimens collected were determined to be non-essential for diagnostic purposes by medical staff and would have otherwise been discarded. |

Note that full information on the approval of the study protocol must also be provided in the manuscript.

# Flow Cytometry

## Plots

Confirm that:

☒ The axis labels state the marker and fluorochrome used (e.g. CD4-FITC).

☒ The axis scales are clearly visible. Include numbers along axes only for bottom left plot of group (a 'group' is an analysis of identical markers).

☒ All plots are contour plots with outliers or pseudocolor plots.

☒ A numerical value for number of cells or percentage (with statistics) is provided.

## Methodology

| | |
|---|---|
| Sample preparation | Microdissected tissue pieces were placed in into nuclei isolation medium containing 10mM Tris pH 8.0 (Ambion) , 250mM sucrose, 25mM KCl (Ambion), 5mM MgCl2 (Ambion) 0.1% Triton-X 100 (Sigma Aldrich), 1% RNasin Plus, 1X protease inhibitor (Promega), and 0.1mM DTT in 1ml dounce homogenizer (Wheaton). Tissue was homogenized using 10 strokes of the loose dounce pestle followed by 10 strokes of the tight pestle and the resulting homogenate was passed through 30µm cell strainer (Miltenyi Biotech) and centrifuged at 900xg for 10 min to pellet nuclei. Nuclei were resuspended in buffer containing 1X PBS (Ambion), 0.8% nuclease-free BSA (Omni-Pur, EMD Millipore), and 0.5% RNasin Plus. Mouse anti-NeuN conjugated to PE (EMD Millipore) was added to preparations at a dilution of 1:500 and samples were incubated for 30 min at 4°C. Control samples were incubated with mouse IgG1,k-PE Isotype control (BD Pharmingen). Samples were then centrifuged for 5 min at 400xg to pellet nuclei and pellets were resuspended in 1X PBS, 0.8% BSA, and 0.5% RNasin Plus. DAPI (4', 6-diamidino-2-phenylindole, ThermoFisher Scientific) was applied to nuclei samples at a concentration of 0.1µg/ml. |
| Instrument | Single nucleus sorting was carried out on either a BD FACSAria II SORP or BD FACSAria Fusion instrument (BD Biosciences) |
| Software | BD Diva Software V8.0 |
| Cell population abundance | We intentionally sorted ~10% NeuN-negative (non-neuronal) and ~90% NeuN-positive (neuronal) nuclei to enrich for neurons. |
| Gating strategy | Nuclei were first gated based on size (forward scatter area, FSC-A) and granularity (side scatter area, SSC-A). B, Nuclei were then gated on DAPI fluorescence, followed by gates to exclude doublets and aggregates (FSC-single cells, SSC-single cells). E, Lastly, nuclei were gated based on NeuN PE ignal (NeuN-PE-A) to differentiate neuronal (NeuN+) and non-neuronal (NeuN-) nuclei. |

☒ Tick this box to confirm that a figure exemplifying the gating strategy is provided in the Supplementary Information.

