## [Peer Review File · Nature]

Manuscript Title: Comparative cellular analysis of motor cortex in human, marmoset, and mouse

Reviewer Comments & Author Rebuttals**Reviewer Reports on the Initial Version:**

Referees' comments:

Referee #1 (Remarks to the Author):

This manuscript by Bakken and colleagues provides a large amount of brain genomic data from humans and other species and will ultimately be an important resource for the neuroscience community. However, I have some suggestions to improve the manuscript and some points that require further clarification.

1) While there are some wonderful highlights and interesting findings (differing proportions of GABAergic vs Glutamatergic cells, comparisons of MTG L4 neurons to M1 etc.), the manuscript is very cumbersome to read often skipping around to different figures, very small text in figures, and many references to companion papers, unpublished papers, or preprints. It seems that there could be some editing and refining of the message the authors want to convey. Some suggestions would be: 1) limiting comparisons of SSv4 vs Cv3: comparisons of smartseq vs drop-seq like technologies have been done before, not much value is added based on the latest chemistries; this also done quite a bit in the companion mouse paper in bioRxiv; the addition of the alternative isoforms between species is interesting but indeed buried in supplemental anyway. 2) Considering removing SNARE-seq: the companion paper by Plongthongkum et al is not accessible to the reviewer and the human dataset is 10 times greater than the marmoset making the species comparison a bit unfair. There is no comparable single-cell ATAC-seq to show how doing SNARE-seq is "better" or comparable to integrating separate snRNA-seq and scATAC-seq from the same tissue. Plus, since it is only from 2 species it is not as valuable as the snRNA-seq or snmC-seq. 2) One assumes that these datasets will become part of larger cell-type atlases being generated. However, the claims about cell-type proportions and any discussions about non-neuronal cells are underpowered due to the method in which the authors obtained the data: FANS with 90% enrichment for NeuN+ cells. This information is buried in the supplement and should be made crystal clear to the reader that this dataset is really only intended to examine neurons. It's actually a shame as it would be quite interesting to have an unbiased assessment of relative proportions of all cell types across the species examined.

3) For the companion papers for which there are bioRxiv preprints, the Berg study seems to take these data and others and do a deep dive, while it is not clear that the mouse data in Yao really should be a separate paper or integrated better with this one. As mentioned already in point #1—it is very challenging for the reader to constantly refer back and forth to other papers.

4) The title of the paper refers to evolution of cellular diversity but this manuscript primarily focuses on the similarities across species. Moreover, a proper assessment of genomic evolution is not carried out as evolutionary distances are not included when making claims about changes among species. I suggest tempering such claims throughout the manuscript.

5) The authors make a big claim about overlapping marker genes across species but wouldn't this be expected as the nuclei are in the same cluster? Have the authors carried species-specific overall differential expression within clusters? Such results would truly address any evolutionary claims. In other words, the authors simply overlap the marker gene per cell type and call whatever doesn't overlap species specific. Not only do the authors not provide statistics for this, this excludes many non-marker genes that may have changed expression differently in different cell types. Species differences may be robust enough to not contradict the results of the overlap, but performing any

serious downstream analysis with these 'species-specific' genes would be propagating substantial error. I suggest the authors perform DEG analysis per cluster across species to determine the species specific genes.

Minor Comments:

- 1) No journal/bioRxiv reference provided for Plongthongkum et al.—this is a companion paper (also submitted to Nature?).
- 2) No journal/bioRxiv reference provided for Liu et al for snmC-seq2—this is a companion paper (also submitted to Nature?).
- 3) Stergachis reference not provided.
- 4) It would be helpful in Fig 1b to provide the number of individuals that went into generating each dataset so the reader does not have to dig for this in the supplement.
- 5) The Nissl stained sections in fig 1a are small and of poor image quality so hard to support the claims in the text.
- 6) What was the rationale for including the GABAergic integrated data in a main figure 2 and the glutamatergic cell types in supplemental? Based on the quality? The number of conserved markers? There are proportionally more glutamatergic neurons so it is surprising that those results went in the supplementary data.
- 7) For fig 3d. why are cluster marker DARs normalized to cluster size? The total number of DARs don't have to linearly increase as the sample size increases, rather one might expect it to increase at first but relatively stabilize later after sample size has enough power. Having said that, since GABAergic clusters tend to be fewer in number, normalized cluster specific DARs can be inflated for GABAergic clusters. This means that the authors conclusion of 'more markers in glutamatergic' will probably hold but it is hard to justify the normalization since it assumes that sample size and p-value has a linear relationship but they do not. Also, the legend writes ' human or marmoset '. This is a barplot so I guess this is just human? Is marmoset similar?
- 8) The comment about "correspondence between human and marmoset was higher for glutamatergic rather than GABAergic neurons (Fig. 3h,j)." is very hard to assess in the figure. It looks equivalent or if anything that there is better correspondence for GABAergic cell types. For example, looking at Fig 3j, it's not clear why there is the use of comparing different human and marmoset subclasses given that the authors talk only about human-marmoset comparison in matched subclasses. In other words, in Fig 3j, why are the comparisons other than the ones at the diagonal interesting?

Additionally, it is difficult following the arguments in the main text in general:

"While most subclasses also showed distinct TFBS activities, correspondence between human and marmoset was higher for glutamatergic rather than GABAergic neurons"

It seems that some glutamatergic neuron markers have similarly conserved activity between human-marmoset compared to GABAergic. But some others are more dissimilar than GABAergic neurons (L6 IT, L2-3 IT, L6 IT, L5 ET). The results don't support the argument.

"For GABAergic neuron subclasses, gene expression profiles were more conserved than TFBS activities, consistent with fewer differences between GABAergic subpopulations based on AC sites (Fig. 3a,b)."

Neither panels have information on TFBS activity. Figure 3j is more helpful but the less similarity for TFBS compared to gene expression argument can be made for both GABAergic and glutamatergic. The results don't appear to support the argument here. Also, isn't it surprising that gene expression profiles are similar but TFBS are not? Interestingly, the authors later argue that TFBS motif enrichment is more conserved (Fig 4) than gene expression. While those results are not very convincing either (see other comments), it would be helpful to see some results and discussion on the distinction between TFBS activity and TFBS motif enrichment. Because, it appears that one is less conserved across species than gene expression (TFBS activity) while the other one is argued to be more conserved (TFBS motif enrichment).

"Interestingly, glutamatergic neurons in L5 and L6 showed higher correspondence between primates based on TFBS activities compared to average expression, suggesting that gene regulatory processes are more highly conserved in these subclasses than target gene expression."

Judging by the diagonal of Figure 3j, it seems to be the opposite. Also, there seems to be similar levels of dissimilarity between TFBS and gene expression plots for GABAergic and glutamatergic clusters. The authors should create some sort of metric for TFBS-gene expression difference and plot conservation for each cluster clearly if they want to make such points.

9) For all of the Venn diagrams it would be helpful to have statistics on the overlaps as there are several claims about more or less overlapping genes but the proportion needs to be considered.

10) In figure 2f, there are some stark contrasts between human and mouse. How do these cell types look in the layers of marmoset? Otherwise, it is hard to know if these differences are primate or human-specific.

11) "Although subclasses had unique marker genes (Fig. 2c, genes listed in Supplementary Table 8) and CH-DMG across species, they had remarkably conserved TFBS motif enrichment (Fig. 4e,f and Extended Data Fig. 8)."

What are the scales in Fig4e and what values were used for the tSNE in Fig 4f (e.g fold change, log pval)? For Fig 4f, if the size of the boxes is the number of TFBS for the given TF-cluster pair, there seems to be quite a bit of variability in the TFBS across species for some TFs that the authors selected to show. Fig 4f and the Extended Fig. 8 shows that within cluster difference is less than across cluster difference indicating conservation but this might also be true for gene expression.

The authors can try averaging cluster marker expression per species per cluster for scRNAseq and plotting tSNE if they want to justify this comparison. Even then one could argue that the TFs with identified motifs tend to be the more crucial hence conserved ones and it's not fair to compare this to an unbiased gene-expression analysis. But as it is, the statement is weakly supported.

12) One wonders if the "paradoxical observation" regarding species differences in consensus cell types is due to some technical issue due to number of cells or sequencing depth?

13) The finding that there is no "exclusive" Betz transcriptomic type—could that be due to a technical issue regarding clustering—does changing the resolution of clustering affect this conclusion. What about in the metacell analysis—does a Betz cell type emerge?

14) "To discern if these TFs may preferentially bind to DNA in ChCs, we tested for TF motif enrichment in hypo-methylated (mCG) DMRs and AC sites genome-wide."

Why is this an interesting thing to check? If these TFs are marker genes for ChCs, they should be binding regions that are in open chromatin formation. The results are very much expected and perhaps belong in supplementary.

15) "Differentially accessible regions (DARs) between cell populations (Fig. 4b) were identified using the "find_all_diff" function (<https://github.com/yanwu2014/chromfunks>) and p-values calculated using a hypergeometric test."

Does this function take total cell accessibility into account as a latent variable? The probability of a read being present or not in a peak is influenced by the total number of reads in peaks for the given cell. Other recent single-cell ATAC-seq papers take this into account in their modelling. While the top DARs are likely to be similar, the final list of DARs may change considerably when this important covariate is taken into account.

16) "Peak regions were called independently for RNA cluster, subclass and class groupings using MACS2 software (<https://github.com/taoliu/MACS>) using the following options "--nomodel --shift 100 --ext 200 --qval 5e-2 -B --SPMR". Peak regions were combined across peak callings and used to generate a single peak count matrix (cell barcodes by chromosomal peak locations) using the "createPmat" function of SnapATAC."

The explanation is not very clear. Are the peaks called by combining all reads or per cluster identified using snapATAC? Was the peak calling performed separately for each subject? What exactly were the 'peak callings'? The authors should be more clear. It would be good to perform peak calling separately for each sample and see whether they differ substantially. If not, simple merging should be acceptable. But if they are quite different - most likely due to technical reasons- this would mean that for many peaks the sample will essentially be 1 if the peaks are just merged. In that case only the consensus peaks should be taken.

17) The following sentence is difficult to understand:

"The AC-level clusters (Fig. 3a,b) that showed similar coverage across individual samples (Extended Data Fig. 6c-f) revealed regions of open chromatin that are extremely cell type specific."

Do the clusters with similar coverage across individuals have more DARs? If so, why would that be important? How is 'extremely cell type specific' any different than the term 'cluster marker'?

18) The authors state that log₂ CPM data were put into Seurat. Were the log₂ data further log-transformed then?

19) Seurat was used for Cv3 data but Pagoda was used for SnareSeq data making the results not entirely comparable.

20) More details on the methylation analysis is needed. Were covariates included in calling DARs? In the absence of the companion paper it is difficult to know exactly what was done. And the examples shown of it "working" seem a bit cherry picked—the marker genes from the previous figures are not shown and the evolutionary comparisons are done in a different manner.

21) Figure 5 a&b legends appear switched.

22) The ephys data descriptions are very convoluted and difficult to follow. It is not even clear what the N is for each experiment.

23) For Figure 7a, can the authors speculate why mouse would have the second greatest # of species-specific marker genes for L5 ET vs IT? (i.e. more than macaque or marmoset?)

24) It is not clear how many cells, sections, and individuals were included for each FISH/IHC confirmation. In most cases we see one representative nuclei in the figure.

25) The finding of neuronal genes in oligos is quite interesting but the interpretation stated below has no reference cited about phagocytosis. "This may represent an oligodendrocyte type that expresses neuronal genes or could represent phagocytosis of parts of neurons and accompanying transcripts that are sequestered in phagolysosomes adjacent to nuclei."

Referee #2 (Remarks to the Author):

A. SUMMARY OF THE KEY RESULTS

The focus of the Bakken et al. manuscript is on the cell diversity of the primary motor cortex (M1) of human, marmoset monkey and mouse. M1 is essential for voluntary fine motor control and have various species differences in terms of direct and indirect innervation of the spinal cord motoneurons and interneurons. L5 of carnivore and primate M1 contains exceptionally large "giganto-cellular" corticospinal neurons (Betz cells in primates). Some primate Betz cells directly synapse onto alpha motor neurons, whereas in cats and rodents these neurons synapse instead onto spinal interneurons. The paper compares cellular components in these three species and it provides cross-species consensus cell type classification and inference of conserved cell type properties across species. This is a huge achievement and I could only read the paper with admiration and enthusiasm. This is a huge operation and it is beginning to bear fruits in these projects that are way beyond the capability of a single laboratory. This comparative evolutionary approach provides a stable platform to define the cellular architecture of our brain and to discover species-specific adaptations. This paper decided to compare one area in three species, but one could have argued for comparing several areas in one species or several areas during development. The study is focusing on M1. Several approaches would have been justified, but starting in M1 is not a bad choice. This is a functionally and anatomically relatively conserved cortical region across mammals. The study also allows the comparison of a variety of methods on similarly isolated tissues.

The paper reports differences in cell type proportions, gene expression, DNA methylation, and chromatin state. The study dedicates a considerable effort to characterize the exceptionally large "giganto-cellular" corticospinal neurons (Betz cells in primates) in M1 in these three species. This is an extremely interesting cell population. I understand that the mouse datasets are also reported in a companion paper, that I did not receive.

B. ORIGINALITY AND SIGNIFICANCE: IF NOT NOVEL, PLEASE INCLUDE REFERENCE

The paper is much more than just presenting a huge dataset that no other institution could deliver. The manuscript also uses this dataset to try to answer some fundamental principles of cortical organization. I believe that this is why this paper deserves to be published in Nature.

1. The paper describes similar cellular complexity on the order of 100 cell types was seen in all three species. The study identified some core conserved genes and make strong predictions about the TF code for cell types and the genes responsible for their evolutionarily constrained functions. Exploring the links between genes and cellular phenotypes for conserved and divergent features will have to be tested, but this can take much more effort and much more detailed analysis, perhaps by extending these studies to birds and reptiles.
2. The paper compares different methodologies and their impact on the outcome of the analysis. They present important comparisons between plate-based (SSv4) and droplet-based (Cv3) RNA-seq of human nuclei. The authors compared results between approximately 10,000 SSv4 and 100,000 Cv3 nuclei. The study reports that on average, SSv4 detected 30% more genes per nucleus and enabled comparisons of isoform usage between cell types. This had 20-fold greater sequencing depth, but SSv4 cost 10 times as much as Cv3 and did not allow detection of additional cell types.
3. The paper describes relative similarities and differences between the three species examined. As expected, the more closely related species are more similar to one another. It would have been interesting to explore whether these similarities and differences are on the same scale in different cortical areas.
4. One important confirmation of previous studies was that the ratio of glutamatergic excitatory projection neurons compared to GABAergic inhibitory interneurons was 2:1 in human compared to 3:1 in marmoset and 5:1 in mouse. This shift in the overall excitation-inhibition balance of the cortex was described by other methods, but this is perhaps the most elegant and powerful demonstration of these principles.
5. Interestingly the relative proportions of GABAergic neuron subclasses and types were similar across species.
6. It was also suggested by previous studies using different methods that there is a large increase in the proportion of L2 and L3 IT in human compared to mouse and marmoset, nevertheless it is very reassuring to see all this with a more sophisticated and reliable method.
7. The paper has some important observations on Layer 4 in M1 in the three species. M1 does have L4-like cells based on the transcriptomic signature, but only a subset of the types compared to granular cortical areas, at much lower density, and scattered rather than aggregated into a tight layer.
8. The study identify the transcriptomic cluster corresponding to Betz cells and use this further to understand gene expression that may underlie their distinctive properties. The study concludes that there does not appear to be an exclusively Betz transcriptomic type. The authors find more than one ET cluster contains neurons with Betz morphology and conclude that betz cells may not in fact be completely restricted to M1 but distribute across other proximal motor-related areas that contribute to the pyramidal tract

These are all very fundamental findings and each point could take years to further study in detail. The study is balancing between the time pressures of releasing this important dataset for the general scientific community, but they also would like to get the most important and most significant findings already analyzed and described. I think this balancing act is successful in this

case.

C. DATA & METHODOLOGY; D. APPROPRIATE USE OF STATISTICS

The authors have a well-established pipeline with a distinguished advisory board, therefore works according to the most modern tools and standards.

E. CONCLUSIONS: ROBUSTNESS, VALIDITY, RELIABILITY

Does the manuscript have flaws, which should prohibit its publication? If so, please provide details.

The manuscript contains huge amount of work. Very impressive data, collected with the state of the art methodologies on a scale that a normal academic-research laboratory could not afford and completely out of reach. It is like sending up a satellite to observe the continents and report back to teams that used small-scale approaches to map the continents. We get staggering insights and views! However, there is also a challenge what to describe and how to interpret the initial results. As I outlined above (1-8) the paper contains some fundamental issues and these are approached responsibly and with understanding. However, I do not have time to go through all the possible ramifications that could be done with this data. Almost all of the above (1-8) points could be extended much further and I am sure the authors are all aware of the possibilities. Moreover, it would take a lot of time to detail the directions that the authors could take with each of these directions.

F. SUGGESTED IMPROVEMENTS: EXPERIMENTS, DATA FOR POSSIBLE REVISION

I shall concentrate my comments on the issue of Betz cells (point 8 above), but I could have picked any of the other points, they could be developed much further with time. The presented analysis is just the tip of the iceberg.

(a) Betz cell and L5 ET neurons are used interchangeably in several parts of the text and this should be changed to discern the potential differences between Betz cells and large L5 neurons. For example, many of the features highlighted within L5 ET neurons are found under the heading "Primate Betz cell specialization", the differences between ET neurons and IT neurons should be written elsewhere or the heading should be changed to reflect the findings found in ET neurons or Betz cells.

(b) In the main text, the authors mention they found x2 Betz cell clusters "Exc L5 FEZF2 ASGR2" & "Exc FEZF2 CSN1S1", however in their figures they state that neurons found within layer 3 of the human motor cortex also clustered to the Betz Exc L5 FEZF2 ASGR2 cluster. The authors should, therefore, either not refer to this as a Betz cell cluster or strictly use the "Exc L3-L5 FEZF2 ASGR2" nomenclature found in their figure throughout the entirety of the text for clarity. Furthermore, a better explanation is needed to explain why L3 neurons and Betz cells cluster together in their dataset.

(c) The authors find several Betz cell clusters, however, they fail to show convincing validation. They only show x2 images of RNAscope staining on Betz cells and draw the conclusion that this represents the transcriptomic profile of their Betz clusters. It is not clear why they selected those probes for staining, are the probe signatures unique to Betz clusters? Or do they just show transcripts that are particularly enriched? The rationale behind these probes should be explained in the text. The best approach would be to improve upon extended data figure 10e to show the distribution and abundances on a larger scale. Moreover, the probes they have selected (Neurofilament, GRIN3A, POU3F1 and SERPINE2) also individually stain many other cells in the primary motor cortex so an accompanying image showing the staining pattern of Betz cells vs non-Betz L5 neurons/non-L5 neurons is needed.

(d) The ISH images presented are not clear and often positive staining cannot be easily seen. For example, within extended data figure 10f it is not clear that both Betz cells are POU3F1 positive despite the text stating otherwise. Clearer images should be used; maybe even single channels and a composite to allow the reader to better investigate the staining.

(e) The authors need to explain how they histologically determined Betz cells from large L5 neurons in their ISH validation. There are at least 5 histological defining features, but it appears the authors only use size and location that can commonly misidentify Betz cells.

(f) More information from the literature is needed to explain the presence of Betz cells in the premotor cortex since it is known that the vast majority are found in the primary motor cortex. Again, a detailed neurohistological description is needed to increase confidence that the human motor neurons from the premotor cortex used for PATCH-Seq were indeed Betz cells, especially since it is not strongly supported in the literature if Betz cells truly exist in the premotor cortex. The citation used here is in reference to an electrophysiological study using only macaques that studied large pyramidal neurons in the premotor cortex and does not mention humans. Large pyramidal neurons are found across the neocortex and this neuron selected for PATCH-Seq may just be a "regular" large pyramidal neuron.

(g) A selection criterion is needed to outline how the authors selected Betz cells for macaque patch-seq analysis, for example, the macaque Betz cell used had a soma size of >65um; size isn't the defining feature of a Betz cell so an explanation on why they believe this neuron is a Betz cell is needed, including information on how they prepared this tissue block. They should also, look at the literature to find the size of an average macaque Betz cell as size varies a lot between species.

(h) Clusters "Exc L5 FEZF2 ASGR2" & "Exc FEZF2 CSN1S1" should be compared with other L5 glutamatergic clusters to discern potential differences and reveal potential Betz specific markers. This would aid in the author's goal to study how Betz cells differ from other neuronal subtypes. At present Betz cell clusters are only compared to IT neurons.

(i) The authors also presume and group Betz cells as layer V extratelencephalic neurons, while this makes sense from a global approach where not a lot is known about the transcriptome of the Betz cell there may be subpopulations of Betz cells that project to other regions and therefore could be accidentally excluded from this analysis when this dogma is implemented.

(j) Authors state that the ion channel subunits found may reflect Betz cell physiology, but again, as the Betz cell clusters also contained other neuronal cell types this should be spatially validated with ISH.

(k) The authors found that ROBO, SLIT and EPHRIN are found at higher amounts in primate ET neurons when compared with primate IT neurons and attributed to axon projection distance. ISH should be performed to see if this is unique to ET neurons or to Betz cells, or if Betz cells with the longest axons express more of these transcripts.

(l) The paper argues that "axon guidance-associated genes are enriched in Betz-containing ET neuron types in primates, possibly explaining why Betz cells in primates directly contact spinal motor neurons rather than spinal interneurons as in rodents." However, all these studies were done at adult stages. Most of the actual guidance factors might have been gone by this stage. I would tune down these claims, since in best case these specific gene expression patterns are involved in the synaptic maintenance, but not the formation.

G. REFERENCES: APPROPRIATE CREDIT TO PREVIOUS WORK

This is the first such comprehensive reports. As such it is highly original. Yes, some aspects of the

debate started decades before this paper, but now one can examine these in a more comprehensive fashion. The authors cite the older literature and put things into context.

On a more subjective note, do you feel that the results presented are of immediate interest to many people in your own discipline, or to people from several disciplines?

All neuroscientists should be interested in this data in some form. I am surprised that there was a relative lack of interest in bioRxiv.org, only one comment so far:

Comment from BioRxiv:

Miguel Angel Garcia-Cabezas • a month ago

The authors of the manuscript entitled "Evolution of cellular diversity in primary motor cortex of human, marmoset, monkey, and mouse" found that "transcriptomically similar cell types were found at similar cortical depths in M1 and MTG, and the OTOGL and LINC01202 types were located in deep L3 and superficial L5 in M1". In the Discussion they state that "M1 is an agranular cortex lacking a L4, although a recent study demonstrated that there are neurons with L4-like properties in mouse¹⁴. Here we confirm and extend this finding in human M1. We find a L4-like neuron type in M1 that aligns to a L4 type in human MTG and is scattered between the deep part of L3 and the superficial part of L5 where L4 would be if aggregated into a layer".

However, within the published literature, the existence of layer IV in the human primary motor cortex was first described by Ramón y Cajal [Estudios sobre la corteza cerebral humana. II La corteza motriz del hombre y mamíferos superiores. Rev Trim Microg. 1899 (4): 117–200]. This finding was later confirmed by Marín-Padilla [Prenatal and early postnatal ontogenesis of the human motor cortex: a Golgi study. I. The sequential development of the cortical layers. Brain Res. 1970; 23 (2): 167-83]. Layer IV has also been described for the primary motor cortex of rhesus macaques qualitatively [Gatter et al. The intrinsic connections of the cortex of area 4 of the monkey. Brain. 1978; 101 (3): 513-41]. More recently, a detailed study in rhesus monkeys with analysis at the cellular and subcellular levels and rigorous analytic methods, provided strong quantitative evidence for the presence and cellular features of layer IV, with discussion of the relevant history and ideas [García-Cabezas & Barbas. Area 4 has layer IV in adult primates. Eur J Neurosci. 2014; 39 (11): 1824-34; and Barbas and Garcia-Cabezas, Motor cortex layer 4: less is more. Trends Neurosci. 2015; 38(5): 259-61].

The authors can claim that they confirm the existence of layer IV in the primary motor area of the human cortex, but they don't extend "this finding in human M1".

If you recommend publication, please outline, in a paragraph or so, what you consider to be the outstanding features.

This paper should be received by Nature with open arms. It is a landmark study that not only contains a huge unique dataset that will attract hundreds of citations, but it already contains some analysis that is answering some fundamental questions. A key result of the current study is the identification of a consensus classification of cell types across species that allows the comparison of relative similarities in human compared to common mammalian model organisms in biomedical research. There are lots of important findings (see 1-8 points above), including Layer 5 corticospinal Betz cells in non-human primate and human and characterization of their highly specialized physiology and anatomy. The data presented in the paper allow a targeted search for genes responsible for species specializations such as the distinctive anatomy, physiology and axonal projections of Betz cells, large corticospinal neurons in primates that are responsible for voluntary fine motor control.

H. CLARITY AND CONTEXT

All fine. It is very well written and put together. It is not an easy read, but it would be very difficult to find things to considerably improve the manuscript.

Author Rebuttals to Initial Comments:

Referee #1 (Remarks to the Author):

This manuscript by Bakken and colleagues provides a large amount of brain genomic data from humans and other species and will ultimately be an important resource for the neuroscience community. However, I have some suggestions to improve the manuscript and some points that require further clarification.

1) While there are some wonderful highlights and interesting findings (differing proportions of GABAergic vs Glutamatergic cells, comparisons of MTG L4 neurons to M1 etc.), the manuscript is very cumbersome to read often skipping around to different figures, very small text in figures, and many references to companion papers, unpublished papers, or preprints. It seems that there could be some editing and refining of the message the authors want to convey. Some suggestions would be: 1) limiting comparisons of Ssv4 vs Cv3: comparisons of smartseq vs drop-seq like technologies have been done before, not much value is added based on the latest chemistries; this also done quite a bit in the companion mouse paper in bioRxiv; the addition of the alternative isoforms between species is interesting but indeed buried in supplemental anyway. 2) Considering removing SNARE-seq: the companion paper by Plongthongkum et al is not accessible to the reviewer and the human dataset is 10 times greater than the marmoset making the species comparison a bit unfair. There is no comparable single-cell ATAC-seq to show how doing SNARE-seq is “better” or comparable to integrating separate snRNA-seq and scATAC-seq from the same tissue. Plus, since it is only from 2 species it is not as valuable as the snRNA-seq or snmC-seq.

We revised the manuscript to condense and clarify the results, in particular combining the two epigenomics figures (SNARE-seq2 and DNA-methylation) into a single figure focused on gene regulation of M1 cell types. Wherever possible, we have simplified figures (for example, removing a complex genome browser tracks panel from the chandelier cell figure) and increased font sizes for readability.

Regarding point 1, we have included Smart-seq v4 because it provides several pieces of useful information. First, we performed laminar dissections in human M1 for Ssv4 and not Cv3 profiling, and alignment of Ssv4 and Cv3 data enables inference of the approximate laminar distributions of transcriptomic clusters as shown in Figure 1c. Layer information also provided an additional validation of the alignment of human and mouse homologous cell types as shown in Figure 2f. Finally, layer information helped identify a L4-like type in M1 that we then validated by in situ labeling (Figure 5). Also, as noted, Ssv4 enables comparison of isoform usage between species, and we have now highlighted these results in main Figure 3. We have de-emphasized comparisons of Ssv4 and Cv3 methods and include only those results necessary to support the findings described above.

Regarding point 2 (SNARE-Seq), we have performed several analyses comparing chromatin and expression both separately and together in two previous Nature Biotechnology publications (<https://www.nature.com/articles/nbt.4038>; <https://www.nature.com/articles/s41587-019-0290-0>). These prior analyses have demonstrated comparable data quality from SNARE-Seq and scATAC-seq methods as well as evidence supporting improvement both for cell type annotation and the detection of potential cis-regulatory elements from jointly profiled RNA and chromatin modalities. As such we do not feel it would be necessary to replicate any such cross-platform comparative analyses. We have now improved references to these prior studies within the manuscript. Further, we have ensured that any reference to our companion Nature Protocols paper (Plongthongkum et al.) is only in relation to its capacity as a methods paper and not a source of such comparisons. We have also included a preprint of Plongthongkum et al. along with the revised manuscript. Given the previously demonstrated effectiveness for the SNARE-Seq, we feel it is not appropriate nor necessary to remove this assay from the manuscript. Furthermore, we agree that species comparison with marmoset is hampered by the latter's shallower sampling size, an unfortunate consequence of the more limiting size and nuclei yield for this region from marmoset compared to human. However, information provided from this assay within species remains valuable and comparisons between species was performed only at the more aggregated subclass level to ensure sufficient sampling depth. Supporting this, we identified a similar trend in the number of differentially accessible regions (DARs) discovered between neuronal subclasses for both human and marmoset, which was also consistent with that found for DNA hypomethylated sites (Fig. 4). We also demonstrate a high correlation in transcription factor expression and binding site activities across species (Fig. 4).

2) One assumes that these datasets will become part of larger cell-type atlases being generated. However, the claims about cell-type proportions and any discussions about non-neuronal cells are underpowered due to the method in which the authors obtained the data: FANS with 90% enrichment for NeuN+ cells. This information is buried in the supplement and should be made crystal clear to the reader that this dataset is really only intended to examine neurons. It's actually a shame as it would be quite interesting to have an unbiased assessment of relative proportions of all cell types across the species examined.

We have added the NeuN sorting criteria to the results along with a statement that "non-neuronal cells were undersampled, and cellular diversity is likely under-represented". We have also made a clearer distinction between neuronal and non-neuronal branches of the taxonomies in Figure 1. While we have likely missed detection of rare non-neuronal types, particularly in human, we think there is value in describing the non-neuronal types that we do detect, such as transcriptomic markers of morphological described subtypes of astrocytes that we report in ED Figure 3.

3) For the companion papers for which there are bioRxiv preprints, the Berg study seems to take these data and others and do a deep dive, while it is not clear that the mouse data in Yao really

should be a separate paper or integrated better with this one. As mentioned already in point #1—it is very challenging for the reader to constantly refer back and forth to other papers.

We recognize that these papers collectively cover many techniques and topics, and the links between them present a challenge to the reader. Yao et al. focuses on cross-modal integration, cell type robustness, and linking to prior studies of mouse cell types. This study builds on this work to compare M1 cell types across species, and it would potentially be more unwieldy to incorporate all of these results in a single manuscript. We have worked toward strong integration between the manuscripts by using the exact same taxonomy of mouse cell types and using many of the same analysis pipelines for comparable datasets. Furthermore, our data integration results provide a link between human and marmoset M1 types and mouse types from Yao et al. Berg et al. uses Patch-seq to characterize supragranular glutamatergic neurons, while we apply this technique to characterize L5 neurons, including Betz cells. For clarity, we have cited other manuscripts for detailed methods rather than duplicating them here.

4) The title of the paper refers to evolution of cellular diversity but this manuscript primarily focuses on the similarities across species. Moreover, a proper assessment of genomic evolution is not carried out as evolutionary distances are not included when making claims about changes among species. I suggest tempering such claims throughout the manuscript.

We thank the reviewer for the suggestion. We have retitled the manuscript “Comparative cellular analysis of motor cortex in human, marmoset, and mouse” and have revised the text to reflect our comparative approach that does not account for phylogeny. We have attempted to highlight differences across species where possible, such as cell type proportions, marker expression, Betz cell properties, etc.

5) The authors make a big claim about overlapping marker genes across species but wouldn't this be expected as the nuclei are in the same cluster? Have the authors carried species-specific overall differential expression within clusters? Such results would truly address any evolutionary claims. In other words, the authors simply overlap the marker gene per cell type and call whatever doesn't overlap species specific. Not only do the authors not provide statistics for this, this excludes many non-marker genes that may have changed expression differently in different cell types. Species differences may be robust enough to not contradict the results of the overlap, but performing any serious downstream analysis with these 'species-specific' genes would be propagating substantial error. I suggest the authors perform DEG analysis per cluster across species to determine the species specific genes.

The reviewer suggests that ‘species-specific’ marker genes should be defined by directly comparing each subclass across species. This is an important question, and we performed this analysis for L1 astrocytes in Extended Data Figure 3e. Most genes have similar expression across species, while ~200 genes differ >8-fold between human and marmoset and 400-500 genes between primates and mouse. However, this analysis does not highlight DE genes that are specifically expressed in L1 astrocytes, which we are most interested in because they may be central to the identity of that cell type. Rather, our intent is to report subclass marker genes and how they vary across species. To check to what degree “species-specific” genes were expressed in other species, we performed further analysis on genes in the ‘species-specific’ portions of the Venn diagrams. For example, we compared expression of ‘human-specific’ Vip markers to marmoset and mouse Vip expression. We then indicated what proportion of human-specific Vip markers had $> 0.25 - 2 \log_{2}FC$ (i.e. >1.25-7.4 fold) expression in human compared to the other two species. We summarized the results for Vip and other GABAergic subclasses in the following figure and added a description to the results: “Note that a majority of species-enriched markers were expressed in other species but at lower levels or reduced specificity (Extended Data Fig. 2a).” We have renamed ‘species-specific’ markers to be ‘species-enriched’ to avoid confusion in our claims.

We have also included methods text stating how we performed ROC analysis comparing each subclass to other nuclei from the same class. Only genes that had AUROCs greater than 0.7, and were expressed in at least 10% of the target subclass were called marker genes. These criteria yielded robust marker genes for each subclass for each species (visualized as a heatmap in Figure 2c). The magnitude of differential expression for conserved ROC-defined marker genes can also be seen in Figure 6c for chandelier cells. We have also now included summary statistics for the ROC analysis in supplementary tables 7 and 10.

Minor Comments:

1) No journal/bioRxiv reference provided for Plongthongkum et al.—this is a companion paper (also submitted to Nature?).

The Plongthongkum et al. companion paper is a Nature Protocols paper that is currently under review. We were not able to submit this to bioRxiv given that it is a protocol paper, but we have provided a pdf of the submitted manuscript upon resubmission. Further, we have now published this protocol on protocols.io and have included this link in the manuscript.

2) No journal/bioRxiv reference provided for Liu et al for snmC-seq2—this is a companion paper (also submitted to Nature?).

We have now included a link to the bioRxiv preprint in the manuscript:
<https://www.biorxiv.org/content/10.1101/2020.04.30.069377v1>

3) Stergachis reference not provided.

We have added the Stergachis reference.

4) It would be helpful in Fig 1b to provide the number of individuals that went into generating each dataset so the reader does not have to dig for this in the supplement.

We have added the number of donors in parentheses; p indicates pooled biological replicates.

5) The Nissl stained sections in fig 1a are small and or poor image quality so hard to support the claims in the text.

We have added higher resolution Nissls in Fig. 1a and added insets to highlight Betz cells in human and larger pyramidal neurons in marmoset and mouse.

6) What was the rationale for including the GABAergic integrated data in a main figure 2 and the glutamatergic cell types in supplemental? Based on the quality? The number of conserved markers? There are proportionally more glutamatergic neurons so it is surprising that those results went in the supplementary data.

This was supplemental due to space limitations. We have consolidated the two epigenomics figures into one and moved the glutamatergic cell types to main figure 3.

7) For fig 3d. why are cluster marker DARs normalized to cluster size? The total number of DARs don't have to linearly increase as the sample size increases, rather one might expect it to increase at first but relatively stabilize later after sample size has enough power. Having said that, since GABAergic clusters tend to be fewer in number, normalized cluster specific DARs can be inflated for GABAergic clusters. This means that the authors conclusion of 'more markers in glutamatergic' will probably hold but it is hard to justify the normalization since it assumes that sample size and p-value has a linear relationship but they do not. Also, the legend writes 'human or marmoset'. This is a barplot so I guess this is just human? Is marmoset similar?

We agree with the reviewer and have now refined these analyses to better account for differences in both cluster sizes and total accessibility levels of the cells. This includes subsampling subclasses to better match cluster sizes and the comparison of clusters to a random background of cells having comparable total peak counts (see methods). We further calculate AUC values testing the separation power of a specific DAR among different subclasses to further ensure accuracy of the subclass specific DARs. With these modifications, we observe a high concordance in the proportion of DARs by cell type between species and across platforms (with DMR proportions, Figure 4), providing further confirmation of our method and findings.

8) The comment about "correspondence between human and marmoset was higher for glutamatergic rather than GABAergic neurons (Fig. 3h,j)." is very hard to assess in the figure. It looks equivalent or if anything that there is better correspondence for GABAergic cell types. For example, looking at Fig 3j, it's not clear why there is the use of comparing different human and marmoset subclasses given that the authors talk only about human-marmoset comparison in matched subclasses. In other words, in Fig 3j, why are the comparisons other than the ones at the diagonal interesting?

Additionally, it is difficult following the arguments in the main text in general:

"While most subclasses also showed distinct TFBS activities, correspondence between human and marmoset was higher for glutamatergic rather than GABAergic neurons"

It seems that some glutamatergic neuron markers have similarly conserved activity between human-marmoset compared to GABAergic. But some others are more dissimilar than GABAergic neurons (L6 IT, L2-3 IT, L6 IT, L5 ET). The results don't support the argument.

"For GABAergic neuron subclasses, gene expression profiles were more conserved than TFBS activities, consistent with fewer differences between GABAergic subpopulations based on AC sites (Fig. 3a,b)."

Neither panels have information on TFBS activity. Figure 3j is more helpful but the less similarity for TFBS compared to gene expression argument can be made for both GABAergic and glutamatergic. The results don't appear to support the argument here. Also, isn't it surprising that gene expression profiles are similar but TFBS are not? Interestingly, the authors later argue that TFBS motif enrichment is more conserved (Fig 4) than gene expression. While those results are not very convincing either (see other comments), it would be helpful to see some results and discussion on the distinction between TFBS activity and TFBS motif enrichment. Because, it appears that one is less conserved across species than gene expression (TFBS activity) while the other one is argued to be more conserved (TFBS motif enrichment).

"Interestingly, glutamatergic neurons in L5 and L6 showed higher correspondence between primates based on TFBS activities compared to average expression, suggesting that gene regulatory processes are more highly conserved in these subclasses than target gene expression."

Judging by the diagonal of Figure 3j, it seems to be the opposite. Also, there seems to be similar levels of dissimilarity between TFBS and gene expression plots for GABAergic and glutamatergic clusters. The authors should create some sort of metric for TFBS-gene expression difference and plot conservation for each cluster clearly if they want to make such points.

A number of the observations highlighted by the reviewer were associated with off diagonal comparisons, and we agree that these are not as clear or as informative as on diagonal or matched subclass comparisons. We have now revised these analyses and integrated them further with those for DNAm, where consistent trends across species can be more conclusive. We do find an overall significant conservation of TFBS activities (AC data) and TF motif enrichments (DNAm data) across species, which was also observed for corresponding TF expression levels, but less so for marker gene expression values (see Figure 4). These results predict more conserved TF networks that may determine either conserved or divergent expression levels depending on whether the genomic locations of TFBS motifs are shared or altered across species. We have now discussed this in the results section "Cell type-specific gene regulation".

9) For all of the Venn diagrams it would be helpful to have statistics on the overlaps as there are several claims about more or less overlapping genes but the proportion needs to be considered.

We thank the reviewer for this suggestion. We have added the following text to the results:

“As expected based on their closer evolutionary distance, human and marmoset shared more markers (on average 25%) with each other compared to with mouse (16%) for 13 of 14 neuronal subclasses (Fig. 2b, 3b), and half of these differences were statistically significant (Bonferroni-adjusted P-value < 0.05; chi-square test) despite few total markers. ”

10) In figure 2f, there are some stark contrasts between human and mouse. How do these cell types look in the layers of marmoset? Otherwise, it is hard to know if these differences are primate or human-specific.

We agree that this would be an important comparison, but we do not have layer dissection information for marmoset. We avoid making claims about the evolutionary timing of the change in layers when we discuss the differences between human and mouse.

11) "Although subclasses had unique marker genes (Fig. 2c, genes listed in Supplementary Table 8) and CH-DMG across species, they had remarkably conserved TFBS motif enrichment (Fig. 4e,f and Extended Data Fig. 8)."

What are the scales in Fig4e and what values were used for the tSNE in Fig 4f (e.g fold change, log pval)? For Fig 4f, if the size of the boxes is the number of TFBS for the given TF-cluster pair, there seems to be quite a bit of variability in the TFBS across species for some TFs that the authors selected to show. Fig 4f and the Extended Fig. 8 shows that within cluster difference is less than across cluster difference indicating conservation but this might also be true for gene expression. The authors can try averaging cluster marker expression per species per cluster for scRNAseq and plotting tSNE if they want to justify this comparison. Even then one could argue that the TFs with identified motifs tend to be the more crucial hence conserved ones and it's not fair to compare this to an unbiased gene-expression analysis. But as it is, the statement is weakly supported.

The scales in the legend for Fig. 4e (now Fig. 4h) are now labeled. The size of the boxes corresponds to the $-\log_{10}(\text{pval})$ of TFBS enrichment in subclass DMRs and is used to construct the tSNE in 4f (now 4j). Given the differences in sampling depth across species, we have emphasized TFBS enrichments that show clear conservation across species. As the reviewer notes, species differences are smaller than subclass differences within species as shown in the tSNE in Fig. 4j. As suggested, we compared conservation between species of TFBS activity, TF marker expression, and all marker expression (Fig. 4i). We find comparable conservation of TFBS activity and TF expression and greater divergence of expression of other subclass markers. While this may be expected, we are not aware of this being reported at this resolution of cell types in the brain using a technology that simultaneously profiles RNA and open chromatin from single nuclei.

12) One wonders if the “paradoxical observation” regarding species differences in consensus cell types is due to some technical issue due to number of cells or sequencing depth?

We agree with the reviewer that it is important to differentiate between technical and biological differences across species. In this study, we took a careful approach to sample the same brain region and use the same experimental techniques across species to help mitigate against technical artifacts. For each species, we identified primary motor cortex based on its distinctive cytoarchitecture, isolated single nuclei by FACS, and profiled transcriptomes and epigenomes using the same -omics methods. We have shown previously that in situ cell type proportions can be estimated based on nuclear sampling (Hodge et al. 2019 Nature), and the differences in subclass proportions that we report are consistent with the literature. In a companion paper, Yao et al. demonstrate a saturation in cell type detection between 60-80k nuclei, and we sample all three species to this depth using 10x Chromium v3. We find highly similar gene detection in human and mouse for all subclasses, although somewhat lower in marmoset due to lower depth sequencing and less complete genome annotation (ED Figure 1h-j). Despite somewhat lower sensitivity marmoset data, we find that transcriptomic profiles of cell types are more similar between human and marmoset than between primates and mouse. Moreover, we have included new ISH validations of species differences in L5 ET neurons that show markedly lower expression of *Cacna1c* and *Kcnc2* in mouse than human (ED Figure 9g) that matches predictions from 10x v3 data. These results support that we have not simply undercounted these transcripts in mouse using snRNA-seq. For epigenomic analyses, we have focused on species conservation because of the concerns raised by the reviewer. For example, Yao et al. 2020 report that TF binding site enrichment is sensitive to the number of cells sampled and the motif of interest. Deeper sampling of single nuclei for profiling of DNA methylation and accessible chromatin will be needed to be adequately powered to make strong claims of species differences in putative gene regulation. Finally, we have removed the phrase “paradoxical observation” since it is plausible that we are able to align consensus types based on shared co-expression patterns of thousands of genes and, over the course of millions of years, a subset of these genes have specialized expression in different species.

13) The finding that there is no “exclusive” Betz transcriptomic type—could that be due to a technical issue regarding clustering—does changing the resolution of clustering affect this conclusion. What about in the metacell analysis—does a Betz cell type emerge?

We reclustered the integrated L5 ET neurons across species at finer resolution and identified subpopulations that aligned across species (bottom row of plot below; top row corresponds to original clusters).

L5 ET neurons as a whole have the highest number of transcripts detected (total UMIs) compared to all other cortical cells, likely due to their larger size. Therefore, we expected that an ET subcluster corresponding to Betz cells in macaque and human would have even more UMIs detected and would also have distinct marker genes that could be used for in situ validation. However, UMIs were comparable across ET subclusters and marker expression differences were graded across clusters (shown in plots below). This was a surprising result, and we have included the following text in the discussion:

“Similarly, in a recent study of fronto-insular cortex, we identified a transcriptomic class consisting of ET neurons that included cells with distinctive spindle-shaped (von Economo neurons) and non-spindle cell bodies (Hodge et al., 2020). Thus, small or transient differences in gene expression can drive large differences in neuron morphology.”

14) "To discern if these TFs may preferentially bind to DNA in ChCs, we tested for TF motif enrichment in hypo-methylated (mCG) DMRs and AC sites genome-wide."

Why is this an interesting thing to check? If these TFs are marker genes for ChCs, they should be binding regions that are in open chromatin formation. The results are very much expected and perhaps belong in supplementary.

We hypothesized that cell type specificity of TFs could come at the level of TF expression and numbers of TF binding sites. Indeed, we find that RORA has increased expression and a significant enrichment of RORA binding sites relative to other motifs in ChC relative to basket cells. NFIB has increased expression but little enrichment of binding sites in ChC. Of course, RORA and NFIB may both contribute to ChC identity and differences from other fast-spiking interneurons and species specializations. But the expansion in the number of potential binding sites for RORA lends additional support to its central role, and we argue should make it a priority for follow-up work. We have laid out the motivation for this analysis in more detail in the results.

15) "Differentially accessible regions (DARs) between cell populations (Fig. 4b) were identified using the "find_all_diff" function (<https://github.com/yanwu2014/chromfunks>) and p-values calculated using a hypergeometric test."

Does this function take total cell accessibility into account as a latent variable? The probability of a read being present or not in a peak is influenced by the total number of reads in peaks for the given cell. Other recent single-cell ATAC-seq papers take this into account in their modelling. While the top DARs are likely to be similar, the final list of DARs may change considerably when this important covariate is taken into account.

This function to identify DARs calculates p values using Fisher's exact test and has been demonstrated in our previous publication (<https://www.nature.com/articles/nbt.4038>). However, as the reviewer suggests, it does not take in account total accessibility as a latent variable. We have modified our strategy to permit DAR discovery using this function while now accounting for differences in total accessibility between cells or clusters. This involves comparison of each cluster against a background set of cells (10,000 for human and 2,000 for marmoset) that were randomly selected from the remaining clusters and that shared similar total accessible peak counts. This way,

DARs were identified for each cluster against a comparable representative background. In support of this method, similar proportions of DARs were discovered across neuronal subtypes for human and marmoset that were highly consistent with proportions of hypomethylated DNA sites (Figure 4). The methods section has been updated accordingly.

16) "Peak regions were called independently for RNA cluster, subclass and class groupings using MACS2 software (<https://github.com/taoliu/MACS>) using the following options "--nomodel --shift 100 --ext 200 --qval 5e-2 -B --SPMR". Peak regions were combined across peak callings and used to generate a single peak count matrix (cell barcodes by chromosomal peak locations) using the "createPmat" function of SnapATAC."

The explanation is not very clear. Are the peaks called by combining all reads or per cluster identified using snapATAC? Was the peak calling performed separately for each subject? What exactly were the 'peak callings'? The authors should be more clear. It would be good to perform peak calling separately for each sample and see whether they differ substantially. If not, simple merging should be acceptable. But if they are quite different - most likely due to technical reasons- this would mean that for many peaks the sample will essentially be 1 if the peaks are just merged. In that case only the consensus peaks should be taken.

At the reviewer's suggestion, we have provided more details for our peak calling strategy in the methods section to improve clarity. Peak calling was performed per cluster groupings that included cells from both samples. Given the comparable sequencing depth and sampling sizes for each of the samples within species (Supplementary Table 14, Extended Data Fig. 5), we did not feel it necessary to run peak calling on each sample individually. To support this, we performed peak calling on each human sample and found that ~93% of peak regions called from H18.30.001 overlapped with those in H18.30.002. We also found that peak counts correlated highly across experiments (mean Pearson's correlation coefficient (r) of 0.99 for human, 0.98 for marmoset). Furthermore, dimension reduction using latent semantic indexing on the final binarized peak count matrices found samples to be well integrated across cell types for both human and marmoset SNARE-Seq2 data (Extended Figure 5). We have updated the results and methods sections to better address this.

17) The following sentence is difficult to understand:

"The AC-level clusters (Fig. 3a,b) that showed similar coverage across individual samples (Extended Data Fig. 6c-f) revealed regions of open chromatin that are extremely cell type specific."

Do the clusters with similar coverage across individuals have more DARs? If so, why would that be important? How is 'extremely cell type specific' any different than the term 'cluster marker'?

This sentence was originally intended to convey two messages: that AC clusters showed similar coverage across individuals; and that AC clusters showed distinct DARs. Since "extremely cell type

specific” is the same as “cluster marker”, we have removed this sentence from the manuscript. We have now significantly modified this section of the manuscript to improve clarity.

18) The authors state that log2 CPM data were put into Seurat. Were the log2 data further log-transformed then?

To integrate SMART-Seqv4 and 10x C3 datasets together for the human consensus cell types, we used $\log_2(\text{cpm} + 1)$ normalized matrices as input and did not utilize Seurat’s normalization functions on raw reads/counts matrices. Therefore, the input data for integration were only log-transformed once.

19) Seurat was used for Cv3 data but Pagoda was used for SnareSeq data making the results not entirely comparable.

Cv3 data was used in the prediction of Marmoset SNARE-seq2 cluster identities, while SMART-Seqv4 was used for prediction of human SNARE-Seq2 clusters. In this way, we were able to generate SNARE-seq2 cluster identities directly matching the corresponding species-specific reference taxonomies. PAGODA2 was only used for PCA analysis and the independent clustering of SNARE-Seq2 data as supporting evidence for the accuracy of cell type annotations predicted from the reference taxonomies (Extended Data Fig. 5). Any direct cross-platform comparisons were performed on counts normalized using Seurat (Extended Data Fig. 5).

20) More details on the methylation analysis is needed. Were covariates included in calling DARs? In the absence of the companion paper it is difficult to know exactly what was done. And the examples shown of it “working” seem a bit cherry picked—the marker genes from the previous figures are not shown and the evolutionary comparisons are done in a different manner.

Calling of differentially methylated regions (DMRs) is described in the Methods section. Briefly, we merged single nuclei methylation data according to different levels of cell types (DNAm clusters and cell subclasses). DMRs were called within each cell type level using methylpy (<https://github.com/yupenghe/methylpy>) as previously described (He et al., 2017; Liu et al., 2020).

DARs were revised to account for total accessibility per nucleus as described in response to reviewer comment #15 above. It is unclear to which figure and markers the reviewer is referring? We have substantially revised the epigenomics figures and have attempted to provide example genes that highlight the broader biological findings. For example, we now include genome tracks for a Lamp5

subtype marker *KIT* and show that DNA methylation and AC data provide convergent evidence for a putative regulatory region within the gene body (Fig. 4d). TFs were selected for Fig. 4h to highlight the general trend of conservation across species.

21) Figure 5 a&b legends appear switched.

The figure legend has been corrected. We thank the reviewer for catching this oversight.

22) The ephys data descriptions are very convoluted and difficult to follow. It is not even clear what the N is for each experiment.

We have made edits in the results, methods and figure legends to improve clarity and have indicated the n for each experiment.

23) For Figure 7a, can the authors speculate why mouse would have the second greatest # of species-specific marker genes for L5 ET vs IT? (i.e. more than macaque or marmoset?)

Mouse had the fewest total number of L5 ET markers (~300) vs. 300-400 for the primates. Despite this, there are more mouse-specific marker genes because mouse is more distinct from all the primates than primate species are from each other. In other words, fewer markers are species-specific among primates because many markers are shared among these species with a more recent common ancestor. In light of that, it is interesting that human has the greatest number of species-specific markers, perhaps pointing toward greater specialization in human L5 ET neurons than other primates.

24) It is not clear how many cells, sections, and individuals were included for each FISH/IHC confirmation. In most cases we see one representative nuclei in the figure.

We have revised the methods section to include details of the number of sections and donors included in each mFISH/IF experiment.

25) The finding of neuronal genes in oligos is quite interesting but the interpretation stated below has no reference cited about phagocytosis. "This may represent an oligodendrocyte type that

expresses neuronal genes or could represent phagocytosis of parts of neurons and accompanying transcripts that are sequestered in phagolysosomes adjacent to nuclei."

We have revised the results text: "This may represent an oligodendrocyte type that expresses neuronal genes or could represent phagocytosis of parts of neurons and accompanying transcripts, similar to the reported phagocytic function of some OPCs (Falcão et al., 2018)." In addition to neuronal transcripts, this oligo type expresses markers of oligo precursors, such as *SOX10*.

Referee #2 (Remarks to the Author):

A. SUMMARY OF THE KEY RESULTS

The focus of the Bakken et al. manuscript is on the cell diversity of the primary motor cortex (M1) of human, marmoset monkey and mouse. M1 is essential for voluntary fine motor control and have various species differences in terms of direct and indirect innervation of the spinal cord motoneurons and interneurons. L5 of carnivore and primate M1 contains exceptionally large "giganto-cellular" corticospinal neurons (Betz cells in primates). Some primate Betz cells directly synapse onto alpha motor neurons, whereas in cats and rodents these neurons synapse instead onto spinal interneurons. The paper compares cellular components in these three species and it provides cross-species consensus cell type classification and inference of conserved cell type properties across species. This is a huge achievement and I could only read the paper with admiration and enthusiasm. This is a huge operation and it is beginning to bear fruits in these projects that are way beyond the capability of a single laboratory. This comparative evolutionary approach provides a stable platform to define the cellular architecture of our brain and to discover species-specific adaptations. This paper decided to compare one area in three species, but one could have argued for comparing several areas in one species or several areas during development. The study is focusing on M1. Several approaches would have been justified, but starting in M1 is not a bad choice. This is a functionally and anatomically relatively conserved cortical region across mammals. The study also allows the comparison of a variety of methods on similarly isolated tissues.

The paper reports differences in cell type proportions, gene expression, DNA methylation, and chromatin state. The study dedicates a considerable effort to characterize the exceptionally large "giganto-cellular" corticospinal neurons (Betz cells in primates) in M1 in these three species. This is an extremely interesting cell population. I understand that the mouse datasets are also reported in a companion paper, that I did not receive.

The mouse datasets are described in Yao et al. 2020 that is available as a preprint:

<https://www.biorxiv.org/content/10.1101/2020.02.29.970558v2>

B. ORIGINALITY AND SIGNIFICANCE: IF NOT NOVEL, PLEASE INCLUDE REFERENCE

The paper is much more than just presenting a huge dataset that no other institution could deliver. The manuscript also uses this dataset to try to answer some fundamental principles of cortical organization. I believe that this is why this paper deserves to be published in Nature.

1. The paper describes similar cellular complexity on the order of 100 cell types was seen in all three species. The study identified some core conserved genes and make strong predictions about the TF code for cell types and the genes responsible for their evolutionarily constrained functions. Exploring the links between genes and cellular phenotypes for conserved and divergent features will have to be tested, but this can take much more effort and much more detailed analysis, perhaps by extending these studies to birds and reptiles.
2. The paper compares different methodologies and their impact on the outcome of the analysis. They present important comparisons between plate-based (SSv4) and droplet-based (Cv3) RNA-seq of human nuclei. The authors compared results between approximately 10,000 SSv4 and 100,000 Cv3 nuclei. The study reports that on average, SSv4 detected 30% more genes per nucleus and enabled comparisons of isoform usage between cell types. This had 20-fold greater sequencing depth, but SSv4 cost 10 times as much as Cv3 and did not allow detection of additional cell types.
3. The paper describes relative similarities and differences between the three species examined. As expected, the more closely related species are more similar to one another. It would have been interesting to explore whether these similarities and differences are on the same scale in different cortical areas.
4. One important confirmation of previous studies was that the ratio of glutamatergic excitatory projection neurons compared to GABAergic inhibitory interneurons was 2:1 in human compared to 3:1 in marmoset and 5:1 in mouse. This shift in the overall excitation-inhibition balance of the cortex was described by other methods, but this is perhaps the most elegant and powerful demonstration of these principles.

5. Interestingly the relative proportions of GABAergic neuron subclasses and types were similar across species.

6. It was also suggested by previous studies using different methods that there is a large increase in the proportion of L2 and L3 IT in human compared to mouse and marmoset, nevertheless it is very reassuring to see all this with a more sophisticated and reliable method.

7. The paper has some important observations on Layer 4 in M1 in the three species. M1 does have L4-like cells based on the transcriptomic signature, but only a subset of the types compared to granular cortical areas, at much lower density, and scattered rather than aggregated into a tight layer.

8. The study identifies the transcriptomic cluster corresponding to Betz cells and uses this further to understand gene expression that may underlie their distinctive properties. The study concludes that there does not appear to be an exclusively Betz transcriptomic type. The authors find more than one ET cluster contains neurons with Betz morphology and conclude that Betz cells may not in fact be completely restricted to M1 but distribute across other proximal motor-related areas that contribute to the pyramidal tract.

These are all very fundamental findings and each point could take years to further study in detail. The study is balancing between the time pressures of releasing this important dataset for the general scientific community, but they also would like to get the most important and most significant findings already analyzed and described. I think this balancing act is successful in this case.

C. DATA & METHODOLOGY; D. APPROPRIATE USE OF STATISTICS

The authors have a well-established pipeline with a distinguished advisory board, therefore works according to the most modern tools and standards.

E. CONCLUSIONS: ROBUSTNESS, VALIDITY, RELIABILITY

Does the manuscript have flaws, which should prohibit its publication? If so, please provide details.

The manuscript contains huge amount of work. Very impressive data, collected with the state of the art methodologies on a scale that a normal academic-research laboratory could not afford and completely out of reach. It is like sending up a satellite to observe the continents and report back to teams that used small-scale approaches to map the continents. We get staggering insights and views! However, there is also a challenge what to describe and how to interpret the initial results. As I outlined above (1-8) the paper contains some fundamental issues and these are approached responsibly and with understanding. However, I do not have time to go through all the possible ramifications that could be done with this data. Almost all of the above (1-8) points could be extended much further and I am sure the authors are all aware of the possibilities. Moreover, it would take a lot of time to detail the directions that the authors could take with each of these directions.

F. SUGGESTED IMPROVEMENTS: EXPERIMENTS, DATA FOR POSSIBLE REVISION

I shall concentrate my comments on the issue of Betz cells (point 8 above), but I could have picked any of the other points, they could be developed much further with time. The presented analysis is just the tip of the iceberg.

(a) Betz cell and L5 ET neurons are used interchangeably in several parts of the text and this should be changed to discern the potential differences between Betz cells and large L5 neurons. For example, many of the features highlighted within L5 ET neurons are found under the heading “Primate Betz cell specialization”, the differences between ET neurons and IT neurons should be written elsewhere or the heading should be changed to reflect the findings found in ET neurons or Betz cells.

Since Betz cells were identified along with smaller pyramidal neurons in the same L5 ET clusters, we focused our transcriptomic analyses on comparisons between ET and IT and between ET across species. Likewise, for Patch-seq we only have morphological information for a subset of recorded cells so we have focused our comparisons to ET vs. IT and ET neurons across species. We have relabeled the heading “L5 ET neuron specialization” to reflect this.

(b) In the main text, the authors mention they found x2 Betz cell clusters “Exc L5 FEZF2 ASGR2” & “Exc FEZF2 CSN1S1”, however in their figures they state that neurons found within layer 3 of the human motor cortex also clustered to the Betz Exc L5 FEZF2 ASGR2 cluster. The authors should, therefore, either not refer to this as a Betz cell cluster or strictly use the “Exc L3-L5 FEZF2 ASGR2” nomenclature found in their figure throughout the entirety of the text for clarity. Furthermore, a better explanation is needed to explain why L3 neurons and Betz cells cluster together in their dataset.

The cluster nomenclature used in the present study reflects the laminar dissections used for SMART-seq4 data generation in human. The reviewer is correct that some of the nuclei in the Exc L3-5 FEZF2 ASGR2 cluster were captured in L3 dissections (Fig. 1c); however, the majority of nuclei in this cluster were captured in L5 dissections. In the tissue sections that we have examined, FISH staining for the ET marker *POU3F1* is restricted to layer 5 in M1 with some stained cells being more superficial and closer to the border with L3 than others. In particular, very large *POU3F1*-expressing cells with Betz morphology appear to be more abundant in deep L5, consistent with descriptions of these cells in the literature. In the case of the L3-5 FEZF2 ASGR2 cluster, we think that the nuclei captured in L3 dissections represent some of these more superficial L5 *POU3F1*-expressing ET cells that were likely incidentally captured in L3 dissections owing to the challenges of identifying a clear boundary between L3 and L5 in fluorescent Nissl stained sections of agranular cortex. As we describe in the present study, the L5 ET clusters that we examined, including the L3-5 FEZF2 ASGR2 cluster, contain a mixture of very large cells with Betz morphology and neurons with pyramidal morphologies consistent with intermixing of these deep and superficial L5 neurons. We have reported similar findings in the fronto-insular cortex where we showed that von Economo neurons form a single transcriptomic cluster with other L5 ET neurons having pyramidal or fork morphologies (Hodge et al., 2020). We have updated the text to ensure that that name of the Exc L3-5 FEZF2 ASGR2 is consistent with the figures.

(c) The authors find several Betz cell clusters, however, they fail to show convincing validation. They only show x2 images of RNAscope staining on Betz cells and draw the conclusion that this represents the transcriptomic profile of their Betz clusters. It is not clear why they selected those probes for staining, are the probe signatures unique to Betz clusters? Or do they just show transcripts that are particularly enriched? The rationale behind these probes should be explained in the text. The best approach would be to improve upon extended data figure 10e to show the distribution and abundances on a larger scale. Moreover, the probes they have selected (Neurofilament, GRIN3A, *POU3F1* and *SERPINE2*) also individually stain many other cells in the primary motor cortex so an accompanying image showing the staining pattern of Betz cells vs non-Betz L5 neurons/non-L5 neurons is needed.

To examine Betz cells using combined IF and mFISH, we selected a combinatorial probe set that was specific for each potential Betz-containing cluster based on snRNA-seq data. A violin plot showing expression of these genes is presented in Extended Data Fig. 9e. Each combinatorial probe set included *POU3F1*, which we find is specifically expressed in ET neurons. Additionally, we used SMI-32 IF to highlight the morphologies of cells labeled with these cluster-specific probe combinations. *NEFH*, the gene that encodes SMI-32 protein, is expressed in all L5 ET clusters (Extended Data Fig. 9e). We agree with the reviewer that these probes can stain other cell types when only a single probe is examined (e.g. GRIN3A is expressed in many interneuron types and several other non-ET excitatory neuron types); however, in combination, they are unique to each of the L5 ET types that we examined. We have updated the text to clarify that we used cluster-specific combinatorial markers to examine these cell types. We agree with the reviewer that it would be interesting to

examine the distribution of these cells on a larger scale, but our current imaging capability limits us to looking at relatively small tissue sections that can be handled on standard-sized glass slides. We have updated the methods section to state that we examined L5 ET clusters only within the dome of the gyrus corresponding to the presumptive trunk-lower limb region of human M1.

(d) The ISH images presented are not clear and often positive staining cannot be easily seen. For example, within extended data figure 10f it is not clear that both Betz cells are POU3F1 positive despite the text stating otherwise. Clearer images should be used; maybe even single channels and a composite to allow the reader to better investigate the staining.

We have updated the images the reviewer mentioned to more clearly show mFISH staining patterns in L5 ET cells. We now include higher magnification images to demonstrate staining for individual RNA spots alongside lower magnification images to illustrate the morphology of these cells with SMI-32 counterstaining.

(e) The authors need to explain how they histologically determined Betz cells from large L5 neurons in their ISH validation. There are at least 5 histological defining features, but it appears the authors only use size and location that can commonly misidentify Betz cells.

We used several criteria for distinguishing Betz cells from other L5 neurons, including gigantocellular somata (>40 μm diameter) and (when possible) the presence of perisomatic dendrites. However, several of the other criteria used in the literature (Rivara et al., 2003) for characterizing Betz cells cannot be easily applied to the type of combined mFISH and IF imaging that we used to examine these cells in the present study. For example, abundant lipofuscin is used to distinguish Betz cells from other large L5 neurons when these cells are examined using Nissl staining. However, with fluorescent imaging, lipofuscin is apparent in many cells of varying sizes and is not a particularly distinguishing feature. We also find that lipofuscin abundance varies widely across different human donors. Likewise, rough endoplasmic reticulum is not visible with the imaging method we used and we could not reliably visualize a more prominent nucleolus with the DAPI counterstaining we employed when comparing Betz cells with other L5 cells. Furthermore, RNAscope mFISH necessitates the use of relatively thin tissue sections (14-16 μm), which frequently results in bisection of these large cells and their associated dendrites making the detection of perisomatic dendrites a challenging criteria to consistently apply across all of the cells that we examined, but we looked at this feature where possible.

(f) More information from the literature is needed to explain the presence of Betz cells in the premotor cortex since it is known that the vast majority are found in the primary motor cortex. Again, a detailed neurohistological description is needed to increase confidence that the human

motor neurons from the premotor cortex used for PATCH-Seq were indeed Betz cells, especially since it is not strongly supported in the literature if Betz cells truly exist in the premotor cortex. The citation used here is in reference to an electrophysiological study using only macaques that studied large pyramidal neurons in the premotor cortex and does not mention humans. Large pyramidal neurons are found across the neocortex and this neuron selected for PATCH-Seq may just be a "regular" large pyramidal neuron.

The literature indeed suggests that Betz cells are enriched in the primary motor cortex, but they are also present in premotor cortex Area 6 (Rivara et al., 2003; Wise 1985; White et al., 1997)). Consistent with this, we find sparse neurons with Betz morphologies in the premotor cortex (Figure X -superior frontal gyrus) near the primary motor cortex in the Allen Human Brain Reference Atlas. Three of the five hallmarks of Betz cells (gigantocellular somata, horizontal, perisomatic dendrites, abundant rough endoplasmic reticulum; Rivara et al., 2003) were readily apparent in pyramidal neurons in the premotor cortex. We could not confidently make assessments of the final two hallmarks of Betz cells (abundant lipofuscin and prominent nucleolus) in these Nissl stained sections. Additionally, as can be seen in the biocytin images in figure 7, the recorded neurons possessed large somata with many perisomatic dendrites. Additional histological hallmarks of Betz cells cannot be assessed in biocytin filled neurons. We have incorporated this description into the manuscript.

A) Neurons with Betz morphology can be found in human premotor cortex near M1. B) Higher magnification image of neurons with Betz morphology (dark Nissl stain indicating abundant rough ER, perisomatic dendrites, large soma) in B) superior frontal gyrus and C) primary motor cortex. Arrows denote perisomatic dendrites. D) High magnification image of biocytin filled neuron in human premotor cortex with Betz morphology and smaller pyramidal neuron (upper left). For Betz cell, note large soma and perisomatic dendrites. Biocytin fills did not permit the assessment of other Betz hallmarks (lipofuscin, large nucleolus, rough ER).

(g) A selection criterion is needed to outline how the authors selected Betz cells for macaque patch-seq analysis, for example, the macaque Betz cell used had a soma size of >65μm; size isn't the defining feature of a Betz cell so an explanation on why they believe this neuron is a Betz cell is needed, including information on how they prepared this tissue block. They should also, look at the literature to find the size of an average macaque Betz cell as size varies a lot between species.

We have now added a description of the criterion for targeting neurons for patch clamp analysis to the methods section. Several of the histological markers used to identify Betz cells (prominent rough endoplasmic reticulum, conspicuous nucleolus) were incompatible with performing physiological recordings. Thus, we utilized soma size (> 40 microns in diameter) as our primary criterion given that somatic size (volume, and/or width/height) reasonably separates Betz cells from other pyramidal neurons in macaques and humans (Rivara et al., 2003; Sherwood et al., 2003, Tigges et al., 1990). To visualize L5 pyramidal neurons in the densely myelinated primary motor cortex, we used viral labeling in organotypic slice cultures. This permitted us to target the very largest neurons for Patch-seq/patch clamp. Occasionally in the fluorescent image we could see three additional Betz cell hallmarks - large tap-root dendrites and dendrites horizontally emanating directly from the somatic compartment. In many of these neurons substantial lipofuscin could be observed. Finally, the diameter of the biocytin filled neuron in the example (Figure 7) is at the upper end of the range of diameters of corticospinal neurons in macaque area 4 (20-60 μm ; Murray and Coulter 1981). Information on tissue preparation can be found under the heading "Brain slice preparation" in the methods section. We have added this description of targeting neurons for patch clamp analysis to the methods section.

(h) Clusters "Exc L5 FEZF2 ASGR2" & "Exc FEZF2 CSN1S1" should be compared with other L5 glutamatergic clusters to discern potential differences and reveal potential Betz specific markers. This would aid in the author's goal to study how Betz cells differ from other neuronal subtypes. At present Betz cell clusters are only compared to IT neurons.

We thank the reviewer for this suggestion. We generated a list of DEGs that were identified by performing a pairwise ROC test between L5 ET and each glutamatergic subclass for the human 10x Cv3 data. We performed a GO analysis for biological processes with the PANTHER v14 classification system on the 232 genes that were L5 ET enriched across all pairwise comparisons and found that axon guidance and EGF-associated injury response genes were significantly overrepresented. The upset plot below shows that many genes show consistent expression differences between L5 ET neurons and other glutamatergic subclasses (>200 shared markers), while some genes are selectively differentially expressed between L5 ET and one or more subclasses.

(i) The authors also presume and group Betz cells as layer V extratelencephalic neurons, while this makes sense from a global approach where not a lot is known about the transcriptome of the Betz cell there may be subpopulations of Betz cells that project to other regions and therefore could be accidentally excluded from this analysis when this dogma is implemented.

We agree that we cannot exclude the possibility that some cells with Betz morphology may not project to extratelencephalic regions and may have a distinct transcriptome from ET-like Betz cells. In situ validation work in this study focused on the trunk/limb region of M1 and imaged only a few cells per section due to their rarity in human cortex. In these sections, all large SMI-32 staining L5 neurons expressed *POU3F1*, a marker of L5 ET neurons. However, Betz cells in other subregions of M1 may have different properties, and future work focused on these cells should more carefully assess their features across M1.

(j) Authors state that the ion channel subunits found may reflect Betz cell physiology, but again, as the Betz cell clusters also contained other neuronal cell types this should be spatially validated with ISH.

We thank the reviewer for this suggestion and we have provided additional data in the revised manuscript to compare expression of 2 of the ion channel genes (*CACNA1C* and *KCNC2*) shown in Fig. 7 between mouse and human using combined mFISH for each of these genes along with probes for the ET marker *POU3F1* and IF for SMI-32. Consistent with snRNA-seq data we find higher expression of these genes in human L5 ET cells with Betz-like morphology than in mouse L5 ET cells.

(k) The authors found that ROBO, SLIT and EPHRIN are found at higher amounts in primate ET neurons when compared with primate IT neurons and attributed to axon projection distance. ISH should be performed to see if this is unique to ET neurons or to Betz cells, or if Betz cells with the longest axons express more of these transcripts.

Close to 100% of neurons in the human L5 ET clusters express these axon guidance genes. The heatmap below shows the proportion of nuclei expressing more than one transcript of the respective genes (red = 100%). Therefore, we conclude that those neurons likely include Betz cells in addition to non-Betz ET neurons.

We agree with the reviewer that it would be interesting to correlate expression of these transcripts with axon length. However, the methods that we have access to do not permit such comparisons as they don't provide any direct information about axon length; therefore, we cannot make any direct inference about the relationship between the amount of these transcripts and axon length using our ISH methods.

(l) The paper argues that "axon guidance-associated genes are enriched in Betz-containing ET neuron types in primates, possibly explaining why Betz cells in primates directly contact spinal motor neurons rather than spinal interneurons as in rodents." However, all these studies were done at adult stages. Most of the actual guidance factors might have been gone by this stage. I would tune down these claims, since in best case these specific gene expression patterns are involved in the synaptic maintenance, but not the formation.

We have removed this point from the discussion and revised the results to point to a role in synaptic maintenance:

“Interestingly, many genes were associated with axon guidance, including from the Robo, Slit and Ephrin families. These genes may contribute to maintaining cortico-motoneuronal connections associated with increasingly dexterous fine motor control across these species (Lemon, 2008).”

G. REFERENCES: APPROPRIATE CREDIT TO PREVIOUS WORK

This is the first such comprehensive reports. As such it is highly original. Yes, some aspects of the debate started decades before this paper, but now one can examine these in a more comprehensive fashion. The authors cite the older literature and put things into context.

On a more subjective note, do you feel that the results presented are of immediate interest to many people in your own discipline, or to people from several disciplines?

All neuroscientists should be interested in this data in some form. I am surprised that there was a relative lack of interest in bioRxiv.org, only one comment so far:

Comment from BioRxiv:

Miguel Angel Garcia-Cabezas • a month ago

The authors of the manuscript entitled “Evolution of cellular diversity in primary motor cortex of human, marmoset, monkey, and mouse” found that “transcriptomically similar cell types were found at similar cortical depths in M1 and MTG, and the OTOGL and LINC01202 types were located in deep L3 and superficial L5 in M1”. In the Discussion they state that “M1 is an agranular cortex lacking a L4, although a recent study demonstrated that there are neurons with L4-like properties in mouse¹⁴. Here we confirm and extend this finding in human M1. We find a L4-like neuron type in M1 that aligns to a L4 type in human MTG and is scattered between the deep part of L3 and the superficial part of L5 where L4 would be if aggregated into a layer”.

However, within the published literature, the existence of layer IV in the human primary motor cortex was first described by Ramón y Cajal [Estudios sobre la corteza cerebral humana. II La corteza motriz del hombre y mamíferos superiores. Rev Trim Microg. 1899 (4): 117–200]. This finding was later confirmed by Marín-Padilla [Prenatal and early postnatal ontogenesis of the human motor cortex: a Golgi study. I. The sequential development of the cortical layers. Brain Res. 1970; 23 (2): 167-83]. Layer IV has also been described for the primary motor cortex of rhesus macaques qualitatively [Gatter et al. The intrinsic connections of the cortex of area 4 of the monkey. Brain. 1978; 101 (3): 513-41]. More recently, a detailed study in rhesus monkeys with analysis at the

cellular and subcellular levels and rigorous analytic methods, provided strong quantitative evidence for the presence and cellular features of layer IV, with discussion of the relevant history and ideas

[García-Cabezas & Barbas. Area 4 has layer IV in adult primates. *Eur J Neurosci*. 2014; 39 (11): 1824-34; and Barbas and Garcia-Cabezas, Motor cortex layer 4: less is more. *Trends Neurosci*. 2015; 38(5): 259-61].

The authors can claim that they confirm the existence of layer IV in the primary motor area of the human cortex, but they don't extend "this finding in human M1".

Thank you for pointing us to this comment. We have added some of these references to the discussion.

If you recommend publication, please outline, in a paragraph or so, what you consider to be the outstanding features.

This paper should be received by Nature with open arms. It is a landmark study that not only contains a huge unique dataset that will attract hundreds of citations, but it already contains some analysis that is answering some fundamental questions. A key result of the current study is the identification of a consensus classification of cell types across species that allows the comparison of relative similarities in human compared to common mammalian model organisms in biomedical research. There are lots of important findings (see 1-8 points above), including Layer 5 corticospinal Betz cells in non-human primate and human and characterization of their highly specialized physiology and anatomy. The data presented in the paper allow a targeted search for genes responsible for species specializations such as the distinctive anatomy, physiology and axonal projections of Betz cells, large corticospinal neurons in primates that are responsible for voluntary fine motor control.

H. CLARITY AND CONTEXT

All fine. It is very well written and put together. It is not an easy read, but it would be very difficult to find things to considerably improve the manuscript.

References

- Falcão, A. M., van Bruggen, D., Marques, S., Meijer, M., Jäkel, S., Agirre, E., Samudiyata, Floriddia, E. M., Vanichkina, D. P., Ffrench-Constant, C., Williams, A., Guerreiro-Cacais, A. O., & Castelo-Branco, G. (2018). Disease-specific oligodendrocyte lineage cells arise in multiple sclerosis. *Nature Medicine*, *24*(12), 1837–1844.
- He, Y., Hariharan, M., Gorkin, D. U., Dickel, D. E., Luo, C., Castanon, R. G., Nery, J. R., Lee, A. Y., Williams, B. A., Trout, D., Amrhein, H., Fang, R., Chen, H., Li, B., Visel, A., Pennacchio, L. A., Ren, B., & Ecker, J. R. (2017). Spatiotemporal DNA Methylome Dynamics of the Developing Mammalian Fetus. In *bioRxiv* (p. 166744). <https://doi.org/10.1101/166744>
- Lemon, R. N. (2008). Descending pathways in motor control. *Annual Review of Neuroscience*, *31*, 195–218.
- Liu, H., Zhou, J., Tian, W., Luo, C., Bartlett, A., Aldridge, A., Lucero, J., Osteen, J. K., Nery, J. R., Chen, H., Rivkin, A., Castanon, R. G., Clock, B., Li, Y. E., Hou, X., Poirion, O. B., Preissl, S., O'Connor, C., Boggeman, L., ... Ecker, J. R. (2020). DNA Methylation Atlas of the Mouse Brain at Single-Cell Resolution. In *bioRxiv* (p. 2020.04.30.069377). <https://doi.org/10.1101/2020.04.30.069377>

Reviewer Reports on the First Revision:

Referees' comments:

Referee #1 (Remarks to the Author):

I appreciate the efforts of the authors to answer all of my concerns. The manuscript is much easier to read and understand now and the evolutionary comparisons are made more clear. The additional details/clarifications on methods as well as inclusion of statistical tests are much appreciated and will be welcomed by the scientific community.

Referee #2 (Remarks to the Author):

As I mentioned in my first review, this paper is much more than just presenting a huge dataset that required huge financial resources that no other institution could deliver. Social media now call

these colossal papers as “expensive data dump”. This paper is not one of these; a lot of careful analysis and thought went into it and provide deep insight. The manuscript uses a large unique dataset to answer some fundamental principles of cortical organization (see my original comments 1-8). I believe that this is why this paper deserves to be published in Nature.

I concentrated my comments on the issue of Betz cells (point 8 in my original review), but I could have picked any of the other 1-7 points, they could be developed much further with time. As I mentioned, the presented analysis is just the tip of the iceberg. Bakken and colleagues responded to most of my criticisms.

- (a) – done
- (b) – done
- (c) – clarifications were made on the cluster-specific combinatorial markers and the areas studied.
- (d) – better documentation with better images were now included
- (e) - the reduced criteria used for distinguishing Betz cells from other L5 neurons is now explained (gigantocellular somata (>40 um diameter) and (when possible) the presence of perisomatic dendrites) and it is explained why the authors could not always use the other criteria (abundant rough endoplasmic reticulum; abundant lipofuscin and prominent nucleolus).
- (f) – further evidence for the presence of neurons with Betz morphology in human premotor cortex near M1 is now presented together with previous literature.
- (g) – The authors spell out that they utilized soma size (> 40 microns in diameter) as primary criterion for their selection criteria for their macaque patch-seq analysis.
- (h) – I love the newly generated a list of DEGs that were identified by performing a pairwise ROC test between L5 ET and each glutamatergic subclass for the human 10x Cv3 data as I recommended in my first review comments. I think this will add to the paper a great deal.
- (i) – The explanation and consideration for the presence of some cells with Betz morphology may not project to extratelencephalic regions and may have a distinct transcriptome from ET-like Betz cells is reasonable.
- (j) – Additional data to Fig. 7 is reassuring.
- (k) – OK
- (l) – The distinction between synaptic development and synaptic maintenance has been clarified on the statements obtained from adult.

The authors also included some suggested references from a comment that I picked up from a comment made at bioRxiv.org. It is interesting that there were no further comments made since April (5 months!). bioRxiv.org could present a superb forum to discuss details of such huge and detailed papers by specialists, but there seem to be a lack of communication or interest and journals do not get help through this route. Of course, this is nothing to do with the authors and this particular paper that I find excellent resource and very valuable contribution to the broad field. I have no further criticisms and recommend publication.

Zoltan Molnar

Author Rebuttals to First Revision:

N/A